# Loss Landscape of Shallow ReLU-like Neural Networks: Stationary Points, Saddle Escape, and Network Embedding

**Frank Zhengqing Wu**[1][*]    **Berfin Şimşek**[2]    **François Gaston Ged**[3][*]

[1] École Polytechnique Fédérale de Lausanne, Lausanne (EPFL), Switzerland
[2] New York University, New York, USA
[3] University of Vienna, Vienna, Austria

## Abstract

In this paper, we study the loss landscape of one-hidden-layer neural networks with ReLU-like activation functions trained with the empirical squared loss using gradient descent (GD). We identify the stationary points of such networks, which significantly slow down loss decrease during training. To capture such points while accounting for the non-differentiability of the loss, the stationary points that we study are directional stationary points, rather than other notions like Clarke stationary points. We show that, if a stationary point does not contain "escape neurons", which are defined with first-order conditions, it must be a local minimum. Moreover, for the scalar-output case, the presence of an escape neuron guarantees that the stationary point is not a local minimum. Our results refine the description of the *saddle-to-saddle* training process starting from infinitesimally small (vanishing) initialization for shallow ReLU-like networks: By precluding the saddle escape types that previous works did not rule out, we advance one step closer to a complete picture of the entire dynamics. Moreover, we are also able to fully discuss how network embedding, which is to instantiate a narrower network with a wider network, reshapes the stationary points.

## 1 Introduction

Understanding the training process of neural networks calls for insights into their loss landscapes. Characterization of the stationary points is a crucial aspect of these studies. In this paper, we investigate the stationary points of the loss of a one-hidden-layer neural network with ReLU-like activation functions. The non-differentiability of the activation function renders such problems non-trivial. It is worth noting that, although the non-differentiable areas only take up zero Lebesgue measure in the parameter space, such areas are often visited by GD and thus should not be neglected.

In particular, we discover that any loss-decreasing path starting from a stationary point must involve changes in the parameters of "escape neurons" (defined in Definition 4.1). The absence of escape neurons guarantees the stationary point to become a local minimum. Our results provide insight into the saddle escape process with minimal assumptions. Supplementing Maennel et al. (2018); Boursier et al. (2022); Chistikov et al. (2024); Boursier and Flammarion (2024); Kumar and Haupt (2024), our results lead to a fuller understanding of the saddle-to-saddle training dynamics, a typical dynamical pattern resulted from vanishing initialization. More specifically, those previous works studied the behavior of gradient flow near a specific type of saddle point (Kumar and Haupt, 2024), and this work precludes the existence of other types in the general case.

We are also able to systematically describe how network embedding reshapes the stationary points by examining whether the "escape neurons" are generated from the embedding process. This directly extends the discussion by Fukumizu et al. (2019) to non-differentiable cases.

---

[*]Correspondence to `zhengqing.wu@epfl.ch`, `fged.math@gmail.com`.

## 1.1 RELATED WORK

**Stationary Points.** Stationary points abound on the loss landscape of neural networks. He et al. (2020); Sharifnassab et al. (2020); Liu et al. (2021); Arjevani and Field (2021); Şimşek et al. (2023) have shown that saddle points and spurious local minima exist in the landscape of non-linear networks. Stationary points may significantly affect training dynamics, leading to plateaus in the loss curve (Saad and Solla, 1995; Amari, 1998; Jacot et al., 2022; Pesme and Flammarion, 2024). When studying these stationary points, it is of both theoretical and practical interest (Fukumizu and Amari, 2000; Şimşek et al., 2021) to discriminate between their different types (Achour et al., 2022), for they affect gradient descent differently (Lee et al., 2017; 2016). Specifically, while local minima and non-strict saddles are not escapable (Achour et al., 2021; Lee et al., 2016; Achour et al., 2022), strict saddles are mostly escapable under mild conditions (Lee et al., 2017; 2016; Jin et al., 2017; Daneshmand et al., 2018; Ziyin et al., 2023). In our setting, the non-differentiability complicates the analysis. To tame such difficulty, previous works took various simplifications, such as only studying the differentiable areas (Zhou and Liang, 2018; He et al., 2020; Sahs et al., 2022), only studying the non-differentiable stationary points yielded by specific constructions (Liu et al., 2021), and only studying the first two orders of the derivatives (Yun et al., 2019). These theories appeared inconclusive due to such simplifications. So far, a systematic characterization of the potentially non-differentiable stationary points in our setting has yet to be established.

In this work, we consider stationary points to be where the one-sided directional derivative (ODD) toward any direction is non-negative, which is known as directional stationary points (Li et al., 2020). This notion effectively captures the points on the loss landscape that slow down GD and create loss plateaus, whether differentiable or not (see Section 3.2 and Appendix C.2). Such a property of this notion remained unnoticed by previous works. Earlier literature on ReLU network training dynamics (Boursier et al., 2022; Lee et al., 2022; Kumar and Haupt, 2024) observed such GD-stagnating points for specific cases but did not provide conditions to identify them in broader setup. Notably, our notion of stationarity also contrasts with other methods based on Clarke subdifferential (Wang et al., 2022; Davis et al., 2020a), subgradient, or right-hand derivative (Cheridito et al., 2022), which may be less suitable for characterizing GD stationarity (see Section 3.2.1).

**Training Dynamics.** Different initialization scales lead to different training dynamics. In the lazy-training regime (Chizat et al., 2019), which has relatively large initialization scales, the dynamics is well captured by the neural tangent kernel theory (Jacot et al., 2018; Allen-Zhu et al., 2019; Arora et al., 2019). In the regime of vanishing initialization, the training exhibits a feature learning behavior (Yang and Hu, 2020), which still awaits a generic theoretical account. However, it has been widely observed that the training dynamics in this regime often displays a saddle-to-saddle pattern, meaning the loss will experience intermittent steep declines punctuated by plateaus where it remains relatively unchanged. We will describe this process for our setting, furthering the discussions from Maennel et al. (2018); Boursier et al. (2022); Chistikov et al. (2024); Boursier and Flammarion (2024); Kumar and Haupt (2024). It is noteworthy that the training dynamics often unveils the implicit biases of the optimization algorithms. In the saddle-to-saddle regime, the learned functions often prefer lower parameter ranks (Maennel et al., 2018; Luo et al., 2021; Jacot et al., 2022) or smaller parameter norms (Boursier et al., 2022; Pesme and Flammarion, 2024), which could be beneficial for generalization.

**Network Embedding.** Network embedding provides a perspective for understanding how the optimization landscape transforms given more parameters, shedding light on the merit of over-parameterization (Livni et al., 2014; Safran and Shamir, 2015; Soudry and Carmon, 2016; Nguyen and Hein, 2017; Soudry and Hoffer, 2017; Du and Lee, 2018; Venturi et al., 2019; Soltanolkotabi et al., 2022). Previously, network embedding has been studied in various setups (Fukumizu and Amari, 2000; Fukumizu et al., 2019; Safran et al., 2020; Şimşek et al., 2021; Zhang et al., 2021), mostly focusing on how network embedding reshapes stationary points. We extend this line of work to the non-differentiable cases.

## 1.2 MAIN CONTRIBUTION

In this paper, we study the aforementioned topics for one-hidden-layer networks with ReLU-like activation functions trained by the empirical squared loss. Our contributions are as follows.
- Noticing that the non-differentiability only lies within the hyper-planes orthogonal to training inputs, and derivatives can be handled easily on both sides of those hyperplanes, we develop a rou-

tine to fully investigate the ODDs of the loss. With its help, we identify, classify, and characterize the stationary points with non-differentiability fully considered.

- We preclude saddle escape types that previous papers (Maennel et al., 2018; Boursier et al., 2022; Chistikov et al., 2024; Boursier and Flammarion, 2024; Kumar and Haupt, 2024) were not able to rule out. We show that saddle escape must involve the parameter changes of escape neurons (Definition 4.1) and apply this to describe the saddle-to-saddle dynamics.
- We study whether network embedding preserves stationarity or local minimality.

## 2 SETUP

**Network Architecture.** We study one-hidden-layer networks as illlustrated in Figure 1.

For a given input $\mathbf{x} \in \mathbb{R}^d$, the network output is:

$$\hat{\mathbf{y}}(H, W; \mathbf{x}) = \hat{\mathbf{y}}(\mathbf{P}; \mathbf{x}) = H\rho(W\mathbf{x}). \qquad (1)$$

In Equation (1), $W \in \mathbb{R}^{|I| \times d}$ and $H = (h_{ji}) \in \mathbb{R}^{|J| \times |I|}$ are respectively the input and output weight matrices, and $I$ and $J$ are the sets of indices of the hidden neurons and output neurons. The vector $\mathbf{P} \in \mathbb{R}^D$ contains all the trainable parameters $W$ and $H$, so that $D = |J| \times |I| + |I| \times d$. We denote the row of $W$ corresponding to hidden neuron $i$ by $\mathbf{w}_i \in \mathbb{R}^d$. The componentwise activation function $\rho(\cdot)$ reads:

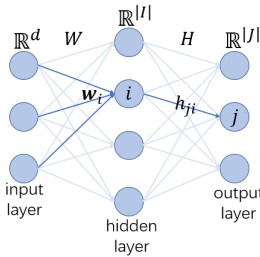

Figure 1: network architecture

$$\rho(z) = \alpha^+ \mathbb{1}_{z \geq 0} z + \alpha^- \mathbb{1}_{z < 0} z; \quad \alpha^+, \alpha^-, z \in \mathbb{R}.$$

In this paper, the *assumptions* we make are $\alpha^+ \neq \alpha^-$ and $d > 1$. Both are for brevity in exposition, rather than being indispensable for the validity of the outcomes.

**Loss Function.** The training input/targets are $\mathbf{x}_k/\mathbf{y}_k$ with $k \in K$, where $K$ denotes the set of the sample indices. The empirical squared loss function takes the form:

$$\mathcal{L}(\mathbf{P}) = \frac{1}{2} \sum_{k \in K} \|\hat{\mathbf{y}}_k(\mathbf{P}) - \mathbf{y}_k\|^2 = \frac{1}{2} \sum_{j \in J} \sum_{k \in K} (\hat{y}_{kj}(\mathbf{P}) - y_{kj})^2, \qquad (2)$$

where $\hat{\mathbf{y}}_k(\mathbf{P}) := \hat{\mathbf{y}}(\mathbf{P}; \mathbf{x}_k)$, and $\hat{y}_{kj}, y_{kj}$ are the components of $\hat{\mathbf{y}}_k$ and $\mathbf{y}_k$ at the output neuron $j$.

## 3 STATIONARY POINTS

In this section, we derive the ODDs and identify the stationary points of the loss.

### 3.1 DERIVING ONE-SIDED DIRECTIONAL DERIVATIVES

Non-differentiability does not completely prohibit the investigation into derivatives. For example, studying the left and right hand derivatives at the origin of $f(x) = |x|$ suffices to describe the function locally, despite the non-differentiability. Generalizing this methodology to higher dimensions, we can handle the derivatives of the loss function by deriving them in different directions. Note that the loss of ReLU-like networks is continuous everywhere and piecewise $C^\infty$, which enables us to study its ODDs, and the ODDs are informative enough to characterize its first-order properties.

It is not hard to see that our neural network is always differentiable with respect to the output weights $h_{ji}$. Non-differentiability only arises from input weights $\mathbf{w}_i$. More precisely, the network (thus the loss) is non-differentiable at $\mathbf{P}$ if and only if a weight $\mathbf{w}_i$ is orthogonal to some training input $\mathbf{x}_k$'s (see Figure 2 for an illustration). Nonetheless, when $\mathbf{w}_i$ moves a sufficiently small distance along a fixed direction, the sign of $\mathbf{w}_i \cdot \mathbf{x}_k$ is fixed for all $k \in K$. Namely, $\mathbf{w}_i \cdot \mathbf{x}_k$ only moves on either the positive leg, the negative leg of the activation function $\rho(\cdot)$, or stays at the kink, but it will not cross the kink during the movement. With $\mathbf{w}_i$ constrained within such a small local region, the loss is a polynomial with respect to it. This allows us to study the ODDs. Below, we specify some general directions where the ODDs can be written in closed forms.

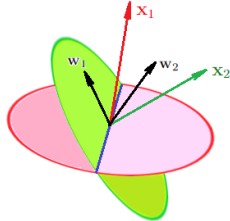

Figure 2: *A diagram demonstrating the non-differentiability with respect to the input weights.* Here, we show a case where input weights and training inputs are 3-dimensional. There are two training inputs and two input weights. $\mathbf{w}_1$ lies in a plane orthogonal to $\mathbf{x}_2$. The loss is thus locally non-differentiable since it contains the term $\rho(\mathbf{w}_1 \cdot \mathbf{x}_2)$. However, we can compute the ODD with respect to $\mathbf{w}_1$ since the loss function with $\mathbf{w}_1$ constrained on either side of the plane or on the plane is a polynomial of $\mathbf{w}_1$.

**Definition 3.1.** Consider a nonzero $\mathbf{w}_i \in \mathbb{R}^d$. The *radial direction* of $\mathbf{w}_i$ is $\mathbf{u}_i := \frac{\mathbf{w}_i}{\|\mathbf{w}_i\|}$. Moreover, any unit vector $\mathbf{v}_i$ orthogonal to $\mathbf{u}_i$ is called a *tangential direction*. For $\mathbf{w}_i = \mathbf{0}$, the *radial direction* does not exist, and a *tangential direction* is defined to be any unit vector $\mathbf{v}_i \in \mathbb{R}^d$.

*Remark* 3.2. The tangential direction is always definable given our assumption of $d > 1$.

Consider a non-zero $\mathbf{w}_i \in \mathbb{R}^d$ and let $\mathbf{w}_i' = \mathbf{w}_i + \Delta\mathbf{w}_i \neq \mathbf{w}_i$ be in the neighborhood of $\mathbf{w}_i$. The vector $\Delta\mathbf{w}_i$ admits a unique decomposition $\Delta\mathbf{w}_i = \Delta r_i \mathbf{u}_i + \Delta s_i \mathbf{v}_i$ with $\Delta s_i \geq 0$ and $\mathbf{v}_i$ being a specific tangential direction of $\mathbf{w}_i$. Fixing a direction for the derivative is thus equivalent to fixing a tangential direction. For a generic function $f$, $\frac{\partial f(\mathbf{w}_i)}{\partial r_i} := \lim_{\Delta r_i \to 0+} \frac{f(\mathbf{w}_i + \Delta r_i \mathbf{u}_i) - f(\mathbf{w}_i)}{\Delta r_i}$ and $\frac{\partial f(\mathbf{w}_i)}{\partial s_i} := \lim_{\Delta s_i \to 0+} \frac{f(\mathbf{w}_i + \Delta s_i \mathbf{v}_i) - f(\mathbf{w}_i)}{\Delta s_i}$ are the *radial derivative* and the *tangential derivative*.

For $\mathbf{w}_i = \mathbf{0}$, we can likewise define the tangential derivative for every tangential direction $\mathbf{v}_i$, and by convention, set the undefinable $\mathbf{u}_i$ and the radial derivative to zero.

*Remark* 3.3. The notation of $\frac{\partial}{\partial s_i}$ always implies a fixed tangential direction $\mathbf{v}_i$.

We can then study the ODDs with respect to the output weights and the radial/tangential directions of the input weights.

**Lemma 3.4.** *We first denote* $(\hat{y}_{kj} - y_{kj}) := e_{kj}$. *Then, we define the following quantities:*

$$\mathbf{d}_{ji} := \sum_{\substack{k: \\ \mathbf{w}_i \cdot \mathbf{x}_k > 0}} \alpha^+ e_{kj}\mathbf{x}_k + \sum_{\substack{k: \\ \mathbf{w}_i \cdot \mathbf{x}_k < 0}} \alpha^- e_{kj}\mathbf{x}_k,$$

$$\mathbf{d}_{ji}^{\mathbf{v}_i} := \lim_{\Delta s_i \searrow 0+} \left( \sum_{\substack{k: \\ (\mathbf{w}_i + \Delta s_i \mathbf{v}_i) \cdot \mathbf{x}_k > 0}} \alpha^+ e_{kj}\mathbf{x}_k + \sum_{\substack{k: \\ (\mathbf{w}_i + \Delta s_i \mathbf{v}_i) \cdot \mathbf{x}_k < 0}} \alpha^- e_{kj}\mathbf{x}_k \right). \tag{3}$$

*We have that*

$$\frac{\partial \mathcal{L}(\mathbf{P})}{\partial h_{ji}} = \mathbf{w}_i \cdot \mathbf{d}_{ji}, \quad \forall\, (j,i) \in J \times I, \tag{4}$$

$$\frac{\partial \mathcal{L}(\mathbf{P})}{\partial r_i} = \sum_{j \in J} h_{ji}\mathbf{d}_{ji} \cdot \mathbf{u}_i, \quad \forall\, i \in I, \tag{5}$$

$$\frac{\partial \mathcal{L}(\mathbf{P})}{\partial s_i} = \sum_{j \in J} h_{ji}\mathbf{d}_{ji}^{\mathbf{v}_i} \cdot \mathbf{v}_i, \quad \forall\, i \in I, \forall\, \textit{tangential direction } \mathbf{v}_i\text{'s of } \mathbf{w}_i. \tag{6}$$

The proof for the above is in Appendix A. Such ODD computation can also be adapted for neural networks with more than one hidden layer, which is explained in Appendix B.

It is easy to check that the network is linear with respect to the output weights and the norm of the input weights, therefore $\frac{\partial \mathcal{L}}{\partial h_{ji}}$ and $\frac{\partial \mathcal{L}}{\partial r_i}$ are continuous everywhere. But this is not the case for $\frac{\partial \mathcal{L}}{\partial s_i}$.

## 3.2 IDENTIFYING STATIONARY POINTS

To capture the GD-stagnating points on the loss landscape while accounting for the non-differentiability, we invoke the notion of directional stationarity to define stationary points.

**Definition 3.5.** A set of parameters $\overline{\mathbf{P}}$ is a stationary point of the loss in our setting if $\lim_{\alpha \searrow 0+} \frac{\mathcal{L}(\overline{\mathbf{P}} + \alpha \mathbf{d}) - \mathcal{L}(\overline{\mathbf{P}})}{\alpha} \geq 0$, for all $\mathbf{d} \in \mathbb{R}^D$.

Notice that the above definition of stationarity also incorporates the smooth stationary points, which are where the gradient vanishes and also exist in the loss landscape of ReLU-like networks. By

precluding first-order loss-decreasing paths in all directions, the above definition captures points near which GD infinitely decelerates or effectively comes to a halt.[1] Capturing these phenomena is the desideratum of the notion of stationary points, which Definition 3.5 intuitively satisfies. We showcase this on a scalar function in Appendix C.1. We also provide examples of non-differentiable stationary points in Appendix C.2.

Exploiting the structure of our problem, we arrive at an equivalent definition of stationary points:

**Definition 3.6.** A set of parameters $\overline{\mathbf{P}}$ is a stationary point of the loss in our setting if the following holds: (1) $\frac{\partial \mathcal{L}(\overline{\mathbf{P}})}{\partial h_{ji}} = 0, \forall (j, i) \in J \times I$; (2) $\frac{\partial \mathcal{L}(\overline{\mathbf{P}})}{\partial r_i} = 0, \forall i \in I$; (3) $\frac{\partial \mathcal{L}(\overline{\mathbf{P}})}{\partial s_i} \geq 0, \forall i \in I, \forall$ tangential direction $\mathbf{v}_i$'s of $\overline{\mathbf{w}}_i$.

For directions along which the loss is continuously differentiable, we require the derivatives to be $0$, which explains the first two conditions. Along the other directions, our definition permits upward slopes in one direction and the opposite at a stationary point, as indicated in the third condition.

The equivalence between Definition 3.5 and Definition 3.6 is intuitive. For a generic function, there is a maximal subspace in its parameter space where the function is differentiable, which we call the differentiable subspace for convenience. The complementary subspace of the differentiable subspace is where the function is non-differentiable, which we call the non-differentiable subspace.

To examine all the ODDs surrounding a point in the parameter space, as required by Definition 3.5, it suffices to check the ODDs in the differentiable subspace and the non-differentiable subspace separately. This is because an ODD toward an arbitrary direction $\mathbf{d}$ in the parameter space can be written as a linear combination of the ODD along the projection of $\mathbf{d}$ onto the differentiable subspace and the ODD along the project of $\mathbf{d}$ onto the non-differentiable subspace. The ODDs along the coordinate axes that span the differentiable subspace are sufficient to describe any ODD within the differentiable subspace, hence the first two conditions in Definition 3.6. However, the ODDs toward all the directions in the non-differentiable subspace need to be examined in the general case, hence the last condition in Definition 3.6.

Notably, Definitions 3.5 and 3.6 are tailored to capture the stationarity of GD as it admits positive slopes around stationary points. If we were to define stationarity for, say, gradient ascent, we would admit negative slopes around stationary points instead. One implication of such a design is that, while all local minima are stationary points under Definitions 3.5 and 3.6, this is not the case for non-differentiable local maxima. An interesting side note can be made about the rarity of local maxima on our loss landscape, whether they are differentiable or not. We can prove that a necessary condition for $\mathbf{P} \in \mathbb{R}^D$ to become a local maximum is $\hat{\mathbf{y}}_k(\mathbf{P}) = 0$ for all $k \in K$, as shown in Appendix D. Such a result extends the arguments regarding the non-existence of differentiable local maxima by Liu (2021); Botev et al. (2017), offering a qualitative perspective of the loss landscape.

**Definition 3.7.** A *saddle point* is a stationary point that is not a local minimum or a local maximum.

### 3.2.1 COMPARISON WITH OTHER NOTIONS OF STATIONARITY

Previous results on non-smooth non-convex stationarity mostly focus on Clarke stationary points (Davis et al., 2020b; Yun et al., 2019), which is where the Clarke subdifferential (Clarke, 1975) includes zero. But, such a notion of stationarity is not the best fit for studying GD. For example, the origins of $f(x) = |x|$ and $f(x) = -|x|$ are both Clarke stationary points, but only the former stalls GD and qualifies as a stationary point of GD. Notably, directional stationary points are Clarke stationary points with no negative slopes around (see Appendix C.3). As Clarke stationary points are known to be where stochastic subgradient descent converges (Davis et al., 2020b), our notion of stationarity offers a more refined understanding of such first-order methods. It is also noteworthy that Clarke subdifferential (Clarke, 1975) is a widely adopted tool for characterizing gradient flow of ReLU networks (Boursier et al., 2022; Chistikov et al., 2024; Kumar and Haupt, 2024). Given directional stationarity is more relevant for training dynamics, it might be more advisable to use Fréchet subdifferential, which underlies the definition of directional stationarity Li et al. (2020).[2]

---

[1]We consider GD oscillating around a certain point to be effectively equivalent to GD coming to a halt.

[2]In plain words, if the Fréchet subdifferential at a point contains $\mathbf{0}$, it is a directional stationary point. For more details, please refer to Li et al. (2020) for a review.

Another widely known stationarity criterion is to have the subgradient set to contain $\mathbf{0}$. However, such a notion does not apply to our problem since the subgradient cannot be defined for non-convex functions (see Remark C.4).

Cheridito et al. (2022) defined the stationary points of ReLU networks to be where the right-hand directional derivatives along the canonical coordinate axes are zero. However, in the presence of non-smoothness, ODDs derived along the canonical axes might not be informative enough to characterize the local structure of the function. We demonstrate this with a toy example in Appendix C.4.

## 4 MAIN RESULTS

In this section, we classify and characterize stationary points (Section 4.1) and discuss its implication for the training dynamics in the vanishing initialization limit (Section 4.2). We will also describe how network embedding reshapes the stationary points (Section 4.3).

### 4.1 PROPERTIES OF STATIONARY POINTS

**Definition 4.1.** At a stationary point, a hidden neuron $i$ is *an escape neuron* if and only if there exist $j' \in J$ and a tangential direction $\mathbf{v}_i$ such that $\frac{\partial \mathcal{L}(\overline{\mathbf{P}})}{\partial s_i} = \sum_{j \in J} \overline{h}_{ji} \mathbf{d}_{ji}^{\mathbf{v}_i} \cdot \mathbf{v}_i = 0$, and $\mathbf{d}_{j'i}^{\mathbf{v}_i} \cdot \mathbf{v}_i \neq 0$.

**Theorem 4.2.** *Let $\overline{\mathbf{P}}$ be a stationary point. If $\overline{\mathbf{P}}$ does not contain escape neurons, then $\overline{\mathbf{P}}$ is a local minimum. Furthermore, this sufficient condition is also a necessary one when $|J| = 1$.*

*Remark* 4.3. An intuition of Definition 4.1 and Theorem 4.2 for the scalar-output case is in Appendix E. Theorem 4.2 is proved by Appendix F.1 and Appendix F.2.

Proving Theorem 4.2 requires performing Taylor expansion methodically toward all directions in the parameter space. Taylor expansion requires differentiability and thus might seem, at first glance, prohibited for our problem. We solve this by aligning the coordinate system with the non-differentiable edges in the parameter space, so that the loss is always $C^{\infty}$ within each orthant. Then, we can use Taylor expansion based on the ODDs (of any orders) along the aligned coordinate axes to accurately characterize the function within the orthants, as it would be like studying a differentiable function but limiting the scope to an orthant. We provide visualization for such a method in Appendix G.

Since we can also compute the ODDs for deeper nets with an aligned coordinate system, as showcased in Appendix B, it is conceptually easy to also compose such stationary-point-classifying theorems for deeper nets. However, deeper nets entail a more cumbersome proof as there will be higher-ordered terms to be controlled. Hence, we limit our current discussion to the shallow structure, which already suffices to solve unresolved problems from previous works of training dynamics and network embedding, discussed in Sections 4.2 and 4.3, respectively.

In the following, we refer to local minima without escape neurons as *type-1 local minima*, and to the others as *type-2 local minima*, which does not exist when $|J| = 1$ as per Theorem 4.2. We tend to argue that type-2 local minima are rare and provide our reasoning in Appendix F.2.1.

We also show that the stationary local maxima and saddle points of the loss offer "second-order decrease" along the escape path, which can be formalized as below.

**Corollary 4.4.** *If $\overline{\mathbf{P}}$ is a non-minimum stationary point of $\mathcal{L}$ with $|J| = 1$, then there exist a unit vector $\ell \in \mathbb{R}^D$ and a constant $C < 0$ such that $0 > \mathcal{L}(\overline{\mathbf{P}} + \delta\ell) - \mathcal{L}(\overline{\mathbf{P}}) \sim C\delta^2$, as $\delta \to 0+$.*

Corollary 4.4 is explained in Remark F.2. It is established by characterizing the loss-decreasing path caused by the variation of escape neuron parameters. It draws similarities between the potentially non-differentiable saddle points in our setting and smooth strict saddles. It is known that saddle points in shallow linear networks are strict (Kawaguchi, 2016).[3] The above corollary thus serves to extend (Kawaguchi, 2016) to non-differentiable stationary points.

### 4.1.1 A NUMERICAL EXPERIMENT FOR THEOREM 4.2

We present a numerical experiment to illustrate Theorem 4.2, also serving as a precursor to our discussion on training dynamics in Section 4.2. We train a ReLU network that has 50 hidden neurons

---

[3](Kawaguchi, 2016) also proved that saddle points in the *true loss* of shallow ReLU networks are strict.

(a) loss curve      (b) direction of $\mathbf{w}_i$ (rad)      (c) $\|\mathbf{w}_i\|$      (d) $h_{j_0 i}$

Figure 3: *Evolution of all parameters during the training process from vanishing initialization.* **(a)** The loss curve encounters three plateaus, the last of which corresponds to a local minimum (confirmed by Theorem 4.2). We mark the end of the plateaus with dashed vertical lines. **(b)** The input weights that are not associated with dead neurons are grouped at several angles. **(c) & (d)** Grouped neurons have their amplitude increased from near-zero values, which coincides with the saddle escape. The movements of $\|\mathbf{w}_i\|$ and $|h_{j_0 i}|$ are synchronous (see Fact 4.5).

with GD using 5 training samples $(\mathbf{x}_k, y_k)$. The inputs are $\mathbf{x}_k = (x_k, 1) \in \mathbb{R}^2$, and $x_k$'s are $-1$, $-0.6$, $-0.1$, $0.3$, $0.7$. The target $y_k$'s, are $0.28$, $-0.1$, $0.03$, $0.23$, $-0.22$. All the parameters are initialized independently with law $\mathcal{N}\left(0, (5 \times 10^{-6})^2\right)$.[4] The training lasts $500k$ epochs with a step size of $0.001$. We will identify the saddle points and local minima encountered during training.

The training process is visualized in Figure 3. Figure 3a is the loss curve, where the plateaus reflect the parameters moving near stationary points. Figures 3b to 3d show the evolution of input weight orientations, input weight norms, and output weights. The input weight orientations are the counterclockwise angles between $\mathbf{w}_i$'s and the vector $(1, 0)$.

Each curve in Figures 3b to 3d corresponds to one hidden neuron. We see that most of the $\mathbf{w}_i$'s group at several angles before all the parameters gain considerable amplitudes. This has been well studied by Maennel et al. (2018); Kumar and Haupt (2024); Boursier and Flammarion (2024). We denote these hidden neurons by red and yellow in the figures and call them group 1 and group 2 neurons. Notice that the $\mathbf{w}_i$'s between $3.93$ and $5.67$ $rad$ (the grey-tinted area in Figure 3b) have $\rho(\mathbf{w}_i \cdot \mathbf{x}_k) = 0$ for all $k \in K$. This means that the gradients of all the parameters associated with these hidden neurons are zero (from Lemma 3.4). These hidden neurons are the black curves[5] in the figures and are called the *dead neurons*.

We also note that the group 1 neurons are attracted to and stuck at a direction ($\approx 2.53$ rad) that is orthogonal to one of the training inputs ($\mathbf{x}_5 = (0.7, 1)$). This implies that the GD traverses a non-differentiable region of the loss landscape throughout the training.

In our example, the parameters escape from 2 saddles, which happens at about epochs $42k$ and $152k$, reflected by the rapid loss drop in Figure 3a. The last plateau in the loss curve is the result of a local minimum. To numerically verify this, we perturb the parameters obtained after 500k epochs of training with small noises, and the resulting loss change is always non-negative (see Appendix H).

That the last plateau is a local minimum can also be confirmed by applying Theorem 4.2. In the following, we show that the stationary points that the network parameter is close to at epoch 42k and 152k have escape neurons, while the stationary point stalling the loss decrease in the end does not. For clarity, the index of the only output neuron is specified to be $j_0$.

**Fact 4.5.** *Throughout training,* $(\|\mathbf{w}_i\|^2 - h_{j_0 i}^2)$ *remains unchanged.*[6] *Thus, with vanishing (small but non-vanishing, resp.) initialization, we have* $\|\mathbf{w}_i\| = |h_{j_0 i}|$ $(\|\mathbf{w}_i\| \approx |h_{j_0 i}|$, *resp.) during training.*

Definition 4.1 indicates that the escape neurons have $h_{j_0 i} = 0$ (also explained in Appendix E). By Fact 4.5, the amplitudes of all the parameters associated with escape neurons must be small. On the other hand, dead neurons (which only exist in the ReLU case) are not escape neurons by definition.

Thus, when the network nears a saddle point, it must contain neurons that have small amplitudes and are not dead neurons. We will refer to such neurons as *small living neurons* in the following. They are the ones corresponding to the escape neurons in the nearby saddle point. In Figure 3, at

---

[4] We chose such an initialization scale to simulate the vanishing initialization since further diminishing it will not qualitatively change the patterns of the training dynamics.

[5] Note, there are black curves in Figures 3c and 3d. They stay too close to the zero line to show up.

[6] This conclusion was proved in Section 9.5 of Maennel et al. (2018).

epoch 42k (152k, resp.), neurons belonging to both group 1 and 2 (group 2, resp.) are small living neurons. However, after the network reaches the last plateau, small living neurons are depleted, meaning the stationary point it was approaching must be a local minimum. A formal statement of such observations for vanishing initialization is in Corollary 4.6.

We also conducted two other numerical experiments whose results are presented in Appendix I. One used 3-dimensional input and scalar output, and the other used 2-dimensional input and output. The former shows similar behavior as the experiment in this section, the latter reveals escape neurons that have non-zero output weights, which exist when $|J| > 1$, as predicted by Definition 4.1.

## 4.2 SADDLE ESCAPE IN THE SADDLE-TO-SADDLE DYNAMICS

Networks trained from vanishing initialization exhibit *saddle-to-saddle dynamics* (Boursier et al., 2022; Jacot et al., 2022; Pesme and Flammarion, 2024), meaning the loss experiences intermittent decreases punctuated by long plateaus. Multiple previous works have studied this dynamics (Maennel et al., 2018; Boursier et al., 2022; Chistikov et al., 2024; Boursier and Flammarion, 2024). However, previous discussions regarding saddle escape were usually premised on strong assumptions, such as orthogonality (Boursier et al., 2022) or correlatedness of training inputs (Chistikov et al., 2024). Without such assumptions, previous works could not go beyond the escape from the origin, which is trivially a saddle point (Maennel et al., 2018; Boursier and Flammarion, 2024). These simplifications/narrowing imposed a specific form on all the saddle points being discussed. Such a form was summarized by Kumar and Haupt (2024), and they further noted that there could be other types of saddle points that remained undiscussed (in their Section 5.2). However, we show that no other types of saddle points need to be considered in such dynamics and demonstrate qualitatively how saddle points are escaped from. Aligned with previous works, the discussion in this section is for scalar-output ReLU networks.

We can see that, Theorem 4.2 and Fact 4.5 only allow gradient flow from vanishing initialization to approach saddle points that have hidden neurons whose parameters are all zero,[7] which coincide exactly with saddle points previously studied by Kumar and Haupt (2024); Maennel et al. (2018); Boursier et al. (2022); Chistikov et al. (2024); Boursier and Flammarion (2024), according to Section 5.2 of Kumar and Haupt (2024). To avoid ambiguity, we present our result formally:

**Corollary 4.6.** *Following the setup and notation introduced in Section 2, let $\mathbf{P}(t) = (\mathbf{w}_i(t), h_{j_0 i}(t))_{i \in I} \in \mathbb{R}^D$ be the parameter trajectory of a one-hidden-layer scalar-output ReLU network trained with the empirical squared loss $\mathcal{L}$, where $t \geq 0$ denotes time. Suppose the trajectory satisfies $\frac{d\mathbf{P}}{dt} = -\frac{\partial \mathcal{L}}{\partial \mathbf{P}}$, in which we specify $\frac{\partial \operatorname{ReLU}(x)}{\partial x} = \mathbb{1}_{\{x > 0\}}$ in the chain rule.[8] Let $\mathbf{A} \in \mathbb{R}^D$ be an arbitrary vector. We initialize the network with $\mathbf{P}(0) := \sigma \mathbf{A}$. Under these conditions, if $\lim_{\sigma \searrow 0^+} \left( \inf_t \|\mathbf{P}(t) - \overline{\mathbf{P}}\| \right) = 0$, where $\overline{\mathbf{P}} = (\overline{\mathbf{w}}_i, \overline{h}_{j_0 i})_{i \in I}$ is a non-minimum stationary point, we must have $\overline{\mathbf{w}}_i = \mathbf{0}$ and $\overline{h}_{j_0 i} = 0$ for some $i \in I$.*

We next discuss how the gradient flow escapes from the saddle points. In the vicinity of the saddle points, the network has hidden neurons with small parameters, which can either be dead neurons or small living neurons. It is easy to see that locally perturbing dead neurons does not change the loss (Fukumizu et al., 2019) and they cannot move since they have zero gradients. On the other hand, a direct consequence of Theorem 4.2 (proved in Appendix F.1 and explained in Remark F.1) states:

**Fact 4.7** (A partial restatement of Theorem 4.2). *A strictly loss-decreasing path from a stationary point of the network must involve parameter variations of escape neurons.*

As a result, saddle escape must change the parameters of small living neurons, and such parameter change of small living neurons exploits the loss-decreasing path offered by the escape neurons in the nearby saddle point (Fact 4.7). Moreover, when the training stalls near a saddle point, the small living neurons are always attracted to where their parameter amplitudes will be increased (Maennel et al., 2018). Combining all these, we know that escaping from saddles always entails amplitude increase of small living neurons.

---

[7]It is worth noting that there could be saddle points with $h_{j_0 i} = 0$ but $\mathbf{w}_i \neq 0$ on the entire landscape. However, Fact 4.5 states that such saddle points are not visited if we start from vanishing initialization.

[8]This subgradient-style differential cannot help us characterize the stationarity of non-convex functions (as discussed in Appendix C.2.2), but suffices to identify a unique gradient flow without involving additional concepts. Notably, this is also how PyTorch implements the derivative of ReLU.

Other forms of saddle escape cannot take place. This contrasted with other different but similar setups. For example, in Fukumizu et al. (2019); Safran et al. (2020); Pesme and Flammarion (2024),[9] the existence of several input weights with the same direction suffices to make a strict saddle, meaning saddle escape can take place without involving small living neurons.

In the rest of this section, we will combine our results with previous works to give a full understanding of the entire saddle-to-saddle dynamics, which is summarized in Figure 4.

The initial phase of training will have the $\mathbf{w}_i$'s that are not associated with dead neurons to group at finitely many attracting directions when all the parameters have negligible amplitudes. Such a phase is termed the *alignment phase* (Maennel et al., 2018; Luo et al., 2021; Boursier and Flammarion, 2024; Kumar and Haupt, 2024) and can be observed in Figure 3b. The neurons whose input weights are grouped closely will be called an *effective neuron* in the following, as they correspond to one kink position in the learned function.

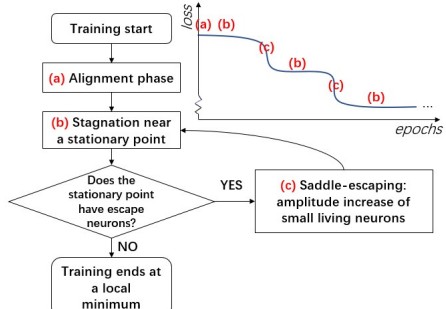

After the initial alignment phase, the grouped neurons will have their amplitude increased from near-zero values. Notice that these neurons are small living neurons before the amplitude increase. Those that contribute to one effective neuron would have their amplitudes increased together (Maennel et al., 2018; Boursier et al., 2022; Chistikov et al., 2024),[10] which exploits the loss decreasing path offered by escape neurons in the saddle point nearby (Fact 4.7). To describe such a process, Boursier et al. (2022) and Chistikov et al. (2024) rigorously characterized the gradient flow for simplified cases, revealing trajectories consistent with Figure 4. The training process terminates in a local minimum after all the small living neurons are depleted.

Moving forward, it will be meaningful to investigate the exact dynamics between saddles in the general setup. Boursier et al. (2022); Chistikov et al. (2024) already tackled this for the training process with orthogonal and correlated training inputs. These simplifications averted the simultaneous norm change of different groups of neurons during saddle escape,

Figure 4: *A flow chart for the training process.* We preclude other schemes of saddle escape, for example, by splitting aligned neurons, which is possible in Fukumizu et al. (2019); Safran et al. (2020); Pesme and Flammarion (2024). Also, note that besides the indispensable amplitude increase of small living neurons, saddle escape might also be accompanied by amplitude and orientation changes of other neurons.

which often takes place in general. An example is the training process after epoch 152k in Figure 3.

Notably, as the loss landscape is continuous, the insight we draw for vanishing initialization might be extrapolated to small but non-vanishing initialization. Concretely, in the latter case, we can still observe that the accelerations of loss decrease also roughly coincide with the amplitude increase of small living neurons. This means that the acceleration of loss decrease in this regime also exploits the loss-decreasing path given by the escape neurons in the nearby saddle points, similar to the vanishing initialization regime. We elucidate such an observation in Appendix J.

### 4.3 How Network Embedding Reshapes Stationary Points

Network embedding is the process of instantiating a network using a wider network without changing the network function, or at least, the output of the network evaluated at all the training input $\mathbf{x}_k$'s. In this section, we demonstrate how network embedding reshapes stationary points, which offers a perspective on the merit of over-parameterization. In particular, we answer the following question: Does network embedding preserve stationarity or local minimality? Such questions were partially addressed by (Fukumizu et al., 2019) for ReLU networks. With a more complete characterization of stationary points and local minima in Sections 3.2 and 4.1, we are able to extend their results.

---

[9]See Appendix P for the setting of Fukumizu et al. (2019). Safran et al. (2020) studied a teacher-student setup with true loss. Pesme and Flammarion (2024) studied diagonal linear networks.

[10]The amplitude increase of one group of small living neurons creates one kink in the learned function (Figure 16).

Following the naming in (Fukumizu et al., 2019), there are three network embedding strategies for ReLU(-like) networks: unit replication, inactive units, and inactive propagation. We will showcase the usability of our theory for unit replication, which is the most technically involved, in the main text. The proof for unit replication and the full discussion for the other two embedding strategies are presented in Appendices M to O.

The process of unit replication is illustrated in Figure 5: given a set of parameters $\mathbf{P}$, replace the hidden neuron $i_0 \in I$ associated with $\mathbf{w}_{i_0}$ and $h_{ji_0}$, with a new set of hidden neurons $\{i_0^l; l \in L\}$ associated with weights $\mathbf{w}_{i_0^l} = \beta_l \mathbf{w}_{i_0}$ and $h_{ji_0^l} = \gamma_l h_{ji_0}$ for all $j \in J$. To preserve the network function, we specify: $\beta_l > 0$, $\sum_{l \in L} \beta_l \gamma_l = 1$, for all $l \in L$. We can prove the following for unit replication.

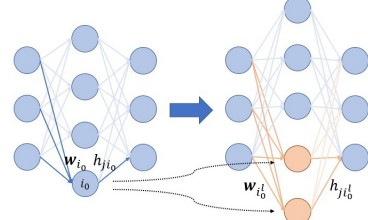

Figure 5: unit replication

**Proposition 4.8.** *In our setting, unit replication performed on neuron $i_0$ preserves the stationarity of a stationary point if and only if at least one of the following conditions is satisfied: (1) All tangential derivatives of $\mathbf{w}_{i_0}$ are $0$. (2) $\gamma_l \geq 0, \forall\, l \in L$.*
*Moreover, unit replication performed on neuron $i_0$ preserves the type-1 local minimality of a type-1 local minimum if and only if at least one of the following holds: (1) All tangential derivatives of $\mathbf{w}_{i_0}$ are $0$. (2) $\gamma_l > 0, \forall\, l \in L$.*

*Remark* 4.9. (Fukumizu et al., 2019) was not able to discuss stationarity preservation for ReLU networks as the non-differentiability hindered the invocation of stationarity conditions for differentiable functions. This problem has been hurdled completely in the current paper. Moreover, (Fukumizu et al., 2019) was only able to discuss local minimality preservation for a highly restricted type of unit replication. Our proposition above nevertheless holds for all unit replication types in general, though the scope is limited to type-1 local minimality preservation.

*Remark* 4.10. At first sight, the statement about the preservation of type-1 local minima in the above lemma contradicts Theorem 10 of (Fukumizu et al., 2019), which suggests that embedding a local minimum should result in a strict saddle. Nonetheless, the assumptions of that theorem does not apply to our setting. For more details, please refer to Appendix P.

It is noteworthy that unit replication always preserves stationarity for differentiable networks (Zhang et al., 2021), which no longer holds for the non-differentiable ReLU-like networks.

**A closing remark.** Our discussion here extends multiple previous results regarding network embedding that did not consider the non-differentiable cases. Particularly, we refine the picture of the embedding principle of neural networks (Zhang et al., 2021), promote a better understanding of overparameterization (Şimşek et al., 2021; Fukumizu et al., 2019; Zhang et al., 2021), and provide theoretical guarantee for training schemes that involves the operations of network embedding (Wu et al., 2019; Wang et al., 2024). A more detailed discussion on these topics are deferred to Appendix Q.

## 5 SUMMARY AND DISCUSSION

In this paper, we identify, classify, and characterize the stationary points for one-hidden-layer neural networks with ReLU-like activation functions. We also study the saddle escape process in the training dynamics with vanishing initialization and the effect of network embedding on stationary points. These can lead to several future directions. First, our approach of studying ODDs and aligning coordinate axes for Taylor expansion can be extrapolated to other architectures, including convolutional networks, residue networks, transformers, etc. Second, with a better geometric understanding of the potentially non-differentiable stationary points (such as Corollary 4.4), we are at a better place to characterize the performance of gradient-based optimization algorithms on such landscapes, which are neither smooth nor convex. Third, with the stationary points identified, we can study the basins of attractions of gradient flow. Lastly, the findings of this paper might help formalize the low-rank bias formed in generic saddle-to-saddle dynamics.

ACKNOWLEDGEMENTS

We thank Prof. Martin Jaggi from the Machine Learning and Optimization Laboratory at EPFL and Prof. Clément Hongler from the Chair of Statistical Field Theory at EPFL for their valuable suggestions and feedback. We also appreciate the insightful input from the reviewers, area chairs, and program chairs.

REPRODUCIBILITY STATEMENT

The code for all the numerical experiments can be found at https://github.com/ZhengqingUUU/relu_like_NN_loss_landscape.

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

# Appendix

## TABLE OF CONTENTS

## A    PROOF OF LEMMA 3.4

### A.1   DERIVATION OF EQUATION (4)

We compute

$$\frac{\partial \hat{y}_{kj'}(\mathbf{P})}{\partial h_{ji}} = \frac{\partial}{\partial h_{ji}} \sum_{i' \in I} h_{j'i'} \rho(\mathbf{w}_{i'} \cdot \mathbf{x}_k) = \mathbb{1}_{\{j=j'\}} \rho(\mathbf{w}_i \cdot \mathbf{x}_k).$$

Thus, we have

$$\frac{\partial}{\partial h_{ji}} \mathcal{L}(\mathbf{P}) = \frac{\partial}{\partial h_{ji}} \Big( \frac{1}{2} \sum_{j' \in J} \sum_{k \in K} (\hat{y}_{kj'} - y_{kj'})^2 \Big) = \sum_{j' \in J} \sum_{k \in K} e_{kj'} \frac{\partial \hat{y}_{kj'}}{\partial h_{ji}} = \sum_{k \in K} e_{kj} \rho(\mathbf{w}_i \cdot \mathbf{x}_k) = \mathbf{w}_i \cdot \mathbf{d}_{ji},$$

### A.2   DERIVATION OF EQUATION (5)

For all $x, y \in \mathbb{R}$, define

$$\widetilde{\rho}_y(x) := \begin{cases} \alpha_+ x & \text{if } y > 0, \\ \rho(x) & \text{if } y = 0, \\ \alpha_- x & \text{if } y < 0. \end{cases}$$

The map $\widetilde{\rho}.(\cdot)$ will be convenient to compute the following derivatives.

Let $i$ be such that $\|\mathbf{w}_i\| \neq 0$, let $\mathbf{u}_i = \frac{\mathbf{w}_i}{\|\mathbf{w}_i\|}$. Recall that the derivatives $\frac{\partial}{\partial r_i}, \frac{\partial}{\partial s_i}$ are defined below Definition 3.1. Below, we write $\mathbf{w}_i = r_i \mathbf{u}_i$ and take the derivative at $r_i = \|\mathbf{w}_i\|$. Note that $\rho(r_i \mathbf{w}_i \cdot \mathbf{x}_k) = r_i \rho(\mathbf{w}_i \cdot \mathbf{x}_k)$, that is, along the direction $\mathbf{u}_i$, the activation $\rho$ is linear. We thus have that

$$\frac{\partial \hat{y}_{kj}(\mathbf{P})}{\partial r_i} = h_{ji} \frac{\partial \rho(r_i \mathbf{u}_i \cdot \mathbf{x}_k)}{\partial r_i} = h_{ji} \rho(\mathbf{u}_i \cdot \mathbf{x}_k).$$

We then get

$$\begin{aligned} \frac{\partial \mathcal{L}(\mathbf{P})}{\partial r_i} &= \sum_{j \in J} \sum_{k \in K} e_{kj} \frac{\partial \hat{y}_{kj}}{\partial r_i} = \sum_{j \in J} \sum_{k \in K} e_{kj} h_{ji} \rho(\mathbf{u}_i \cdot \mathbf{x}_k) \\ &= \sum_{j \in J} \sum_{k \in K} e_{kj} h_{ji} \mathbf{u}_i \cdot \widetilde{\rho}_{\mathbf{u}_i \cdot \mathbf{x}_k}(\mathbf{x}_k) \\ &= \sum_{j \in J} h_{ji} \mathbf{u}_i \cdot \left( \sum_{\substack{k: \\ \mathbf{w}_i \cdot \mathbf{x}_k > 0}} \alpha^+ e_{kj} \mathbf{x}_k + \sum_{\substack{k: \\ \mathbf{w}_i \cdot \mathbf{x}_k < 0}} \alpha^- e_{kj} \mathbf{x}_k \right) \end{aligned}$$

$$= \sum_{j \in J} h_{ji} \mathbf{u}_i \cdot \mathbf{d}_{ji},$$

where $\mathbf{d}_{ji}$ was defined in Lemma 3.4.

## A.3 Derivation of Equation (6)

Recall that when taking the derivative $\frac{\partial}{\partial s_i}$, a specific direction $\mathbf{v}_i$ is chosen. One can check that

$$\frac{\partial \rho(\mathbf{w}_i \cdot \mathbf{x}_k)}{\partial s_i} = \mathbb{1}_{\{\mathbf{w}_i \cdot \mathbf{x}_k \neq 0\}} \widetilde{\rho}_{\mathbf{w}_i \cdot \mathbf{x}_k}(\mathbf{v}_i \cdot \mathbf{x}_k) + \mathbb{1}_{\{\mathbf{w}_i \cdot \mathbf{x}_k = 0\}} \rho(\mathbf{v}_i \cdot \mathbf{x}_k)$$

$$= \widetilde{\rho}_{\mathbf{w}_i \cdot \mathbf{x}_k}(\mathbf{v}_i \cdot \mathbf{x}_k)$$

This gives us

$$\frac{\partial \hat{y}_{kj}(\mathbf{P})}{\partial s_i} = h_{ji} \frac{\partial \rho(\mathbf{w}_i \cdot \mathbf{x}_k)}{\partial s_i} = h_{ji} \widetilde{\rho}_{\mathbf{w}_i \cdot \mathbf{x}_k}(\mathbf{v}_i \cdot \mathbf{x}_k)$$

We thus have that

$$\frac{\partial \mathcal{L}(\mathbf{P})}{\partial s_i} = \sum_{j \in J} \sum_{k \in K} e_{kj} \frac{\partial \hat{y}_{kj}(\mathbf{P})}{\partial s_i} = \sum_{j \in J} \sum_{k \in K} e_{kj} \widetilde{\rho}_{\mathbf{w}_i \cdot \mathbf{x}_k}(\mathbf{v}_i \cdot \mathbf{x}_k)$$

$$= \sum_{j \in J} \sum_{k \in K} e_{kj} \mathbf{v}_i \cdot \left( \mathbb{1}_{\{\mathbf{w}_i \cdot \mathbf{x}_k \neq 0\}} \widetilde{\rho}_{\mathbf{w}_i \cdot \mathbf{x}_k}(\mathbf{x}_k) + \mathbb{1}_{\{\mathbf{w}_i \cdot \mathbf{x}_k = 0\}} \widetilde{\rho}_{\mathbf{v}_i \cdot \mathbf{x}_k}(\mathbf{x}_k) \right)$$

$$= \sum_{\substack{k: \\ (\mathbf{w}_i + \Delta s_i \mathbf{v}_i) \cdot \mathbf{x}_k > 0}} \alpha^+ e_{kj} \mathbf{x}_k + \sum_{\substack{k: \\ (\mathbf{w}_i + \Delta s_i \mathbf{v}_i) \cdot \mathbf{x}_k < 0}} \alpha^- e_{kj} \mathbf{x}_k$$

$$= \sum_{j \in J} h_{ji} \mathbf{v}_i \cdot \mathbf{d}_{ji}^{\mathbf{v}_i},$$

where $\Delta s_i > 0$ is sufficiently small and where $\mathbf{d}_{ji}^{\mathbf{v}_i}$ was defined in Lemma 3.4.

## B Extending Lemma Lemma 3.4 to Networks with Multiple Hidden Layers

In this section, we demonstrate how we can compute the radial and tangential derivatives with respect to weights that are in ReLU-like networks with multiple hidden layers.

The notation of this section inherits from the one-hidden-layer case for the most part. Nonetheless, we augment the notation for the hidden layer weights with an index to number the layers.

The training inputs are $\mathbf{x}_k$'s, and we also denote them by $\mathbf{x}_k^{(0)} \in \mathbb{R}^{|I_0|}$ (for the sake of notation simplicity). The training targets are $\mathbf{y}_k \in \mathbb{R}^{|J|}$. The input weight matrix in hidden layer $l \in \{1, \cdots, L\}$ is $W^{(l)} \in \mathbb{R}^{|I_{l-1}| \times |I_l|}$. $I_l$ is the set of hidden neurons in layer $l$. The hidden neuron weights corresponding to one row of $W^{(l)}$ is $\mathbf{w}_i^{(l)}$, where $i \in I_l$. The output weight matrix is $H \in \mathbb{R}^{|J| \times |I_L|}$, each row of which is $\mathbf{h}_j$ with $j \in J$.

The neural network is defined recursively with the following:

$$\eta_k^{(l)} = W^{(l)} \mathbf{x}_k^{(l-1)}, \quad \mathbf{x}_k^{(l)} = \rho(\eta_k^{(l)}), \quad \forall l \in \{1, 2, \cdots, L\}; \quad \hat{\mathbf{y}}_k = H \mathbf{x}_k^{(L)}.$$

## B.1 Output Weight Derivatives

We first compute the network outputs' derivatives with respect to the output weights.

$$\frac{\partial \hat{y}_{kj'}}{\partial \mathbf{h}_j} = \mathbb{1}_{j=j'} \mathbf{x}_k^{(L)}$$

Then we have:

$$\frac{\partial \mathcal{L}}{\partial \mathbf{h}_j} = \sum_{k \in K} \sum_{j' \in J} e_{kj'} \frac{\partial y_{kj'}}{\partial \mathbf{h}_j} = \sum_{k \in K} e_{kj} \mathbf{x}_k^{(L)}$$

## B.2 Hidden Neuron Weight Derivatives

For a specific layer $l^0 \in \{1, 2, \cdots, L\}$, and a specific direction of $\Delta \mathbf{w}_i^{(l^0)} = \Delta r_i^{(l^0)} \mathbf{u}_i^{(l^0)} + \Delta s_i^{(l^0)} \mathbf{v}_i^{(l^0)}$, where $\mathbf{u}_i^{(l^0)}$ and $\mathbf{v}_i^{(l^0)}$ are the radial direction and a tangential direction of $\mathbf{w}_i^{(l^0)}$, and $i \in I_{l^0}$, we compute the radial derivative and tangential derivative.

### Radial Derivative

$$\frac{\partial \hat{y}_{kj}}{\partial r_i^{(l^0)}} = \frac{\partial \hat{y}_{kj}}{\partial x_{ki}^{(l^0)}} \frac{\partial x_{ki}^{(l^0)}}{\partial r_i^{(l^0)}} = \frac{\partial \hat{y}_{kj}}{\partial x_{ki}^{(l^0)}} \rho(\mathbf{u}_i^{(l^0)} \cdot \mathbf{x}_k^{(l^0-1)})$$

Thus:

$$\begin{aligned}
\frac{\partial \mathcal{L}}{\partial r_i^{(l^0)}} &= \sum_{j \in J} \sum_{k \in K} e_{kj} \frac{\partial \hat{y}_{kj}}{\partial r_i^{(l^0)}} \\
&= \sum_{j \in J} \sum_{k \in K} e_{kj} \frac{\partial \hat{y}_{kj}}{\partial x_{ki}^{(l^0)}} \rho(\mathbf{u}_i^{(l^0)} \cdot \mathbf{x}_k^{(l^0-1)}),
\end{aligned}$$

which is reminiscent of the one-hidden-layer case.

### Tangential Derivative

Again, we start by computing the output's derivative.

$$\frac{\partial \hat{y}_{kj}}{\partial s_i^{(l^0)}} = \frac{\partial \hat{y}_{kj}}{\partial x_{ki}^{(l^0)}} \frac{\partial x_{ki}^{(l^0)}}{\partial s_i^{(l^0)}} = \frac{\partial \hat{y}_{kj}}{\partial x_{ki}^{(l^0)}} \tilde{\rho}_{\mathbf{w}_i^{(l^0)} \cdot \mathbf{x}_k^{(l^0-1)}} \left( \mathbf{v}_i^{(l^0)} \cdot \mathbf{x}_k^{(l^0-1)} \right)$$

Thus:

$$\begin{aligned}
\frac{\partial \mathcal{L}}{\partial s_i^{(l^0)}} &= \sum_{j \in J} \sum_{k \in K} e_{kj} \frac{\partial \hat{y}_{kj}}{\partial s_i^{(l^0)}} \\
&= \sum_{j \in J} \sum_{k \in K} e_{kj} \frac{\partial \hat{y}_{kj}}{\partial x_{ki}^{(l^0)}} \tilde{\rho}_{\mathbf{w}_i^{(l^0)} \cdot \mathbf{x}_k^{(l^0-1)}} \left( \mathbf{v}_i^{(l^0)} \cdot \mathbf{x}_k^{(l^0-1)} \right) \\
&= \sum_{j \in J} \sum_{\substack{k: \\ (\mathbf{w}_i^{(l^0)} + \Delta s_i \mathbf{v}_i) \cdot \mathbf{x}_k > 0}} \alpha^+ e_{kj} \frac{\partial \hat{y}_{kj}}{\partial x_{ki}^{(l^0)}} \mathbf{v}_i^{(l^0)} \cdot \mathbf{x}_k^{(l^0-1)} \\
&\quad + \sum_{j \in J} \sum_{\substack{k: \\ (\mathbf{w}_i^{(l^0)} + \Delta s_i \mathbf{v}_i) \cdot \mathbf{x}_k < 0}} \alpha^- e_{kj} \frac{\partial \hat{y}_{kj}}{\partial x_{ki}^{(l^0)}} \mathbf{v}_i^{(l^0)} \cdot \mathbf{x}_k^{(l^0-1)},
\end{aligned}$$

where $\Delta s_i > 0$ is arbitrarily small. Again, this is reminiscent of the one-hidden-layer case.

### Computing $\frac{\partial \hat{y}_{kj}}{\partial x_{ki}^{(l^0)}}$

$\frac{\partial \hat{y}_{kj}}{\partial x_{ki}^{(l^0)}}$ shows up in the formula or radial directional derivative and tangential directional derivative. In this part, we compute the vector containing this value, $\frac{\partial \hat{y}_{kj}}{\partial \mathbf{x}_k^{(l^0)}}$. This derivative can be derived from the routine of back-propagation, adapted to accommodate the non-differentiability of the activation function.

We first compute:

$$\frac{\partial \hat{\mathbf{y}}_k}{\partial \mathbf{x}_k^{(L)}} = H^\intercal$$

Then, for $l \in \{l^0, l^0 + 1, \cdots, L - 1\}$, we have:

$$
\begin{aligned}
\frac{\partial \hat{\mathbf{y}}_k}{\partial \mathbf{x}_k^{(l)}} &= \frac{\partial \boldsymbol{\eta}_k^{(l+1)}}{\partial \mathbf{x}_k^{(l)}} \frac{\partial \mathbf{x}_k^{(l+1)}}{\partial \boldsymbol{\eta}_k^{(l+1)}} \frac{\partial \hat{\mathbf{y}}_k}{\partial \mathbf{x}_k^{(l+1)}} \\
&= \left( W_l^{(l)} \right)^{\mathsf{T}} \operatorname{diag}(\boldsymbol{\alpha}_k^{(l)}) \frac{\partial \hat{\mathbf{y}}_k}{\partial \mathbf{x}_k^{(l+1)}},
\end{aligned}
$$

where the vector of $\boldsymbol{\alpha}_k^{(l)}$ may be componentwise defined as:

$$
\alpha_{ki}^{(l)} = \begin{cases} \alpha^+ & \text{if } \eta_{ki}^{(l)} + \Delta\eta_{ki}^{(l)}(\Delta\mathbf{w}_i^{(l^0)}) > 0 \\ 0 & \text{if } \eta_{ki}^{(l)} + \Delta\eta_{ki}^{(l)}(\Delta\mathbf{w}_i^{(l^0)}) = 0 \\ \alpha^- & \text{if } \eta_{ki}^{(l)} + \Delta\eta_{ki}^{(l)}(\Delta\mathbf{w}_i^{(l^0)}) < 0 \end{cases}
$$

Here, $\Delta\eta_{ki}^{(l)}(\Delta\mathbf{w}_i^{(l^0)})$ is the change of $\eta_{ki}^{(l)}$ when an arbitrarily small perturbation of $\Delta\mathbf{w}_i^{(l^0)}$ is applied to $\mathbf{w}_i^{(l^0)}$. When computing radial derivative, such perturbation can be taken as $\Delta\mathbf{w}_i^{(l^0)} = \Delta r_i^{(l^0)} \mathbf{u}_i^{(l^0)}$. When computing the tangential derivative, such perturbation can be taken as $\Delta\mathbf{w}_i^{(l^0)} = \Delta s_i^{(l^0)} \mathbf{v}_i^{(l^0)}$.

## C  DISCUSSIONS ON STATIONARITY NOTIONS

### C.1  DIRECTIONAL STATIONARITY CAPTURES THE STAGNATING BEHAVIOR OF GD

In this section, we show that the absence of first-order negative slopes at one point on a scalar function prevents GD from stagnating with high probability.

**Proposition C.1.** *Consider a random function $f(x) = \sum_{n=1}^N \alpha_n x^n \mathbb{1}_{\{x<0\}} + \sum_{n=1}^N \beta_n x^n \mathbb{1}_{\{x \geq 0\}}$, where $x \in \mathbb{R}$, $N \in \mathbb{N}$, and the random coefficients $\{\boldsymbol{\alpha}, \boldsymbol{\beta}\} \triangleq \{\alpha_1, \cdots \alpha_N, \beta_1, \cdots, \beta_N\}$ are drawn independently from a distribution that is absolutely continuous with respect to Lebesgue measure. Denote left-hand and right-hand derivatives by $f_-'(\cdot)$ and $f_+'(\cdot)$. We study the GD process $x_{t+1} = x_t - \eta f_+'(x_t)$, where $\eta > 0$ is the step size. Suppose that $f_+'(0) < 0$ or $f_-'(0) < 0$. Then, with probability 1, the following holds. There exists an interval containing the origin $\chi(\boldsymbol{\alpha}, \boldsymbol{\beta}) \triangleq [a(\boldsymbol{\alpha}, \boldsymbol{\beta}), b(\boldsymbol{\alpha}, \boldsymbol{\beta})]$, where $a(\boldsymbol{\alpha}, \boldsymbol{\beta}) < 0$ and $b(\boldsymbol{\alpha}, \boldsymbol{\beta}) > 0$, and the time for escaping from this interval can be upper bounded by $\infty > \tilde{t}(\eta, \boldsymbol{\alpha}, \boldsymbol{\beta}) > 0$, which means if $x_t \in \chi$, then there exist $0 < t' \leq \tilde{t}$ such that $x_{t+t'} \notin \chi$.*

*Remark* C.2. The above proposition studies a random function to simulate that the loss landscape of neuron networks is usually also (a realization) of a random function, with randomness coming from the dataset. The above proposition shows that if the left-hand or right-hand derivative at the origin is negative, then the escape time from the origin $\tilde{t}$ only concerns the realization of the function (determined by $\boldsymbol{\alpha}, \boldsymbol{\beta}$) and the learning rate $\eta$, without involving how close GD gets to the origin. By contrast, saddle points, no matter whether they are differentiable or not, trap GD or gradient flow for a longer time if the trajectory of GD or gradient flow reaches closer to them (Maennel et al., 2018; Boursier et al., 2022; Chistikov et al., 2024). More concretely, if the origin $x = 0$ is a saddle point, then the upper bound for the escape time should be $\tilde{t}(\eta, \boldsymbol{\alpha}, \boldsymbol{\beta}, \hat{d})$, where $\hat{d} \triangleq \inf_{t' \geq 0} |x_{t+t'}|$ is the shortest distance from the trajectory following $x_t$ to the origin 0, and it must be part of the upper bound $\tilde{t}$. For example, suppose $x = 0$ is a conventional smooth saddle point, then, if $\hat{d} \searrow 0$, we have $\tilde{t} \nearrow \infty$; and if $\hat{d} = 0$, we have $\tilde{t} = \infty$ (GD is permanently stuck in this case).

*Proof.* We prove the proposition in different situations.

**Situation 1:**  $f_+'(0), f_-'(0) < 0$.
Since $f_+'(0) < 0$, based on the right continuity of the derivative, we have that there exists $b_1(\boldsymbol{\alpha}, \boldsymbol{\beta}) > 0$ such that $f_+'(x) < 0$, if $x \in [0, b_1]$. Define $m_1^+ := \sup_{x \in [0, b_1]} f_+'(x) < 0$. Then, we

have, if $x_t \in [0, b_1]$ and $t'_1$ satisfies that $x_{t+t'_1-1} \in [0, b_1]$, $x_{t+t'_1} - x_t \geq -\eta m_1^+ t'_1 > 0$. As a result, there exists a time step that GD would step beyond $b_1$. Namely, there exists a step count $t'_1$ such that $x_{t+t'_1} > b_1$, and $t'_1$ must be upper-bounded by: $t'_1 \leq \lceil \frac{b_1(\boldsymbol{\alpha},\boldsymbol{\beta})}{-\eta m_1^+} \rceil + 1 \coloneqq \hat{t}'_1$.

Similarly, since $f'_-(0) < 0$, we can have an interval $[a_1(\boldsymbol{\alpha},\boldsymbol{\beta}), 0)$ with $a_1(\boldsymbol{\alpha},\boldsymbol{\beta}) < 0$ such that $f'_-(x) < 0$, if $x \in [a_1, 0]$. Define $m_1^- \coloneqq \sup_{x \in [a_1, 0)} f'_-(x) < 0$. We have, if $x_t \in [a_1, 0)$ and $t''_1$ satisfies that $x_{t+t''_1-1} \in [a_1, 0)$, $x_{t+t''_1} - x_t \leq \eta m_1^- t''_1 < 0$, where we applied $f'_-(x) = -f'_+(x)$, if $x \in [a_1, 0)$, to the GD update rule. As a result, there exists a time step that GD would step beyond $a_1$. Namely, there exists a step count $t''_1$ such that $x_{t+t''_1} < a_1$, and $t''_1$ must be upper-bounded by: $t''_1 \leq \lceil \frac{a_1(\boldsymbol{\alpha},\boldsymbol{\beta})}{-\eta m_1^-} \rceil \coloneqq \hat{t}''_1$.

Up till now, we have proved that, *for situation 1, there exists an interval* $\chi(\boldsymbol{\alpha},\boldsymbol{\beta}) = [a_1(\boldsymbol{\alpha},\boldsymbol{\beta}), b_1(\boldsymbol{\alpha},\boldsymbol{\beta})]$ *such that, if* $x_t \in \chi$, *there exits* $\tilde{t}_1(\boldsymbol{\alpha},\boldsymbol{\beta},\eta) \coloneqq \max(t'_1, t''_1)$ *such that* $x_{t+t'} \notin \chi$ *for some* $0 < t' \leq \tilde{t}_1(\boldsymbol{\alpha},\boldsymbol{\beta},\eta)$. **Notice that this case is a Clarke stationary point (Wang et al., 2022; Yun et al., 2019). Nonetheless, we prove here that such a point does not slow down GD.**

**Situation 2:** $f'_+(0) < 0, f'_-(0) > 0$.
Since $f'_+(0) < 0$, similar to situation 1, we can have the following. There exists an interval $[0, b_2(\boldsymbol{\alpha},\boldsymbol{\beta})]$ where $b_2(\boldsymbol{\alpha},\boldsymbol{\beta}) > 0$ such that, if $x_t \in [0, b_2(\boldsymbol{\alpha},\boldsymbol{\beta})]$, then there exists a step count $t'_2$ such that $x_{t+t'_2} > b_2$, and $t'_2$ must be upper-bounded by: $t'_2 \leq \lceil \frac{b_2(\boldsymbol{\alpha},\boldsymbol{\beta})}{-\eta m_2^+} \rceil + 1 \coloneqq \hat{t}'_2$, where $m_2^+$ is defined with $m_2^+ \coloneqq \sup_{x \in [0, b_2]} f'_+(x) < 0$.

Moreover, since $f'_-(0) > 0$, we can have an interval $[a_2(\boldsymbol{\alpha},\boldsymbol{\beta}), 0)$ with $a_2(\boldsymbol{\alpha},\boldsymbol{\beta}) > 0$ and $f'_-(x) < 0$, if $x \in [a_2, 0]$. Define $m_2^- \coloneqq \inf_{x \in [a_2, 0)} f'_-(x) > 0$. We have, if $x_t \in [a_2, 0)$ and $t''_2$ satisfies that $x_{t+t''_2-1} \in [a_2, 0)$, $x_{t+t''_2} - x_t \geq \eta m_2^- t''_2 > 0$, where we applied $f'_-(x) = -f'_+(x)$, if $x \in [a_2, 0)$, to the GD update rule. As a result, there exists a time step that GD would step beyond 0. Namely, there exists a step count $t''_2$ such that $x_{t+t''_2} > 0$, and $t''_2$ must be upper-bounded by: $t''_2 \leq \lceil \frac{a_2(\boldsymbol{\alpha},\boldsymbol{\beta})}{\eta m_2^-} \rceil \coloneqq \hat{t}''_2$.

Up till now, we have proved that, *for situation 2, there exists an interval* $\chi(\boldsymbol{\alpha},\boldsymbol{\beta}) = [a_2(\boldsymbol{\alpha},\boldsymbol{\beta}), b_2(\boldsymbol{\alpha},\boldsymbol{\beta})]$ *such that, if* $x_t \in \chi$, *there exits* $\tilde{t}_2(\boldsymbol{\alpha},\boldsymbol{\beta},\eta) \coloneqq \hat{t}'_1 + \hat{t}''_1$ *such that* $x_{t+t'} \notin \chi$ *for some* $0 < t' \leq \tilde{t}_1(\boldsymbol{\alpha},\boldsymbol{\beta},\eta)$.

**Situation 3** : $f'_+(0) > 0, f'_-(0) < 0$. We can prove the proposition holds for this situation similar to what we have done for situation 2.

Other situations account for measure 0 and do not concern the validity of the proposition. $\qquad\square$

*Remark* C.3. Note that there could indeed be cases with probability 0 in which directional stationarity fails to capture the GD-stalling behavior of a point. For example, if $f(x) = x^2 \mathbb{1}_{\{x \leq 0\}} - x \mathbb{1}_{\{x > 0\}}$, and we start GD from $x_0 < 0$ with a learning rate $\eta < \frac{1}{4}$, then the GD will be stuck at $x = 0$. Nevertheless, this point is not a directional stationary point. There might exist analogous situations for our setup where directional stationarity fails to capture GD-stalling points. However, Such situations have probability 0.

## C.2 EXAMPLES OF NON-DIFFERENTIABLE STATIONARY POINTS

Here, we give two examples of non-differentiable (directional) stationary points under GD.

### C.2.1 A NON-DIFFERENTIABLE LOCAL MINIMUM

In Figure 6, we show a non-differentiable local minimum of the function $f(x, y) = |x| + |y|$, which is the origin and denoted by the red x. It is not hard to see that GD performed on the function will approach this point and stop there (or bounce in its vicinity, which can be taken as "GD effectively comes to a halt", as phrased in Section 3.2).

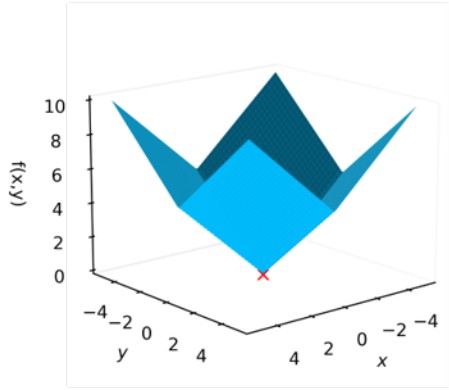

Figure 6: A non-differentiable local minimum

### C.2.2 A Non-differentiable Saddle Point

In Figure 7a, we illustrate a non-differentiable saddle point and GD's behavior near it. The function we investigate in this example is the following:

$$f(x, y) = \begin{cases} y^2 + x & \text{if } x > 0 \text{ and } y > 0 \\ -y^2 + x & \text{if } x > 0 \text{ and } y \leq 0 \\ y^2 - x & \text{if } x \leq 0 \text{ and } y > 0 \\ -y^2 - x & \text{if } x \leq 0 \text{ and } y \leq 0 \end{cases}, \tag{7}$$

which is showed in Figure 7a. The origin is a non-differentiable saddle point. If we perform GD starting from $(x, y) = (1e^{-6}, 1)$, the trajectory of GD will be as shown in Figure 7b, and the change of function value during the GD process will be as shown in Figure 7c. The behavior of GD is much like that near a differentiable saddle. However, one can show that, in this case, given a step size smaller than $\frac{1}{4}$, the non-differentiable saddle point is non-escapable by GD, even though it has second-order decreasing directions. This is in contrast with strict differentiable saddles, which are almost always escapable by GD (Lee et al., 2017; 2016; Jin et al., 2017; Daneshmand et al., 2018).

*Remark* C.4. Note, that the subgradient cannot be defined at the saddle point discussed here. We remind the reader that the subgradient is defined as:

*Definition* C.5. Let $f : \mathbb{R}^n \to \mathbb{R}$ be a convex function. The subgradient set of $f$ at $\mathbf{x}_0$ is defined as:

$$\partial f(\mathbf{x}_0) = \left\{ \mathbf{g} \in \mathbb{R}^n : f(\mathbf{x}) \geq f(\mathbf{x}_0) + \mathbf{g}^T(\mathbf{x} - \mathbf{x}_0), \forall \mathbf{x} \in \mathbb{R}^n \right\}$$

As a matter of fact, the definition of subgradient generally fails at saddle points due to non-convexity. One can verify that the subgradient cannot be defined for the origin of the loss landscape of our setup in general. Since the ODDs toward all the directions are zero at the origin (by Lemma 3.4), if the subgradient $\mathbf{g}$ can be defined, it must mean that $\mathbf{g} = \mathbf{0}$. Taking it to Definition C.5, we find that, if the subgradient at the origin were $\mathbf{g} = \mathbf{0}$, the origin would be a local minimum. However, the origin normally have loss-decreasing path around it.

### C.3 Connection between Directional Stationarity and Clarke Stationarity (Wang et al., 2022; Davis et al., 2020b)

We show that directional stationary points must also be Clarke stationary points. This insight was also given by Li et al. (2020). We prove it here for convenience, after which we show that directional stationary points are Clarke stationary points with no negative ODD around it.

**Proposition C.6.** *Given a function $f : \mathbb{R}^N \to \mathbb{R}$ whose ODD*

$$f'(\mathbf{x}, \mathbf{v}) := \lim_{h \searrow 0^+} \frac{f(\mathbf{x} + h\mathbf{v}) - f(\mathbf{x})}{h}$$

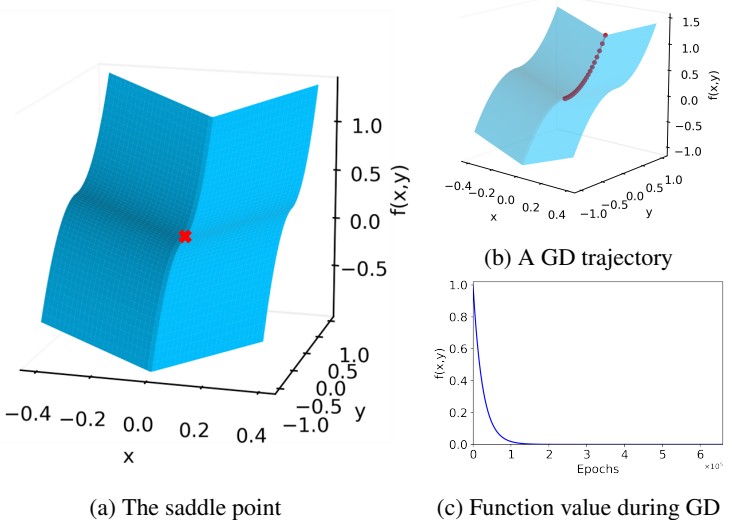

(a) The saddle point      (c) Function value during GD

(b) A GD trajectory

Figure 7: A non-differentiable saddle point

*can be defined for all* $\mathbf{x} \in \mathbb{R}^N$ *and for all* $\mathbf{v} \in \mathbb{R}^N$. *If* $\overline{\mathbf{x}}$ *satisfies that* $f(\overline{\mathbf{x}}, \mathbf{v}) \geq 0$, *for any* $\mathbf{v} \in \mathbb{R}^N$, *so that it is a directional stationary point, then it must also be a Clarke stationary point (Davis et al., 2020b; Wang et al., 2022).*

*Proof.* Clarke stationary points are defined to be where

$$\mathbf{0} \in \partial^\circ f(\mathbf{x}) := \left\{ \boldsymbol{\xi} \in \mathbb{R}^N : \langle \boldsymbol{\xi}, \mathbf{v} \rangle \leq f^\circ(\mathbf{x}, \mathbf{v}), \forall \mathbf{v} \in \mathbb{R}^N \right\}, \tag{8}$$

in which $f^\circ(\mathbf{x}, \mathbf{v})$ is the Clarke generalized directional derivative:

$$f^\circ(\mathbf{x}, \mathbf{v}) = \limsup_{\mathbf{y} \to \mathbf{x}, h \searrow 0+} \frac{f(\mathbf{y} + h\mathbf{v}) - f(\mathbf{y})}{h}. \tag{9}$$

Since Equation (9) is equivalent to taking a supremum over all the ODDs computed for neighboring point $\mathbf{y}$, we must have

$$f^\circ(\overline{\mathbf{x}}, \mathbf{v}) \geq f'(\overline{\mathbf{x}}, \mathbf{v}) \geq 0, \quad \forall \mathbf{v} \in \mathbb{R}^N, \tag{10}$$

in which the second inequality comes from the condition in Proposition C.6. Notice, Equation (10) immediately make Equation (8) hold, which proves the proposition. $\square$

**Fact C.7.** *Directional stationary points are Clarke stationary points that do not have negative ODDs toward any direction around it.*

*Proof.* Following the setup in Proposition C.6. Define $O := \{\mathbf{x} | f'(\mathbf{x}, \mathbf{v}) \geq 0, \forall \mathbf{v} \in \mathbb{R}^N\}$, $C := \{\mathbf{x} | \mathbf{0} \in f^\circ(\mathbf{x}, \mathbf{v})\}$, $L := \{\mathbf{x} | f'(\mathbf{x}, \mathbf{v}) < 0, \text{ for some } \mathbf{v} \in \mathbb{R}^N\}$. To prove the above fact, we need to prove $O = C \setminus L$. By definition $O = \overline{L}$. Thus, $C \setminus L = C \bigcap \overline{L} \subseteq O$. Also, we have that $O \subseteq C$ (which is proved in Proposition C.6) and $O = \overline{L}$, thus we have $O \subseteq O \setminus L$. As a result $O = C \setminus L$. $\square$

### C.4    DISCUSSION ON THE STATIONARITY PROPOSED BY CHERIDITO ET AL. (2022)

Cheridito et al. (2022) defined stationary points to be where the right-hand derivatives along the canonical axes are zero. Please refer to their Equation (2.3) and Definition 2.1 for details. Such a stationarity notion might not suit our purpose as the ODDs on canonical axes might not be informative enough to characterize the local structure of a function in all directions. We will demonstrate this with a toy example.

Consider the origin of the function $f(x, y, z) = |x-y| - |x+y| + z^2$. It has zero right-hand derivative on all the coordinate axes, and thus qualifies as a stationary point by the criterion proposed by Cheridito et al. (2022). However, there exist negative slopes around it, for example, in the direction of $(1, 1, 0)$. For such a function, the stationarity can be verified by:

(1) checking whether the ODDs toward all directions in $\mathbb{R}^3$ are non-negative, which is analogous to Definition 3.5; or

(2) checking whether the directional derivative along the $z$ axis, where the function is continuously differentiable, is zero; then checking whether the ODDs in all the directions on the $(x, y)$-plane are nonnegative; which is analogous to Definition 3.6.

## D  RARITY OF LOCAL MAXIMA

In theory, local maxima can exist in our setting Zhou and Liang (2018). Nonetheless, the theorem below suggests that they are extremely rare.

**Theorem D.1.** *If there exists some input weight vector* $\mathbf{w}_i$ *in* $\mathbf{P} \in \mathbb{R}^D$ *satisfying* $\rho(\mathbf{w}_i \cdot \mathbf{x}_k) \neq 0$ *for some input* $\mathbf{x}_k$*, then,* $\mathbf{P}$ *cannot be a local maximum.*

*Proof.* We need to show that when the parameters $\mathbf{P}$ satisfies the sufficient condition in Theorem D.1, there exist perturbations leading to strict loss increase. Let us focus on one of the hidden neurons (with the subscript of $i$) that has $\rho(\mathbf{w}_i \cdot \mathbf{x}_k) \neq 0$ for some $k \in K$. Let us also specify an output neuron indexed with $j$. We will construct a strictly loss-increasing path only by modifying $h_{ji}$.

Here we could simply use the derivatives that we computed before and show it's positive:

$$\frac{\partial \mathcal{L}(\overline{P})}{\partial h_{ji}} = \sum_{k \in K} e_{kj} \rho(\mathbf{w}_i \cdot \mathbf{x}_k),$$

$$\frac{\partial^2 \mathcal{L}(\overline{P})}{\partial h_{ji}^2} = \sum_{k \in K} \rho(\mathbf{w}_i \cdot \mathbf{x}_k)^2.$$

Hence, we see that either the first derivative is non-zero, in which case there is a loss-increasing direction since the loss is continuously differentiable in $h_{ji}$, or the first derivative is null and the second derivative is strictly positive, in which case we also get a loss-increasing path.

$\square$

*Remark* D.2. Liu (2021) proved the absence of differentiable local maxima for $\|J\| = 1$, and Botev et al. (2017) precluded the existence of differentiable strict local maxima for networks with piecewise linear activation functions. Here, we provide a more generic conclusion, that if $\mathbf{P}$ is a local maximum, no matter whether differentiable or not, then all input weight vectors are not "activating" any of the inputs in the dataset, meaning $\hat{\mathbf{y}}_k(\mathbf{P}) = 0$ for all $k \in K$.

## E  INTUITION OF DEFINITION 4.1 AND THEOREM 4.2

When the output dimension is one, we can understand why the escape neurons, as defined, can lead to loss-decreasing path. Let us denote the only output neuron with the subscript of $j_0$. For the escape neuron, we have that for some tangential direction $\mathbf{v}_i$ fixed, $\frac{\partial \mathcal{L}(\overline{\mathbf{P}})}{\partial s_i} = \overline{h}_{j_0 i} \mathbf{d}_{j_0 i}^{\mathbf{v}_i} \cdot \mathbf{v}_i = 0$ and $\mathbf{d}_{j_0 i}^{\mathbf{v}_i} \cdot \mathbf{v}_i \neq 0$. This must mean that $\overline{h}_{j_0 i} = 0$. If we slightly perturb $\overline{h}_{j_0 i}$ such that it has a different sign than $\mathbf{d}_{j_0 i}^{\mathbf{v}_i} \cdot \mathbf{v}_i$, then the tangential derivative $\frac{\partial \mathcal{L}(\overline{\mathbf{P}})}{\partial s_i}$ after the perturbation is negative, indicating a loss-decreasing direction.

## F  PROOF OF THEOREM 4.2

For brevity, we use $(a_l)_{l \in L}$ to denote a vector that is composed by stacking together the $a_l$'s with $l \in L$.

### F.1 PROOF OF THE SUFFICIENCY

We start by introducing the ODDs of the loss. Fix a unit vector $\Delta \in \mathbb{R}^D$ in the parameter space, along which we investigate the derivative. By doing so, we also fix the vectors $\mathbf{u}_i, \mathbf{v}_i$ for all $i \in I$, since the decomposition $\Delta_{\mathbf{w}_i} = \Delta_{r_i} \mathbf{u}_i + \Delta_{s_i} \mathbf{v}_i$ is unique, i.e. there exists a unique unit tangential vector $\mathbf{v}_i$ orthogonal with $\mathbf{u}_i = \frac{\mathbf{w}_i}{\|\mathbf{w}_i\|}$ and unique $r_i, s_i \geq 0$. By convention, if $\mathbf{w}_i = 0$, we set $\mathbf{u}_i = 0$ and $r_i = 0$. See Figure 8 for a visualization of $\mathbf{u}_i, \mathbf{v}_i$.

The ODD of the loss in direction $\Delta$ is then defined by

$$\partial_\Delta \mathcal{L}(\mathbf{P}) := \lim_{\epsilon \to 0+} \frac{\mathcal{L}(\mathbf{P} + \epsilon\Delta) - \mathcal{L}(\mathbf{P})}{\epsilon}. \tag{11}$$

Note that the limit always exists, since, as we saw in previous sections, the partial derivatives $\frac{\mathcal{L}(\mathbf{P})}{\partial s_i}, \frac{\mathcal{L}(\mathbf{P})}{\partial r_i}$ are always well defined. However, we stress that because of the non-differentiability, we can have that $\partial_\Delta \mathcal{L}(\mathbf{P}) \neq -\partial_{-\Delta} \mathcal{L}(\mathbf{P})$.

Consider a stationary point $\overline{\mathbf{P}}$ of $\mathcal{L}$ with no escape neuron. Also, consider a direction $\Delta$ such that $\partial_\Delta \mathcal{L}(\overline{\mathbf{P}}) = 0$. *Directions with $\partial_\Delta \mathcal{L}(\overline{\mathbf{P}}) > 0$ (which is possible due to the positive tangential slope at input weights) are guaranteed to increase the loss locally.* Our strategy is to show that the ODDs of higher orders are non-negative. For example, at order 2, this means that

$$\partial_\Delta^2 \mathcal{L}(\overline{\mathbf{P}}) := \lim_{\epsilon \to 0+} \frac{\partial_\Delta \mathcal{L}(\overline{\mathbf{P}} + \epsilon\Delta) - \partial_\Delta \mathcal{L}(\overline{\mathbf{P}})}{\epsilon} \geq 0. \tag{12}$$

As a side note, the loss of ReLU-like networks can be understood as numerous patches of linear network loss pieced together. Thus, moving along a fixed direction $\Delta$ from a given point locally exploits the loss of a linear network, which is a polynomial with respect to all the parameters. In other words, the loss landscape is piecewise differentiable ($C^\infty$). As a result, ODDs of any orders along $\Delta$ (whose computations are similar to Equations (11) and (12)) are always definable.

Then, we need to investigate the directions $\Delta$ such that $\partial_\Delta^2 \mathcal{L}(\overline{\mathbf{P}}) = 0$, since the loss could increase, decrease or remain constant in those directions with higher orders. We will see that at order 3, these directions also yield $\partial_\Delta^3 \mathcal{L}(\overline{\mathbf{P}}) = 0$, and at order 4, necessarily, $\partial_\Delta^4 \mathcal{L}(\overline{\mathbf{P}}) \geq 0$, and since all higher order derivatives are null[11], this yields the claim.

These will be achieved by investigating the Taylor expansion of the loss in the direction $\Delta$, which reads for small $\epsilon > 0$ as

$$\mathcal{L}(\overline{\mathbf{P}} + \epsilon\Delta) = \mathcal{L}(\overline{\mathbf{P}}) + \epsilon\partial_\Delta \mathcal{L}(\overline{\mathbf{P}}) + \frac{1}{2!}\epsilon^2 \partial_\Delta^2 \mathcal{L}(\overline{\mathbf{P}}) + \frac{1}{3!}\epsilon^3 \partial_\Delta^3 \mathcal{L}(\overline{\mathbf{P}}) + \frac{1}{4!}\epsilon^4 \partial_\Delta^4 \mathcal{L}(\overline{\mathbf{P}}).$$

#### F.1.1 SECOND-ORDER TERMS

We organize the Hessian matrix at $\overline{\mathbf{P}}$ correspondingly as:

$$H_\mathcal{L} = \begin{bmatrix} \overbrace{\dfrac{\partial^2 \mathcal{L}(\overline{\mathbf{P}})}{\partial h_{j_1 i_1} \partial h_{j_2 i_2}}}^{|I_h| \text{ columns}} & \vdots & \overbrace{\dfrac{\partial^2 \mathcal{L}(\overline{\mathbf{P}})}{\partial r_{i_1} \partial h_{j_2 i_2}}}^{|I_r| \text{ columns}} & \vdots & \overbrace{\dfrac{\partial^2 \mathcal{L}(\overline{\mathbf{P}})}{\partial s_{i_1} \partial h_{j_2 i_2}}}^{|I_s| \text{ columns}} \\ \hdashline \dfrac{\partial^2 \mathcal{L}(\overline{\mathbf{P}})}{\partial h_{j_1 i_1} \partial r_{i_2}} & \vdots & \dfrac{\partial^2 \mathcal{L}(\overline{\mathbf{P}})}{\partial r_{i_1} \partial r_{i_2}} & \vdots & \dfrac{\partial^2 \mathcal{L}(\overline{\mathbf{P}})}{\partial s_{i_1} \partial r_{i_2}} \\ \hdashline \dfrac{\partial^2 \mathcal{L}(\overline{\mathbf{P}})}{\partial h_{j_1 i_1} \partial s_{i_2}} & \vdots & \dfrac{\partial^2 \mathcal{L}(\overline{\mathbf{P}})}{\partial r_{i_1} \partial s_{i_2}} & \vdots & \dfrac{\partial^2 \mathcal{L}(\overline{\mathbf{P}})}{\partial s_{i_1} \partial s_{i_2}} \end{bmatrix} \begin{matrix} \left.\vphantom{\dfrac{\partial}{\partial}}\right\} |I_h| \text{ rows} \\ \left.\vphantom{\dfrac{\partial}{\partial}}\right\} |I_r| \text{ rows} \\ \left.\vphantom{\dfrac{\partial}{\partial}}\right\} |I_s| \text{ rows} \end{matrix} \tag{13}$$

**Computing the second order partial derivatives.** Recall the first order partial derivatives

$$\frac{\partial \mathcal{L}(\mathbf{P})}{\partial h_{ji}} = \sum_{k \in K} e_{kj} \rho(\mathbf{w}_i \cdot \mathbf{x}_k),$$

---

[11] All ODDs of the fourth order are constants, as shown in Appendix F.1.3.

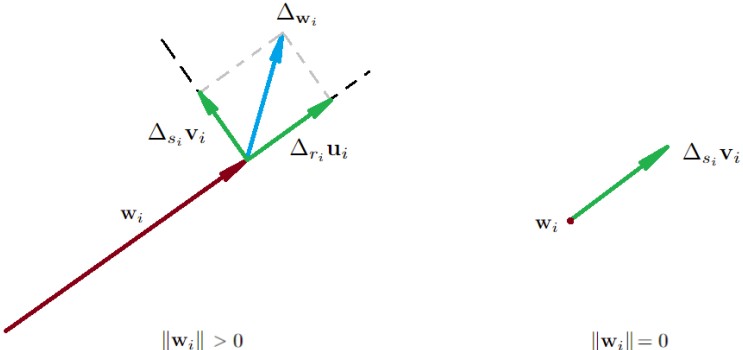

Figure 8: Breakdown of perturbation applied on input weights. Left: Perturbation on $\mathbf{w}_i$ with $\|\mathbf{w}_i\| > 0$ can be decomposed into the radial direction $\mathbf{u}_i$ and the tangential direction $\mathbf{v}_i$. Right: Perturbation applied on a zero input weight can only be decomposed into one direction $\mathbf{v}_i$.

$$\frac{\partial \mathcal{L}(\mathbf{P})}{r_i} = \sum_{j \in J} h_{ji} \sum_{k \in K} e_{kj} \rho(\mathbf{u}_i \cdot \mathbf{x}_k),$$

$$\frac{\partial \mathcal{L}(\mathbf{P})}{\partial s_i} = \sum_{j \in J} h_{ji} \sum_{k \in K} e_{kj} \widetilde{\rho}_{\mathbf{w}_i \cdot \mathbf{x}_k}(\mathbf{v}_i \cdot \mathbf{x}_k).$$

We now compute the second-order derivatives of the loss. Recall that these derivatives are defined for fixed radial and tangential directions, that is, the vectors $\mathbf{u}_i, \mathbf{v}_i, i \in I$ are fixed. We have

$$\frac{\partial^2 \mathcal{L}(\mathbf{P})}{\partial h_{j_1 i_1} \partial h_{j_2 i_2}} = \sum_{k \in K} \frac{\partial e_{k j_1}}{h_{j_2 i_2}} \rho(\mathbf{w}_i \cdot \mathbf{x}_k)$$

$$= \sum_{k \in K} \mathbb{1}_{\{j_1 = j_2\}} \rho(\mathbf{w}_{i_1} \cdot \mathbf{x}_k) \rho(\mathbf{w}_{i_2} \cdot \mathbf{x}_k),$$

$$\frac{\partial^2 \mathcal{L}(\mathbf{P})}{\partial r_{i_1} \partial h_{j i_2}} = \frac{\partial}{\partial h_{j i_2}} \sum_{j' \in J} \sum_{k \in K} e_{kj'} h_{j' i_1} \rho(\mathbf{u}_{i_1} \cdot \mathbf{x}_k)$$

$$= \sum_{k \in K} \left( e_{kj} \rho(\mathbf{u}_{i_1} \cdot \mathbf{x}_k) \mathbb{1}_{\{i_1 = i_2\}} + h_{j i_1} \rho(\mathbf{u}_{i_1} \cdot \mathbf{x}_k) \rho(\mathbf{w}_{i_2} \cdot \mathbf{x}_k) \right),$$

$$\frac{\partial^2 \mathcal{L}(\mathbf{P})}{\partial s_{i_1} \partial h_{j i_2}} = \frac{\partial}{\partial h_{j i_2}} h_{j i_1} \sum_{k \in K} e_{kj} \widetilde{\rho}_{\mathbf{w}_{i_1} \cdot \mathbf{x}_k}(\mathbf{v}_{i_1} \cdot \mathbf{x}_k)$$

$$= \sum_{k \in K} e_{kj} \widetilde{\rho}_{\mathbf{w}_{i_1} \cdot \mathbf{x}_k}(\mathbf{v}_{i_1} \cdot \mathbf{x}_k) \mathbb{1}_{\{i_1 = i_2\}} + h_{j i_1} \sum_{k \in K} \widetilde{\rho}_{\mathbf{w}_{i_1} \cdot \mathbf{x}_k}(\mathbf{v}_{i_1} \cdot \mathbf{x}_k) \rho(\mathbf{w}_{i_2} \cdot \mathbf{x}_k),$$

$$\frac{\partial^2 \mathcal{L}(\mathbf{P})}{\partial r_{i_1} \partial r_{i_2}} = \frac{\partial}{\partial r_2} \sum_{j \in J} \sum_{k \in K} e_{kj} h_{j i_1} \rho(\mathbf{u}_{i_1} \cdot \mathbf{x}_k)$$

$$= \sum_{j \in J} \sum_{k \in K} h_{j i_1} h_{j i_2} \rho(\mathbf{u}_{i_1} \cdot \mathbf{x}_k) \rho(\mathbf{u}_{i_2} \cdot \mathbf{x}_k),$$

$$\frac{\partial^2 \mathcal{L}(\mathbf{P})}{\partial s_{i_1} \partial r_{i_2}} = \frac{\partial}{\partial r_{i_2}} \sum_{j \in J} \sum_{k \in K} e_{kj} h_{j i_1} \widetilde{\rho}_{\mathbf{w}_{i_1} \cdot \mathbf{x}_k}(\mathbf{v}_{i_1} \cdot \mathbf{x}_k)$$

$$= \sum_{j \in J} \sum_{k \in K} h_{j i_1} h_{j i_2} \widetilde{\rho}_{\mathbf{w}_{i_1} \cdot \mathbf{x}_k}(\mathbf{v}_{i_1} \cdot \mathbf{x}_k) \rho(\mathbf{u}_{i_2} \cdot \mathbf{x}_k),$$

$$\frac{\partial^2 \mathcal{L}(\mathbf{P})}{\partial s_{i_1} \partial s_{i_2}} = \frac{\partial}{\partial s_{i_2}} \sum_{j \in J} \sum_{k \in K} e_{kj} h_{j i_1} \widetilde{\rho}_{\mathbf{w}_{i_1} \cdot \mathbf{x}_k}(\mathbf{v}_{i_1} \cdot \mathbf{x}_k)$$

$$= \sum_{j \in J} \sum_{k \in K} h_{ji_1} h_{ji_2} \widetilde{\rho}_{\mathbf{w}_{i_1} \cdot \mathbf{x}_k} (\mathbf{v}_{i_1} \cdot \mathbf{x}_k) \widetilde{\rho}_{\mathbf{w}_{i_2} \cdot \mathbf{x}_k} (\mathbf{v}_{i_2} \cdot \mathbf{x}_k)$$

**Second order partial derivatives at the stationary point.** When evaluated at the stationary point $\overline{\mathbf{P}}$, two of these derivatives simplify. Firstly, we note that, if $\|\mathbf{w}_i\| \neq 0$, then $\sum_{k \in K} (e_{kj} \rho(\mathbf{u}_{i_1} \cdot \mathbf{x}_k) \mathbb{1}_{\{i_1 = i_2\}} = \frac{1}{\|\mathbf{w}_i\|} \frac{\partial \mathcal{L}(\overline{\mathbf{P}})}{\partial h_{ji_1}} = 0$ by stationarity. If $\|\mathbf{w}_i\| = 0$, $\|\mathbf{u_i}\| = 0$ by convention so the sum is null as well. In any case, we have that

$$\frac{\partial^2 \mathcal{L}(\overline{\mathbf{P}})}{\partial r_{i_1} \partial h_{ji_2}} = 0 + \sum_{k \in K} h_{ji_1} \rho(\mathbf{u}_{i_1} \cdot \mathbf{x}_k) \rho(\mathbf{w}_{i_2} \cdot \mathbf{x}_k).$$

Similarly, since we choose the direction $\Delta$ such that $\partial_\Delta \mathcal{L}(\overline{\mathbf{P}}) = 0$, we must have that $\frac{\partial \mathcal{L}(\overline{\mathbf{P}})}{\partial s_i} = 0$. Since there is no escape neuron, necessarily, $\sum_{k \in K} e_{kj} \widetilde{\rho}_{\mathbf{w}_i \cdot \mathbf{x}_k} (\mathbf{v}_i \cdot \mathbf{x}_k) = \mathbf{d}_{ji}^{\mathbf{v}_i} \cdot \mathbf{v_i} = 0$. This implies that

$$\frac{\partial^2 \mathcal{L}(\mathbf{P})}{\partial s_{i_1} \partial h_{ji_2}} = 0 + h_{ji_1} \sum_{k \in K} \widetilde{\rho}_{\mathbf{w}_{i_1} \cdot \mathbf{x}_k} (\mathbf{v}_{i_1} \cdot \mathbf{x}_k) \rho(\mathbf{w}_{i_2} \cdot \mathbf{x}_k). \tag{14}$$

**The Hessian matrix is positive semidefinite.** Define the following three $|K|$-dimensional vectors: for $i \in I, j \in J$, let

$$\mathbf{V}_{h_{ji}} := (\rho(\mathbf{w}_i \cdot \mathbf{x}_k))_{k=1,\dots,|K|}, \tag{15}$$

$$\mathbf{V}_{r_i}^j := (h_{ji} \rho(\mathbf{u}_i \cdot \mathbf{x}_k))_{k=1,\dots,|K|}, \tag{16}$$

$$\mathbf{V}_{s_i}^j := (h_{ji} \widetilde{\rho}_{\mathbf{w}_i \cdot \mathbf{x}_k} (\mathbf{v}_i \cdot \mathbf{x}_k))_{k=1,\dots,|K|}. \tag{17}$$

Then, we assemble the vectors and for $j' \in J$, we define the matrix

$$V_{j'} := \left( \left( \mathbb{1}_{\{j=j'\}} \mathbf{V}_{h_{ji}} \right)_{(j,i) \in I_h}, \left( \mathbf{V}_{r_i}^{j'} \right)_{i \in I_r}, \left( \mathbf{V}_{s_i}^{j'} \right)_{i \in I_s} \right) \in \mathbb{R}^{|K| \times D'},$$

where $\mathbb{1}_{\{j=j'\}}$ is multiplied to each component of $\mathbf{V}_{h_{ji}}$. The vectors are disposed such that each is a column of $V^{j'}$. We thus see that the directional Hessian Equation (13) can be written as

$$\mathcal{H}_\mathcal{L} = \sum_{j' \in J} V_{j'}^\mathsf{T} V_{j'}. \tag{18}$$

Once again, like the vectors $\mathbf{u}_i, \mathbf{v}_i$ for $i \in I$, the Hessian matrix implicitly depends on the direction $\Delta$. Hence, for every fixed unit vector $\Delta \in \mathbb{R}^D$, the Hessian matrix $\mathcal{H}_\mathcal{L}$ is a sum of positive semidefinite Gram matrices, and as such, is positive semidefinite.

We deduce that for all unit vector $\Delta \in \mathbb{R}^D$, it holds that

$$\partial_\Delta^2 \mathcal{L}(\overline{\mathbf{P}}) = \Delta^\mathsf{T} \mathcal{H}_\mathcal{L} \Delta \geq 0,$$

as claimed.

### F.1.2 THIRD-ORDER TERMS

In this section, we continue our reasoning and assume that $\overline{\mathbf{P}}$ is a stationary point with no escape neuron and $\Delta \in \mathbb{R}^D$ is a fixed unitary vector such that $\partial_\Delta^2 \mathcal{L}(\overline{\mathbf{P}}) = \Delta^\mathsf{T} \mathcal{H}_\mathcal{L} \Delta = 0$. In particular, since $\mathcal{H}_\mathcal{L} = \sum_{j \ni J} V_j^\mathsf{T} V_j$ in this case, it holds that $V_j \Delta \in \mathbb{R}^{|K|}$ is a zero vector for all $j \in J$, where $V_j$ is defined below Equation (15). We show below that this entails $\partial_\Delta^3 \mathcal{L}(\overline{\mathbf{P}}) = 0$.

From the formulae for the second-order terms, we can see that there can only be six types of non-zero third-order derivatives, namely, $\frac{\partial^3 \mathcal{L}(\overline{\mathbf{P}})}{\partial h_{ji_1} \partial r_{i_1} \partial h_{ji_2}}$, $\frac{\partial^3 \mathcal{L}(\overline{\mathbf{P}})}{\partial h_{ji_1} \partial r_{i_1} \partial r_{i_2}}$, $\frac{\partial^3 \mathcal{L}(\overline{\mathbf{P}})}{\partial h_{ji_1} \partial r_{i_1} \partial s_{i_2}}$, $\frac{\partial^3 \mathcal{L}(\overline{\mathbf{P}})}{\partial h_{ji_1} \partial s_{i_1} \partial h_{ji_2}}$, $\frac{\partial^3 \mathcal{L}(\overline{\mathbf{P}})}{\partial h_{ji_1} \partial s_{i_1} \partial r_{i_2}}$, and $\frac{\partial^3 \mathcal{L}(\overline{\mathbf{P}})}{\partial h_{ji_1} \partial s_{i_1} \partial s_{i_2}}$.

Recall the vectors $\mathbf{V}_{h_{ji}}^{j'}, \mathbf{V}_{r_i}^{j'}, \mathbf{V}_{s_i}^{j'}$ defined in Equation (15), and defined the following vectors

$$\mathbf{V}_{hr_i} = (\rho(\mathbf{u}_i \cdot \mathbf{x}_k))_{k \in \{1,2,\cdots,K\}},$$

$$\mathbf{V}_{hs_i} = \left(\widetilde{\rho}_{\mathbf{w}_i \cdot \mathbf{x}_k}(\mathbf{v}_i \cdot \mathbf{x}_k)\right)_{k \in \{1,2,\cdots,K\}},$$

The third order partial derivatives can be conveniently expressed using them. For example, we compute the first one

$$\frac{\partial^3 \mathcal{L}(\overline{\mathbf{P}})}{\partial h_{ji_1} \partial r_{i_1} \partial h_{ji_2}} = \frac{\partial}{\partial h_{ji_2}} \sum_{k \in K} \left(e_{kj} \rho(\mathbf{u}_{i_1} \cdot \mathbf{x}_k) + h_{ji_1} \rho(\mathbf{u}_{i_1} \cdot \mathbf{x}_k) \rho(\mathbf{w}_{i_1} \cdot \mathbf{x}_k)\right)$$

$$= \left(1 + \mathbb{1}_{\{i_1 = i_2\}}\right) \sum_{k \in K} \rho(\mathbf{u}_{i_1} \cdot \mathbf{x}_k) \rho(\mathbf{w}_{i_2} \cdot \mathbf{x}_k)$$

$$= \left(1 + \mathbb{1}_{\{i_1 = i_2\}}\right) \mathbf{V}_{hr_{i_1}} \cdot \mathbf{V}_{h_{ji_2}}.$$

The other derivatives follow similar easy calculations that are left to the reader, yielding the following results:

$$\frac{\partial^3 \mathcal{L}(\overline{\mathbf{P}})}{\partial h_{ji_1} \partial r_{i_1} \partial r_{i_2}} = \left(1 + \mathbb{1}_{\{i_1 = i_2\}}\right) \mathbf{V}_{hr_{i_1}} \cdot \mathbf{V}^j_{r_{i_2}}$$

$$\frac{\partial^3 \mathcal{L}(\overline{\mathbf{P}})}{\partial h_{ji_1} \partial r_{i_1} \partial s_{i_2}} = \left(1 + \mathbb{1}_{\{i_1 = i_2\}}\right) \mathbf{V}_{hr_{i_1}} \cdot \mathbf{V}^j_{s_{i_2}}$$

$$\frac{\partial^3 \mathcal{L}(\overline{\mathbf{P}})}{\partial h_{ji_1} \partial s_{i_1} \partial h_{ji_2}} = \left(1 + \mathbb{1}_{\{i_1 = i_2\}}\right) \mathbf{V}_{hs_{i_1}} \cdot \mathbf{V}_{h_{ji_2}}$$

$$\frac{\partial^3 \mathcal{L}(\overline{\mathbf{P}})}{\partial h_{ji_1} \partial s_{i_1} \partial r_{i_2}} = \left(1 + \mathbb{1}_{\{i_1 = i_2\}}\right) \mathbf{V}_{hs_{i_1}} \cdot \mathbf{V}^j_{r_{i_2}}$$

$$\frac{\partial^3 \mathcal{L}(\overline{\mathbf{P}})}{\partial h_{ji_1} \partial s_{i_1} \partial s_{i_2}} = \left(1 + \mathbb{1}_{\{i_1 = i_2\}}\right) \mathbf{V}_{hs_{i_1}} \cdot \mathbf{V}^j_{s_{i_2}}.$$

In the third-order ODD of the loss at $\overline{\mathbf{P}}$ in direction $\Delta$, each of the above derivatives appear. More specifically, if $i_1 \neq i_2$, then any of them is counted $3! = 6$ times (the number of orderings that yield the same derivative) and if $i_1 = i_2$, then it is counted 3 times. We thus see that

$$\partial^3_\Delta \mathcal{L}(\overline{\mathbf{P}}) = \sum_{j \in J} \left(6 \sum_{i_1 \neq i_2 \in I} \Delta_{h_{ji_1}} \Delta_{r_{i_1}} \mathbf{V}_{hr_{i_1}} \left(\Delta_{h_{ji_2}} V_{h_{ji_2}} + \Delta_{r_{i_2}} V_{r_{i_2}} + \Delta_{s_{i_2}} V_{s_{i_2}}\right)\right.$$

$$\left. + 2 \times 3 \sum_{i_1 \in I} \Delta_{h_{ji_1}} \Delta_{r_{i_1}} \mathbf{V}_{hr_{i_1}} \left(\Delta_{h_{ji_1}} V_{h_{ji_1}} + \Delta_{r_{i_1}} V_{r_{i_1}} + \Delta_{s_{i_1}} V_{s_{i_1}}\right)\right)$$

$$+ \sum_{j \in J} \left(6 \sum_{i_1 \neq i_2 \in I} \Delta_{h_{ji_1}} \Delta_{s_{i_1}} \mathbf{V}_{hs_{i_1}} \left(\Delta_{h_{ji_2}} V_{h_{ji_2}} + \Delta_{r_{i_2}} V_{r_{i_2}} + \Delta_{s_{i_2}} V_{s_{i_2}}\right)\right.$$

$$\left. + 2 \times 3 \sum_{i_1 \in I} \Delta_{h_{ji_1}} \Delta_{s_{i_1}} \mathbf{V}_{hs_{i_1}} \left(\Delta_{h_{ji_1}} V_{h_{ji_1}} + \Delta_{r_{i_1}} V_{r_{i_1}} + \Delta_{s_{i_1}} V_{s_{i_1}}\right)\right)$$

$$= 6 \sum_{j \in J} \sum_{i_1, i_2 \in I} \Delta_{h_{ji_1}} \left(\Delta_{r_{i_1}} \mathbf{V}_{hr_{i_1}} + \Delta_{s_{i_1}} \mathbf{V}_{hs_{i_1}}\right) V_j \Delta$$

Recall from the discussion at the start of the subsection that since $\overline{\mathbf{P}}$ is a stationary point with no escape neuron and since $\partial^2_\Delta \mathcal{L}(\overline{\mathbf{P}}) = 0$, it holds that

$$V_j \Delta = 0. \tag{19}$$

This shows that $\partial^3_\Delta \mathcal{L}(\overline{\mathbf{P}}) = 0$, as claimed.

### F.1.3 FOURTH-ORDER TERMS

This is the last step of our argument, where we show that $\partial^4_\Delta \mathcal{L}(\overline{\mathbf{P}}) \geq 0$ always holds.

We compute the fourth-order partial derivatives of the loss. Note from the third order partial derivatives that only three of them can be non null, namely $\frac{\partial^4 \mathcal{L}(\overline{\mathbf{P}})}{\partial h_{ji_1} \partial r_{i_1} \partial h_{ji_2} \partial r_{i_2}}$, $\frac{\partial^4 \mathcal{L}(\overline{\mathbf{P}})}{\partial h_{ji_1} \partial s_{i_1} \partial h_{ji_2} \partial s_{i_2}}$ and

$\frac{\partial^4 \mathcal{L}(\overline{\mathbf{P}})}{\partial h_{ji_1} \partial r_{i_1} \partial h_{ji_2} \partial s_{i_2}}$. It is straightforward – but cumbersome therefore left to the reader – to check the following:

$$\frac{\partial^4 \mathcal{L}(\overline{\mathbf{P}})}{\partial h_{ji_1} \partial r_{i_1} \partial h_{ji_2} \partial r_{i_2}} = (1 + \mathbb{1}_{\{i_1 = i_2\}}) \mathbf{V}_{hr_{i_1}} \cdot \mathbf{V}_{hr_{i_2}}$$

$$\frac{\partial^4 \mathcal{L}(\overline{\mathbf{P}})}{\partial h_{ji_1} \partial s_{i_1} \partial h_{ji_2} \partial s_{i_2}} = (1 + \mathbb{1}_{\{i_1 = i_2\}}) \mathbf{V}_{hs_{i_1}} \cdot \mathbf{V}_{hs_{i_2}},$$

$$\frac{\partial^4 \mathcal{L}(\overline{\mathbf{P}})}{\partial h_{ji_1} \partial r_{i_1} \partial h_{ji_2} \partial s_{i_2}} = (1 + \mathbb{1}_{\{i_1 = i_2\}}) \mathbf{V}_{hr_{i_1}} \cdot \mathbf{V}_{hs_{i_2}}.$$

As for the third order, depending on whether $i_1$ equals $i_2$, the number of occurrences of these derivatives changes between $4!$ and $4!/2$, and the fourth-order ODD reads

$$\partial_\Delta^4 \mathcal{L}(\overline{\mathbf{P}}) = \sum_{j \in J} \left( \sum_{i_1 \neq i_2 \in I} 4! \Delta_{h_{ji_1}} \Delta_{r_{i_1}} \Delta_{h_{ji_2}} \Delta_{r_{i_2}} \mathbf{V}_{hr_{i_1}} \cdot \mathbf{V}_{hr_{i_2}} + \sum_{i_1 \in I} 2 \frac{4!}{2 \times 2} \Delta_{h_{ji_1}}^2 \Delta_{r_{i_1}}^2 \mathbf{V}_{hr_{i_1}} \cdot \mathbf{V}_{hr_{i_1}} \right)$$

$$+ \sum_{j \in J} \left( \sum_{i_1 \neq i_2 \in I} 4! \Delta_{h_{ji_1}} \Delta_{s_{i_1}} \Delta_{h_{ji_2}} \Delta_{s_{i_2}} \mathbf{V}_{hs_{i_1}} \cdot \mathbf{V}_{hs_{i_2}} + \sum_{i_1 \in I} 2 \frac{4!}{2 \times 2} \Delta_{h_{ji_1}}^2 \Delta_{r_{i_1}}^2 \mathbf{V}_{hs_{i_1}} \cdot \mathbf{V}_{hs_{i_1}} \right)$$

$$+ \sum_{j \in J} \left( \sum_{i_1 \neq i_2 \in I} 4! \Delta_{h_{ji_1}} \Delta_{r_{i_1}} \Delta_{h_{ji_2}} \Delta_{s_{i_2}} \mathbf{V}_{hr_{i_1}} \cdot \mathbf{V}_{hs_{i_2}} + \sum_{i_1 \in I} 2 \frac{4!}{2} \Delta_{h_{ji_1}}^2 \Delta_{r_{i_1}} \Delta_{s_{i_1}} \mathbf{V}_{hr_{i_1}} \cdot \mathbf{V}_{hs_{i_1}} \right)$$

$$= \frac{4!}{2} \sum_{j \in J} \left( \sum_{i_1, i_2 \in I} \Delta_{h_{ji_1}} \Delta_{r_{i_1}} \Delta_{h_{ji_2}} \Delta_{r_{i_2}} \mathbf{V}_{hr_{i_1}} \cdot \mathbf{V}_{hr_{i_2}} + \Delta_{h_{ji_1}} \Delta_{s_{i_1}} \Delta_{h_{ji_2}} \Delta_{s_{i_2}} \mathbf{V}_{hs_{i_1}} \cdot \mathbf{V}_{hs_{i_2}} \right.$$

$$\left. + 2 \Delta_{h_{ji_1}} \Delta_{r_{i_1}} \Delta_{h_{ji_2}} \Delta_{s_{i_2}} \mathbf{V}_{hr_{i_1}} \cdot \mathbf{V}_{hs_{i_2}} \right)$$

$$= \frac{4!}{2} \sum_{j \in J} \left( \sum_{i_1, i_2 \in I} (\Delta_{h_{ji_1}} \Delta_{r_{i_1}} \mathbf{V}_{hr_{i_1}} + \Delta_{h_{ji_1}} \Delta_{s_{i_1}} \mathbf{V}_{hs_{i_1}}) \cdot (\Delta_{h_{ji_2}} \Delta_{r_{i_2}} \mathbf{V}_{hr_{i_2}} + \Delta_{h_{ji_2}} \Delta_{s_{i_2}} \mathbf{V}_{hs_{i_2}}) \right)$$

$$= \frac{4!}{2} \sum_{j \in J} \overline{\mathbf{V}}_j^\intercal \overline{\mathbf{V}}_j,$$

where we just introduced the vector $\overline{\mathbf{V}}_j$, defined by

$$\left( \sum_{i \in I} \Delta_{h_{ji}} \Delta_{r_i} \mathbf{V}_{hr_i} + \Delta_{h_{ji}} \Delta_{s_i} \mathbf{V}_{hs_i} \right)_{j \in J}.$$

We see from the above that $\partial_\Delta \mathcal{L}(\overline{\mathbf{P}}) \geq 0$, as claimed, which concludes the proof of the sufficiency in Theorem 4.2.

*Remark* F.1 (about Fact 4.7). Note, at a stationary point, if a perturbation direction $\Delta$ does not perturb the parameters of escape neurons, then the above proof also effectively shows that such a perturbation can not strictly decrease the loss, which gives rise to Fact 4.7.

## F.2 PROOF OF THE NECESSITY IN THEOREM 4.2

The necessity of the condition in Theorem 4.2, which holds in the scalar-output case, will be proved in the following via contradiction. Namely, if $\overline{\mathbf{P}}$ is a stationary point on the loss landscape with at least one hidden neuron being an escape neuron, then this stationary point cannot be a local minimum. Let us select one such hidden neuron $i$, associated with $(\overline{h}_{j_0 i}$ and $\overline{\mathbf{w}}_i)$. To construct the loss-decreasing path, we perturb $\overline{h}_{j_0 i}$ and $\overline{\mathbf{w}}_i$ with $\Delta h_{j_0 i}$ and $\Delta \mathbf{w}_i = \Delta s_i \mathbf{v}_i$, respectively; $\mathbf{v}_i$ being one of the tangential directions satisfying and $\frac{\partial \mathcal{L}(\overline{\mathbf{P}})}{\partial s_i} = 0$ and $\mathbf{d}_{j_0 i}^{\mathbf{v}_i} \cdot \mathbf{v}_i \neq 0$. We assert that we can design the following perturbation to strictly decrease the loss: $\Delta h_{ji} = -\operatorname{sgn}(\mathbf{d}_{j_0 i}^{\mathbf{v}_i} \cdot \mathbf{v}_i) a, \Delta s_i = b$, with $a, b > 0$ sufficiently small and satisfying certain conditions, which is discussed below.

To study the loss change after perturbing the weights as described, we resort to Taylor expansion again. In this case, we only need to look into terms of no higher than the second order.

For the first-order terms, we have:

$$T_1(\overline{\mathbf{P}}, \Delta\mathbf{P}') = \frac{\partial\mathcal{L}(\overline{\mathbf{P}})}{\partial h_{j_0 i}}\Delta h_{j_0 i} + \frac{\partial\mathcal{L}(\overline{\mathbf{P}})}{\partial s_i}\Delta s_i. \tag{20}$$

Since $\overline{\mathbf{P}}$ is a stationary point, we must have $\frac{\partial\mathcal{L}(\overline{\mathbf{P}})}{\partial h_{j_0 i}} = 0$. Moreover, we have $\frac{\partial\mathcal{L}(\overline{\mathbf{P}})}{\partial s_i} = 0$, as this is how we choose the tangential perturbation direction $\mathbf{v}_i$. Thus we have $T_1(\overline{\mathbf{P}}, \Delta\mathbf{P}') = 0$.

Then we calculate the second-order terms. Three types of second-order derivatives are involved, $\frac{\partial^2\mathcal{L}(\overline{\mathbf{P}})}{\partial h_{j_0 i}{}^2}$, $\frac{\partial^2\mathcal{L}(\overline{\mathbf{P}})}{\partial s_i^2}$, and $\frac{\partial^2\mathcal{L}(\overline{\mathbf{P}})}{\partial h_{j_0 i}\partial s_i}$. Notice that, in Equation (14), we could simplify the last derivative given that the neuron is not an escape neuron, which we can no longer do here. This causes the Hessian matrix to be indefinite.

Notice that, we must also have $\overline{h}_{j_0 i} = 0$ to fulfill the requirements of an escape neuron in the scalar output case. Hence, we have $\mathbf{V}_{s_i}^{j_0} = 0$. Thus, the second-order terms can be derived as:

$$T_2(\overline{\mathbf{P}}, \Delta\mathbf{P}') = \frac{\partial^2\mathcal{L}(\overline{\mathbf{P}})}{\partial h_{j_0 i}{}^2}(\Delta h_{j_0 i})^2 + \frac{\partial^2\mathcal{L}(\overline{\mathbf{P}})}{\partial s_i^2}(\Delta s_i)^2 + 2\frac{\partial^2\mathcal{L}(\overline{\mathbf{P}})}{\partial h_{j_0 i}\partial s_i}\Delta h_{j_0 i}\Delta s_i \tag{21a}$$

$$= \mathbf{V}_{h_{j_0 i}}\cdot\mathbf{V}_{h_{j_0 i}}(\Delta h_{j_0 i})^2 + \underbrace{\mathbf{V}_{s_i}^{j_0}\cdot\mathbf{V}_{s_i}^{j_0}}_{(=0)}(\Delta s_i)^2 + 2\big(\mathbf{d}_{j_0 i}^{\mathbf{v}}\cdot\mathbf{v}_i + \mathbf{V}_{h_{j_0 i}}\cdot\underbrace{\mathbf{V}_{s_i}^{j_0}}_{(=0)}\big)\Delta h_{j_0 i}\Delta s_i \tag{21b}$$

$$= \|\mathbf{V}_{h_{j_0 i}}\|^2 a^2 + 2\big(\mathbf{d}_{j_0 i}^{\mathbf{v}_i}\cdot\mathbf{v}_i\big)\big(-\operatorname{sgn}(\mathbf{d}_{j_0 i}^{\mathbf{v}_i}\cdot\mathbf{v}_i)\big)ab \tag{21c}$$

$$= \|\mathbf{V}_{h_{j_0 i}}\|^2 a^2 - 2|\mathbf{d}_{j_0 i}^{\mathbf{v}_i}\cdot\mathbf{v}_i|ab \tag{21d}$$

Note, we have specified that both $a$ and $b$ are positive. Thus, if we choose these two numbers such that:

$$b > \frac{\|\mathbf{V}_{h_{j_0 i}}\|^2}{2|\mathbf{d}_{j_0 i}^{\mathbf{v}_i}\cdot\mathbf{v}_i|}a. \tag{22}$$

Then we have $T_2(\overline{\mathbf{P}}, \Delta\mathbf{P}')$ being strictly negative, indicating a strictly loss-decreasing path, which completes the proof.

*Remark* F.2 (about Corollary 4.4). When we perturb the parameters of the escape neurons as above, we are exploiting second-order loss-decreasing paths, which yields the statement of Corollary 4.4.

### F.2.1 WHY CANNOT WE PROVE THE NECESSITY FOR NETWORKS WITH MULTIPLE OUTPUT NEURONS?

The construction of the loss-decreasing path in this section is based on the observation that the Hessian matrix is no longer a positive semi-definite matrix when there exist escape neurons for $|J| = 1$ case. However, in the case where $|J| > 1$, we cannot draw a similar conclusion. When the network has multiple output neurons, there is currently no guarantee that escape neurons will introduce negative eigenvalues in the Hessian matrix. More specifically, like the $|J| = 1$ case, some $\frac{\partial^2\mathcal{L}(\overline{\mathbf{P}})}{\partial s_i\partial h_{ji}}$ term in the Hessian matrix is no longer simply a dot product of two vectors, $\mathbf{V}_{s_i}^j\cdot\mathbf{V}_{h_{ji}}$, but is added with a non-zero term of $\mathbf{d}_{ji}^{\mathbf{v}_i}\cdot\mathbf{v}_i$. Owing to this, we can no longer write the Hessian matrix as a sum of Gram matrices as in Equation (18). Nonetheless, this do not necessarily mean that the Hessian matrix admits negative eigenvalues, which is why the necessity in Theorem 4.2 does not hold for general cases.

More concretely, the Hessian matrix for a multidimensional output network at a stationary point with escape neurons can be denoted as:

$$H_{\mathcal{L}} = \sum_{j\in J}\underbrace{\big(\tilde{H}_j + V_j^{\mathsf{T}}V_j\big)}_{\text{defined to be } H_{\mathcal{L}}^j} \tag{23}$$

Compared with Equation (18), Equation (23) has an extra term ($\tilde{H}_j$) in the summation which contains the terms of $\mathbf{d}_{ji}^{\mathbf{v}_i} \cdot \mathbf{v}_i \neq 0$ at the positions held by $\frac{\partial^2 \mathcal{L}(\overline{\mathbf{P}})}{\partial h_{ji} \partial s_i}$ or $\frac{\partial^2 \mathcal{L}(\overline{\mathbf{P}})}{\partial s_i \partial h_{ji}}$ in the Hessian matrix. It is easy to see that $H_{\mathcal{L}}^j$ is not positive semi-definite as long as $\tilde{H}_j$ is not a zero matrix.

Consequently, if the sufficient condition in Theorem 4.2 is not satisfied, then the Hessian matrix constructed based on the perturbation becomes a summation of matrices, some of which are indefinite. However, there can be no theoretical guarantee that such a summation cannot lead to a positive semi-definite matrix in the end. Thus, even if the sufficient condition in Theorem 4.2 is violated, the Hessian matrix may still be positive semi-definite. *However, despite the possibility of this case, the chance of a summation of indefinite matrices and positive semidefinite matrices resulting in a positive semidefinite matrix is relatively small. Even if the Hessian matrix is positive semidefinite, the existence of escape neurons seems to hinder us from claiming anything general regarding the loss change incurred by the third-order terms.*[12] *This is because our current discussion regarding the third-order terms (in Appendix F.1.2) relies on the description of the flat directions in the second-order terms being Equation* (19)*, which is premised on the absence of escape neurons. Hence, we believe that local minima contradicting the sufficient condition of Theorem 4.2 (the so-called type-2 local minima) should be rare if exist. The rigorous characterization regarding the "rarity" will be conducted in the future.*

## G  TAYLOR EXPANSION WITH AN ALIGNED COORDINATE SYSTEM

Taylor expansion requires the function to be differentiable. As a result, the non-differentiability of ReLU-like networks prohibits us from directly invoking Taylor expansion (based on the canonical coordinate system). We circumvent this problem by choosing a coordinate system whose axes correspond to the radial/tangential directions of the input weights and the output weights. We clarify this with the following example.

We study a one-hidden-layer ReLU network with 2-dimensional input and scalar output, and there is only one hidden neuron. The input weight is $\mathbf{w}_1 = (w_{11}, w_{12})$, and the output weight is $h_{j_01}$. There is one training sample, denoted by $\mathbf{x}_1 = (0.9, 0.3)^\intercal$ and $y_1 = 3$. We fix $h_{j_01} = 0.5$ and plot the loss as a function of $(w_{11}, w_{12})$ in both panels of Figure 9. The dotted gray line segments in the plots indicate a non-differentiable edge on the loss surface. We investigate a specific input weight $(0.15, -0.45)$, denoted by the black arrows in Figure 9.

In Figure 9a, we study how the movement of the input weight changes the loss with the canonical axes (the green arrows). Each canonical axis corresponds to an input weight component. More concretely, if we perform Taylor expansion with these axes, we will compute the partial derivatives of $\frac{\partial \mathcal{L}}{\partial (\pm w_{11})}$, $\frac{\partial \mathcal{L}}{\partial (\pm w_{12})}$, $\frac{\partial^2 \mathcal{L}}{\partial (\pm w_{11}) \partial (\pm w_{12})}$, etc. The "$\pm$" sign here means that we need to compute the ODD along both the positive and negative directions along the axes to account for non-differentiability. Nonetheless, these partial derivatives do not suffice to describe the function surface since non-differentiability exists within the orthants between the axes. As a result, the ODDs computed along such canonical axes cannot compose Taylor expansions that accurately characterize the loss surface within the orthants.

In this paper, we study the axes that are aligned with the non-differentiability on the loss surface, which is indicated by the red arrows in Figure 9b. Notice that these red arrows are exactly the radial and tangential directions of the input weight $\mathbf{w}_1$. The loss within any orthant bounded by these aligned axes is always differentiable. Hence, the partial derivatives with respect to these axes suffice to compose Taylor expansion that accurately describes the function anywhere in any orthant. Such a method is equivalent to studying differentiable functions but limiting the scope to only one specific orthant. Specifically, we limit the scope of Taylor expansion to one orthant by taking $\Delta s_i \searrow 0^+$ rather than $\Delta s_i \to 0$ in Equation (3).

Some back-of-the-envelope derivation can reveal that the loss is always continuously differentiable along the radial direction of input weights, which can also be observed in Figure 9b. Hence, the partial derivative along the same direction as $\mathbf{w}_1$ suffices to describe the radial derivative, which means we do not need to study the partial derivative along the direction of $-\mathbf{w}_1$. However, as the

---

[12]Note that the fourth-order terms always lead to non-negative loss changes, as shown in Appendix F.1.3

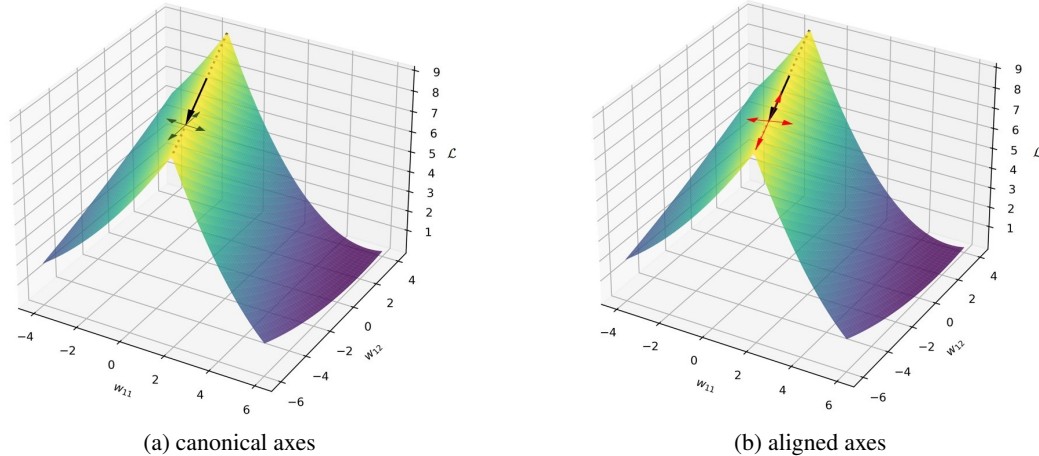

(a) canonical axes            (b) aligned axes

Figure 9: *Comparison between the canonical axes and the aligned axes*. The curved surface is the loss plotted as a function of the input weight $\mathbf{w}_1 = (w_{11}, w_{12})$ of the only hidden neuron, which is non-differentiable. The black arrow $(0.15, -0.45)$ denotes an input weight vector that has the same direction as the non-differentiable edge (denoted by the gray dotted line segments). The green arrows in **(a)** represent the canonical axes, while the red arrows in **(b)** represent the aligned coordinate axes (composed of the radial directions and tangential directions of $\mathbf{w}_1$).

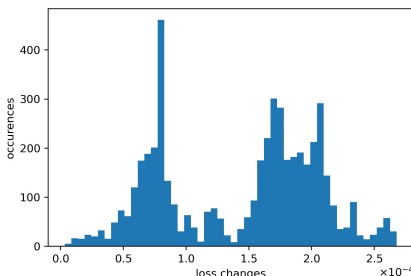

Figure 10: An occurrence histogram showing the loss changes incurred by the 5000 perturbations.

loss is not differentiable along the tangential directions, we need to study the partial derivative along both tangential directions of $\mathbf{w}_1$.

## H    VERIFYING LOCAL MINIMALITY IN A REAL CASE

The fact that the last long plateau in Figure 3a corresponds to a local minimum can be verified with a numerical experiment. The verification can be conducted by perturbing the parameters at the end of the training with small noises and investigating whether such perturbations yield negative loss change. In our experiment, we perturb all the parameters by adding noise to all of them. The noise added to each of the parameters is independently generated from a zero-centered uniform distribution $\mathcal{U}(-\zeta, \zeta)$, where $\zeta > 0$ is a small value making sure that the network parameters after perturbation still stays in its neighborhood. Namely, the loss function restricted on the line segment connecting the parameters before and after the perturbation should be $C^{\infty}$, meaning the line segment should not stretch across a non-differentiable area in the parameter space. In our case, we chose $\zeta \approx 1.28 \times 10^{-4}$. We conducted the perturbation for 5000 times and recorded the loss change incurred by each perturbation, which is illustrated in a frequency histogram in Figure 10. One can see that all of the perturbations resulted in positive loss change, agreeing with our theoretical prediction.

# I  ADDITIONAL NUMERICAL EXPERIMENTS

## I.1  3-DIMENSIONAL INPUT, SCALAR OUTPUT

In this example, we train the network with the following $\mathbf{x}_k$'s as inputs:

$$\mathbf{x}_1 = \begin{pmatrix} -0.3 \\ -0.75 \\ -0.5 \end{pmatrix}, \quad \mathbf{x}_2 = \begin{pmatrix} -0.2 \\ -0.2 \\ 0.4 \end{pmatrix}, \quad \mathbf{x}_3 = \begin{pmatrix} -0.6 \\ 1.0 \\ -1.0 \end{pmatrix}, \quad \mathbf{x}_4 = \begin{pmatrix} -0.4 \\ 0.4 \\ 0.3 \end{pmatrix},$$

$$\mathbf{x}_5 = \begin{pmatrix} 0.6 \\ -0.1 \\ -0.7 \end{pmatrix}, \quad \mathbf{x}_6 = \begin{pmatrix} 0.4 \\ -0.9 \\ 0.3 \end{pmatrix}, \quad \mathbf{x}_7 = \begin{pmatrix} 0.2 \\ 0.2 \\ -0.5 \end{pmatrix}.$$

The targets $y_k$ are as follows: $y_1 = -0.5$, $y_2 = 0.1$, $y_3 = -0.6$, $y_4 = 0.3$, $y_5 = 0.8$, $y_6 = -0.3$, $y_7 = -0.1$.

We use a one-hidden-layer ReLU network with 50 hidden neurons whose parameters are initialized with law $\mathcal{N}\left(0, (9.51 \times 10^{-11})^2\right)$. The training process is visualized in Figure 11, which is quite similar to that in Figure 3.

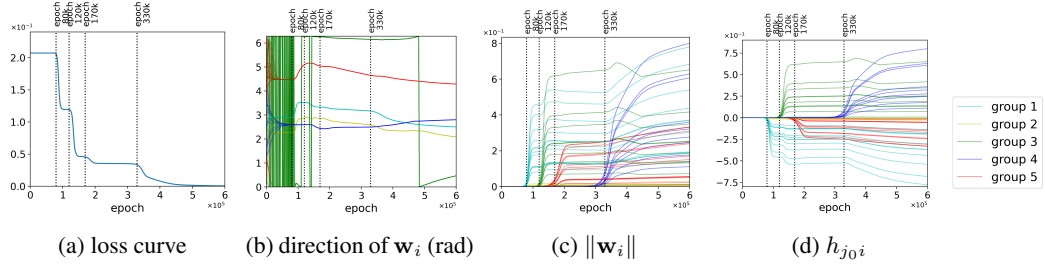

(a) loss curve    (b) direction of $\mathbf{w}_i$ (rad)    (c) $\|\mathbf{w}_i\|$    (d) $h_{j_0 i}$

Figure 11: *Evolution of all parameters during the training process from vanishing initialization. The network visualized here has 3-dimensional input and scalar output. The direction of $\mathbf{w}_i$ in* (b) *is the direction of its first two components. The training dynamics is similar to that in Figure 3. addle escape (marked by the vertical dotted lines) is always accompanied by the amplitude growth of grouped small living neurons.*

In short, the existence of small living neurons indicates that the stationary point is a saddle point, while their absence means that the stationary point is a local minimum. Moreover, saddle escape is always triggered by the amplitude increase of small living neurons. In this example, the training trajectory escaped from four saddle points and eventually ended up at the global minimum with zero loss.

It is worth noting that, in the above example, when escaping from the first saddle, two groups of small living neurons (group 1 and 2) grew together, which is also consistent with Figure 4.

## I.2  2-DIMENSIONAL INPUT AND OUTPUT

In this example, we train the network with the following $\mathbf{x}_k$'s as inputs:

$$\mathbf{x}_1 = \begin{pmatrix} -0.3 \\ 0.5 \end{pmatrix}, \quad \mathbf{x}_2 = \begin{pmatrix} 1.0 \\ 1.0 \end{pmatrix}, \quad \mathbf{x}_3 = \begin{pmatrix} -0.6 \\ -1.0 \end{pmatrix}, \quad \mathbf{x}_4 = \begin{pmatrix} 0.4 \\ -0.4 \end{pmatrix}.$$

The targets $\mathbf{y}_k$'s are:

$$\mathbf{y}_1 = \begin{pmatrix} 0.6 \\ -0.5 \end{pmatrix}, \quad \mathbf{y}_2 = \begin{pmatrix} 0.5 \\ -1.0 \end{pmatrix}, \quad \mathbf{y}_3 = \begin{pmatrix} -0.4 \\ 0.6 \end{pmatrix}, \quad \mathbf{y}_4 = \begin{pmatrix} 0.8 \\ 0.2 \end{pmatrix}.$$

We use a one-hidden-layer ReLU network with 50 hidden neurons whose parameters are initialized with law $\mathcal{N}\left(0, (9.51 \times 10^{-11})^2\right)$. The training process is visualized in Figure 12.

This training process escaped from 3 saddles. Notably, the last saddle escape involves the parameter variation of neurons that are not small living neurons, which contrasts the scalar-output case,

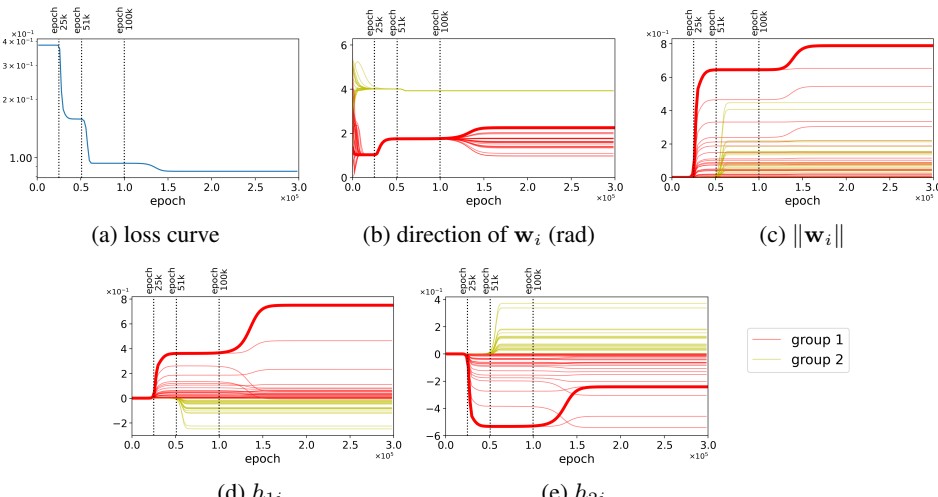

(a) loss curve      (b) direction of $\mathbf{w}_i$ (rad)      (c) $\|\mathbf{w}_i\|$

(d) $h_{1i}$      (e) $h_{2i}$

Figure 12: *Evolution of all parameters during the training process from vanishing initialization.* The network visualized here has 2-dimensional input and output. In this case, the output weights associating with one hidden neuron have 2 components, tracked by **(d)** and **(e)**. The saddle escape (marked by vertical dotted lines) can be caused by the amplitude increasing of small living neurons or the splitting of neurons that already have non-negligible amplitudes. The trajectory of one neuron is thickened in **(b)**-**(e)** for discussion.

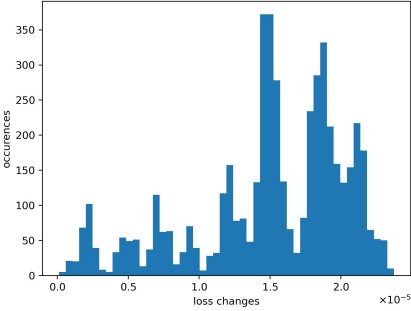

Figure 13: An occurrence histogram showing the loss changes incurred by the 5000 perturbations applied to the end of training in the 2D output case.

as escape neurons in the multidimensional output case might have non-zero output weights (Definition 4.1).

Eventually, the training reached a local minimum. We verify that it is a local minimum by locally perturbing the network parameter at the end of training with independently drawn noise for 5000 times. The resulting changes in loss are all positive. The occurrence histogram demonstrating the loss changes incurred by the 5000 perturbations are shown in Figure 13.

As it is a complicated case, **we identify the escape neurons in each saddle encountered during training**. To facilitate the presentation, we first present the evolution of errors at both output components, $e_{k1} \coloneqq \hat{y}_{k1} - y_{k1}$ and $e_{k2} \coloneqq \hat{y}_{k2} - y_{k2}$, which are shown in Figure 14.

*The first saddle point is the origin.* As all the output weights are zero, we know that the $\frac{\partial \mathcal{L}}{\partial s_i} = 0$ for any $i \in I$ and tangential direction $\mathbf{v}_i$ (from Equation (6)). We remind the reader that $\mathbf{v}_i$ is the tangential direction implied in the tangential derivative $\frac{\partial \mathcal{L}}{\partial s_i}$. We will show that there exists a specific tangential direction $\hat{\mathbf{v}}_i$ such that $\mathbf{d}_{1i}^{\hat{\mathbf{v}}_i} \cdot \hat{\mathbf{v}}_i \neq 0$, which makes the origin a saddle point by Definition 4.1 and Theorem 4.2.

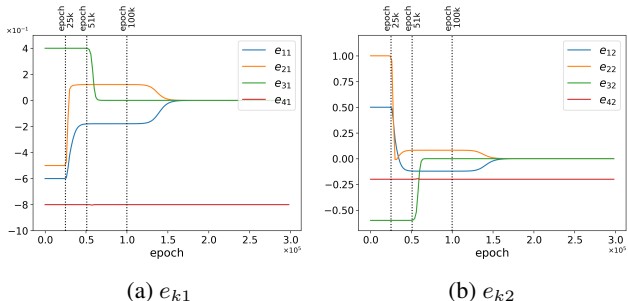

(a) $e_{k1}$          (b) $e_{k2}$

Figure 14: Errors at both components of output.

We find $\hat{\mathbf{v}}_i = (0.85, 0.51)$, which corresponds the angle of $1.03$ rad in Figure 12b, where the group 1 neuron gathered at epoch 25k. Knowing that $e_{11} = -0.6$ and $e_{21} = -0.5$ (which can be roughly read from Figure 14 at epoch 25k), we have:

$$\mathbf{d}_{1i}^{\hat{\mathbf{v}}_i} \cdot \hat{\mathbf{v}}_i = (e_{11}\mathbf{x}_1 + e_{21}\mathbf{x}_2) \cdot (0.85, 0.51) = (-0.32, -0.2) \cdot (0.85, 0.51) = -0.37 \neq 0.$$

We thus identified the escape neurons to be the ones that have zero-amplitude parameters at this saddle. Escaping from the first saddle was triggered by the movement of a group of small living neurons (group 1 neurons) exploiting the loss-decreasing path offered by such escape neurons.

*At the second saddle point, group 1 neurons have non-negligible amplitudes while group 2 neurons indicate the existence of zero-amplitude neurons in the nearby saddle.* In other words, if the parameters are placed exactly at the saddle point, the parameters of group 2 neurons should all be zero. Consequently, at the saddle point, $\frac{\partial \mathcal{L}}{\partial s_i} = 0$ for any $i$ that denotes a group 2 neuron and any tangential direction $\mathbf{v}_i$ (from Equation (6)). In this case, we can find a tangential direction $\tilde{\mathbf{v}}_i$ of group 2 neurons such that $\mathbf{d}_{1i}^{\tilde{\mathbf{v}}_i} \cdot \tilde{\mathbf{v}}_i \neq 0$. We specify that $\tilde{\mathbf{v}}_i = (-0.65, -0.76)$, corresponding to $3.22$ rad in Figure 12b, which is where group 2 neurons were attracted to before the second saddle point was escaped. Reading from Figure 14 that $e_{31} = 0.4$ and $e_{41} = -0.8$, we have:

$$\mathbf{d}_{1i}^{\tilde{\mathbf{v}}_i} \cdot \tilde{\mathbf{v}}_i = (e_{31}\mathbf{x}_3 + e_{41}\mathbf{x}_4) \cdot (-0.65, -0.76) = (-0.48, 0.56) \cdot (-0.65, -0.76) = -0.11 \neq 0.$$

This tells us that, the neurons with zero parameters are escape neurons at the second saddle. Escaping from the second saddle was then triggered by the movement of a group of small living neurons (group 2) exploiting the loss-decreasing path offered by such escape neurons.

*At the third saddle point, both groups of neurons have non-negligible amplitudes.* We will show that one of the group 2 neurons is approximately an escape neuron. It is not exactly an escape neuron since the parameter is not exactly at the saddle point but in the vicinity. The parameter trajectory of this specific neuron is denoted by thickened curves in Figure 12. In this part, $i$ denotes that single hidden neuron. The input weight of this neuron is $(-0.12, 0.63)$ (which can be computed from the readings in Figures 12b and 12c), and we pick one of its tangential derivatives $\check{\mathbf{v}}_i = (0.98, 0.19)$. We will show that $\mathbf{d}_{1i}^{\check{\mathbf{v}}_i} \cdot \check{\mathbf{v}}_i \neq 0$ and $\frac{\partial \mathcal{L}}{\partial \check{s}_i}$ (the tangential direction along $\check{\mathbf{v}}_i$) equals zero, making this neuron (approximately) an escape neuron.

We can roughly read from Figures 14a and 14b that $e_{11} = -0.18$, $e_{21} = 0.12$, $e_{12} = 0.12$, $e_{22} = 0.08$ at epoch 100k. With this, we can compute that

$$\mathbf{d}_{1i}^{\check{\mathbf{v}}_i} \cdot \check{\mathbf{v}}_i = (e_{11}\mathbf{x}_1 + e_{21}\mathbf{x}_2) \cdot (0.98, 0.19) = (0.18, 0.03) \cdot (0.98, 0.19) = 0.18 \neq 0.$$

$$\mathbf{d}_{2i}^{\check{\mathbf{v}}_i} \cdot \check{\mathbf{v}}_i = (e_{12}\mathbf{x}_1 + e_{22}\mathbf{x}_2) \cdot (0.98, 0.19) = (0.12, 0.02) \cdot (0.98, 0.19) = 0.12.$$

We can also read from Figures 12d and 12e to get $h_{1i} = 0.37$, $h_{2i} = -0.53$. These leads to

$$\frac{\partial \mathcal{L}}{\partial \check{s}_i} = h_{1i}\mathbf{d}_{1i}^{\check{\mathbf{v}}_i} \cdot \check{\mathbf{v}}_i + h_{2i}\mathbf{d}_{2i}^{\check{\mathbf{v}}_i} \cdot \check{\mathbf{v}}_i = 0.00.$$

With these computations, we show that the neuron we study is (approximately) an escape neuron. We can apply the same computation to any group 2 neurons at the third saddle point to find that all of them are (approximately) escape neurons. Eventually, their movement (splitting) caused the parameter to escape from this saddle.

*With the above, we show that the experiment is consistent with our Fact 4.7.* All saddle escapes are the results of parameter variation of escape neurons in the nearby saddle point.

*We can also analyze that the last stationary point at which the training process ended is a local minimum.* At the end of the training, all the input weights of the hidden neurons lie within the angle range of $(\frac{1}{4}\pi, 1\frac{1}{4}\pi)$, which means that $\mathbf{w}_i \cdot \mathbf{x}_4 < 0$, and $\{\mathbf{x}_4, y_4\}$ is the only training sample that has not been perfectly fitted yet, as shown in Figure 14. Thus, according to Equation (3),

$$\mathbf{d}_{ji}^{\mathbf{v}_i} = 0, \quad \text{for all } (i, j) \in I \times J \text{ and all tangential directions } \mathbf{v}_i. \tag{24}$$

This is due to the following: the first summation of Equation (3) is zero because all the error terms involved are zero. The second summation of Equation (3) is also zero because $\alpha^- = 0$. Moreover, $\mathbf{d}_{ji}^{\mathbf{v}_i} = 0$ disqualifies all the hidden neurons from being escape neurons, according to Definition 4.1.

## J APPLICABILITY OF FIGURE 4 TO SMALL BUT NON-VANISHING INITIALIZATION

Networks are rarely trained in the vanishing initialization regime. For one thing, the alignment of input weights might excessively reduce the number of kinks in the learned function and prevent the training from reaching global minima Boursier and Flammarion (2024), as is the case in Figure 3. In this section, we train the network initialized with law $\mathcal{N}\left(0, (8.75 \times 10^{-4})^2\right)$ (by rescaling the initialization weights in Section 4.1.1 without changing their direction). The process is visualized in Figure 15. The larger initialization scale enables the network to reach zero loss. With this example, we show how our insights derived from vanishing initialization can be applied to small but non-vanishing initialization.

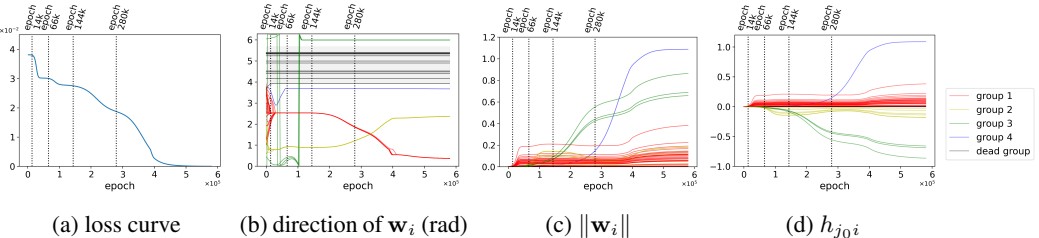

(a) loss curve      (b) direction of $\mathbf{w}_i$ (rad)      (c) $\|\mathbf{w}_i\|$      (d) $h_{j_0 i}$

Figure 15: *Evolution of all parameters during the training process with small but non-vanishing initialization scale.* There are more groups of neurons formed compared to Figure 3. The loss experiences repeated accelerations and decelerations, influenced by nearby saddle points. The accelerations of loss decreasing roughly coincide with the amplitude increase of groups of small living neurons.

The loss curve for this initialization scheme is shown in Figure 15a, where one can observe the decrease of loss accelerated and decelerated several times. This indicates that the network parameters were affected by several nearby saddle points. We mark the beginnings of the accelerations with dotted vertical lines in Figure 15a. The evolution of all the parameters is summarized in Figures 15b to 15d. Just as in Figure 3, we color-code the curves in Figures 15b to 15d based on the $\mathbf{w}_i$ grouping.

*We can observe that the accelerations of loss decreasing also roughly coincide with the amplitude increase of small living neurons. This means that the acceleration of loss decreasing in this regime also exploits the loss-decreasing path given by the escape neurons in the nearby saddle points, similar to the vanishing initialization regime.*

We also note two differences between small but non-vanishing and vanishing initialization. First, with the former, the loss curve no longer has extremely flat plateaus, and the small living neurons no longer appear extremely small in amplitude. This is because the gradient flow from small but non-vanishing initialization does not get as close to the stationary points as does the trajectory from vanishing initialization Jacot et al. (2022). Second, the former gives rise to four living neuron groups, corresponding to four kinks,[13] which suffices to reach zero loss; while the latter only creates two

---

[13]See Figure 17 for the emergence of the four kinks.

kinks and gets stuck at a high loss. The vanishing initialization limit induces a low-rank bias on the weights (Maennel et al., 2018; Jacot et al., 2022; Chistikov et al., 2024), which is alleviated by larger initialization scales to yield better expressivity with more kinks in the learned function (See Appendix L). A detailed account of this is in Appendix K. The investigation of the minimal initialization scale that results in global minima is a meaningful future direction.

## K  How Larger Initialization Scale Alleviates the Low-rank Bias

The low-rank bias induced by small initialization is due to the distinctive dynamical pattern induced by such initialization. The rotation speed of a neuron is, loosely speaking, $O(1)$; while the amplitude increase of a neuron is slower when its amplitude is smaller, as observed by Maennel et al. (2018); Boursier et al. (2022); Chistikov et al. (2024). Thus, given the initialization is sufficiently small, the neurons will have enough time to gather close to the attracting angles: The attracting angles are determined by the network function, which remains almost fixed (approximately as a zero function) during the early alignment phase when all the neurons have negligible amplitudes. As a result, the attracting angles are also almost fixed during this phase, and the neurons have enough time to approach them. This process causes a lower rank in the weight, which is preserved throughout training. With (slightly) larger initialization, the neurons' amplitude increase becomes much faster, and the network function will deviate from the zero function before the neurons rotate to the original attracting angles. The rapid variation of network function gives birth to new attracting angles and thus leads to a higher rank in the weight matrix, as shown in Figure 15. There, the group 4 neurons and group 3 neurons does not manage to merge with group 1 and group 2, respectively, before being steered away by newly generated attracted angles. However, such merging took place thoroughly with vanishing initialization (Figure 3).

## L  The Evolution of the Learned Function

In this section, we demonstrate the evolution of the learned function for both vanishing initialization and slightly larger initialization. We highlight the output function before and after the amplitude increase of the grouped living neurons.

The learned functions from vanishing initialization are shown in Figure 16. Remember that, to simulate vanishing initialization, we initialized all the parameters in the network independently with law $\mathcal{N}\left(0, (5 \times 10^{-6})^2\right)$. In Figure 16, we only draw the network function (using the blue line) with respect to the first component of the input, since the second component is always 1 in the training data to simulate the bias. Namely, in Figure 16, we draw the following function:

$$\tilde{y}(x) = \hat{y}\big((x,1)\big),\ x \in \mathbb{R}, \tag{25}$$

where $\hat{y}(\cdot)$ is the network function in Equation (1).

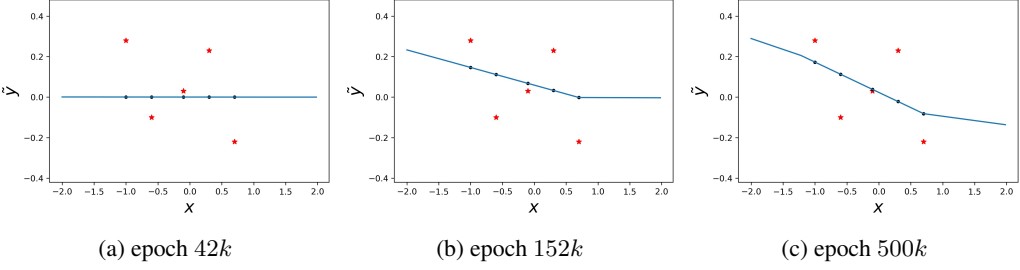

(a) epoch $42k$        (b) epoch $152k$        (c) epoch $500k$

Figure 16: The evolution of the learned function during training with an initialization standard deviation of $5 \times 10^{-6}$.

The five red stars are $(x_k, y_k)$'s, $x_k$'s being the first component of all the training samples, and $y_k$'s are the scalar target values. The five dots corresponds to $(x_k, \tilde{y}(x_k))$.

One can observe that, at epoch 42k, the network function is, approximately, a zero function. After the amplitude increase of the group 1 neurons, by epoch 152k, the network function has learned

one kink. The amplitude increase starting from roughly epoch 152k forms the second kink of the network function. Afterward, the network stops evolving, keeping the two kinks before the end of training (epoch 500k).

The evolution of the learned function from a slightly larger initialization scale is shown in Figure 17. In this experiment, the parameters are initialized independently with law $\mathcal{N}\left(0,\left(8.75 \times 10^{-4}\right)\right)$. We can observe that, with the amplitude increase of 4 groups of small living neurons at roughly epochs $14k$, $66k$, $144k$, and $280k$; there emerged 4 kinks. Notice, there should be 4 kinks in the learned function at the end. However, only 3 of them are shown, because the last one is located too far away from the input range we show in the image.

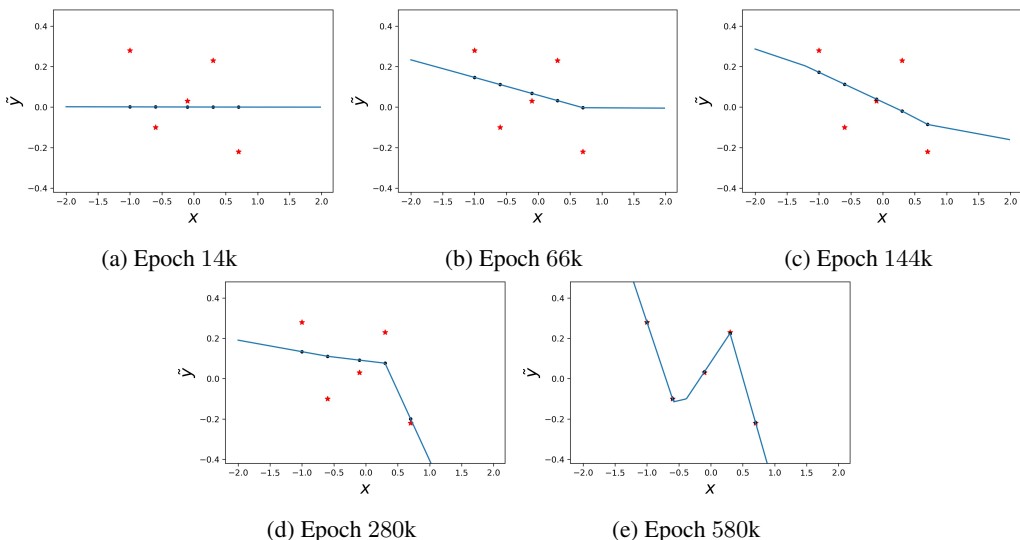

(a) Epoch 14k        (b) Epoch 66k        (c) Epoch 144k

(d) Epoch 280k        (e) Epoch 580k

Figure 17: Evolution of the learned function during the training starting with a larger initialization scale.

## M   PROOF OF PROPOSITION 4.8

### M.1   CONDITIONS FOR UNIT REPLICATION TO PRESERVE STATIONARITY

Let us denote the parameters of the stationary point before unit replication by $\overline{\mathbf{P}}$ and the parameters after by $\overline{\mathbf{P}}'$. Then, Definition 3.6 dictates that $\overline{\mathbf{P}}$ must satisfy:

$$\left.\frac{\partial \mathcal{L}(\mathbf{P})}{\partial h_{ji}}\right|_{\mathbf{P}=\overline{\mathbf{P}}} = 0, \ \forall\, j \in J, i \in I; \tag{26a}$$

$$\left.\frac{\partial \mathcal{L}(\mathbf{P})}{\partial r_{i}}\right|_{\mathbf{P}=\overline{\mathbf{P}}} = 0, \ \forall\, i \in I; \tag{26b}$$

$$\left.\frac{\partial \mathcal{L}(\mathbf{P})}{\partial s_{i}}\right|_{\mathbf{P}=\overline{\mathbf{P}}} \geq 0, \forall\, i \in I, \ \forall\, \text{tangential direction } \mathbf{v}_{i} \text{ of } \overline{\mathbf{w}}_{i}. \tag{26c}$$

It is easy to check that, after unit replication, the above still holds for the hidden neurons within the set $I \backslash \{i_0\}$, which are the hidden neurons untouched by unit replication, since the network function is not changed by this process. For the rest of the hidden neurons in $\overline{\mathbf{P}}'$, which are indexed by $i_0^l \in L$, we can also deduce whether they conform to the conditions for stationarity. We have the following:

$$\left.\frac{\partial \mathcal{L}(\mathbf{P})}{\partial h_{ji_0^l}}\right|_{\mathbf{P}=\overline{\mathbf{P}}'} = \mathbf{w}_{i_0^l} \cdot \mathbf{d}_{ji_0^l} = \beta_l \mathbf{w}_{i_0} \cdot \mathbf{d}_{ji_0} = \beta_l \left.\frac{\partial \mathcal{L}(\mathbf{P})}{\partial h_{ji_0}}\right|_{\mathbf{P}=\overline{\mathbf{P}}} = 0. \tag{27}$$

If $\|\mathbf{w}_{i_0^l}\| = \beta_l \|\mathbf{w}_{i_0}\| > 0$, then the newly generated hidden neurons have radial directions:

$$\left.\frac{\partial \mathcal{L}(\mathbf{P})}{\partial r_{i_0^l}}\right|_{\mathbf{P}=\overline{\mathbf{P}}'} = \sum_{j \in J} h_{ji_0^l} \mathbf{d}_{ji_0^l} \cdot \mathbf{u}_{i_0^l} = \gamma_l \sum_{j \in J} h_{ji_0} \mathbf{d}_{ji_0} \cdot \mathbf{u}_{i_0} = \gamma_l \underbrace{\left.\frac{\partial \mathcal{L}(\mathbf{P})}{\partial r_{i_0}}\right|_{\mathbf{P}=\overline{\mathbf{P}}}}_{(=0)} = 0. \qquad (28)$$

If $\|\mathbf{w}_{i_0^l}\| = \beta_l \|\mathbf{w}_{i_0}\| = 0$, then, by convention, the radial derivative $\left.\frac{\partial \mathcal{L}(\mathbf{P})}{\partial r_{i_0^l}}\right|_{\overline{\mathbf{P}}'}$ are set to zero without loss of rigor.

As for the tangential derivatives of the newly generated hidden neurons, we have:

$$\left.\frac{\partial \mathcal{L}(\mathbf{P})}{\partial s_{i_0^l}}\right|_{\mathbf{P}=\overline{\mathbf{P}}'} = \sum_{j \in J} h_{ji_0^l} \mathbf{d}_{ji_0^l} \cdot \mathbf{v}_{i_0^l} = \gamma_l \sum_{j \in J} h_{ji_0} \mathbf{d}_{ji_0} \cdot \mathbf{v}_{i_0} = \gamma_l \underbrace{\left.\frac{\partial \mathcal{L}(\mathbf{P})}{\partial s_{i_0}}\right|_{\mathbf{P}=\overline{\mathbf{P}}}}_{(\geq 0)}, \qquad (29)$$

where we take $\mathbf{v}_{i_0^l} = \mathbf{v}_{i_0}$. Namely, we are implying that any tangential direction of $\mathbf{w}_{i_0^l}$ must also be a tangential direction of $\mathbf{w}_{i_0}$. This is justified by the fact that $\beta_l > 0, \forall\, l \in L$.

With the above, we can infer $\left.\frac{\partial \mathcal{L}(\mathbf{P})}{\partial s_{i_0^l}}\right|_{\mathbf{P}=\overline{\mathbf{P}}'} \geq 0$ for all possible tangential direction $\mathbf{v}_{i_0^l}$'s if and only if either of the following is true:

1. $\frac{\partial \mathcal{L}(\overline{\mathbf{P}})}{\partial s_{i_0}} = 0, \forall$ tangential direction $\mathbf{v}_{i_0}$ of $\mathbf{w}_{i_0}$,
2. $\gamma_l \geq 0, \forall\, l \in L$;

which concludes the proof.

## M.2 CONDITIONS FOR UNIT REPLICATION TO PRESERVE TYPE-1 LOCAL MINIMALITY

The necessary and sufficient condition to preserve type-1 local minimality after unit replication is to avoid generating escape neurons and to preserve conditions for stationarity. Thus, we prove Proposition 4.8 by seeking necessary and sufficient conditions to achieve both these two goals. Let us denote the parameters of the stationary point before unit replication by $\overline{\mathbf{P}}$ and the parameters after by $\overline{\mathbf{P}}'$.

### M.2.1 AVOID GENERATING ESCAPE NEURONS

First, let us consider two unit replication schemes:

1. *Replicate a tangentially flat hidden neuron* Choose to replicate a hidden neuron $i_0$ satisfying $\left.\frac{\partial \mathcal{L}(\mathbf{P})}{\partial s_{i_0}}\right|_{\mathbf{P}=\overline{\mathbf{P}}} = 0$, for all tangential direction $\mathbf{v}_{i_0}$'s.
2. *Replicate with active propagation* Let $\gamma_l \neq 0, \forall\, l \in L$.

**Proposition M.1.** *Unit replication process conforming to either one of the two methods is sufficient and necessary for avoiding escape neurons.*

*Proof.*

**Sufficiency**:

Let us discuss the first way, replicating a tangentially flat hidden neuron. Since we have:

$$\left.\frac{\partial \mathcal{L}(\mathbf{P})}{\partial s_{i_0}}\right|_{\mathbf{P}=\overline{\mathbf{P}}} = \sum_{j \in J} \overline{h}_{ji_0} \mathbf{d}_{ji_0}^{\mathbf{v}_{i_0}} \cdot \mathbf{v}_{i_0} = 0, \quad \forall \text{ tangential direction } \mathbf{v}_{i_0} \text{ of } \overline{\mathbf{w}}_{i_0}. \qquad (30)$$

Based on the definition of type-1 local minima, we know that:

$$\mathbf{d}_{ji_0}^{\mathbf{v}_{i_0}} \cdot \mathbf{v}_{i_0} = 0, \quad \forall j \in J, \forall \text{ tangential direction } \mathbf{v}_{i_0} \text{ of } \overline{\mathbf{w}}_{i_0}. \qquad (31)$$

After unit replication, we have: For all $l \in L$, (1) $\mathbf{d}_{ji_0^l}^{\mathbf{v}_{i_0^l}} = \mathbf{d}_{ji_0}^{\mathbf{v}_{i_0}}$, $\forall\, j \in J$, for all tangential direction $\mathbf{v}_{i_0}$ of $\overline{\mathbf{w}}_{i_0}$; (2) $\overline{\mathbf{w}}_{i_0^l} = \beta_l \overline{\mathbf{w}}_{i_0}$ with $\beta_l > 0$, which means any tangential direction of $\overline{\mathbf{w}}_{i_0^l}$ are also a tangential direction of $\overline{\mathbf{w}}_{i_0}$. These, combined with Equation (31), leads to:

$$\mathbf{d}_{ji_0^l}^{\mathbf{v}_{i_0^l}} \cdot \mathbf{v}_{i_0^l} = 0, \ \ \forall\, l \in L, \forall\, j \in J, \forall \text{ tangential direction } \mathbf{v}_{i_0^l} \text{ of } \overline{\mathbf{w}}_{i_0^l}. \tag{32}$$

According to Definition 4.1, such $i_0^l$ cannot be escape neurons.

Next, let us discuss the second way, replicating with active propagation. If the replicated hidden neuron is a tangentially flat hidden neuron, then the discussion above will already guarantee that the unit replication process does not introduce escape neurons. Thus, we only need to focus on the following type of neurons:

$$\left.\frac{\partial \mathcal{L}(\mathbf{P})}{\partial s_{i_0}}\right|_{\mathbf{P}=\overline{\mathbf{P}}} = \sum_{j \in J} \overline{h}_{ji_0} \mathbf{d}_{ji_0}^{\mathbf{v}_{i_0}} \cdot \mathbf{v}_{i_0} > 0, \ \text{ for some tangential direction } \mathbf{v}_{i_0} \text{ of } \overline{\mathbf{w}}_{i_0}. \tag{33}$$

In this case, for any direction $\mathbf{v}_{i_0}$ with $\left.\frac{\partial \mathcal{L}(\mathbf{P})}{\partial s_{i_0}}\right|_{\mathbf{P}=\overline{\mathbf{P}}} = \sum_{j \in J} \overline{h}_{ji_0} \mathbf{d}_{ji_0}^{\mathbf{v}_{i_0}} \cdot \mathbf{v}_{i_0} > 0$, we have that, after unit replication, in the same direction ($\mathbf{v}_{i_0^l} = \mathbf{v}_{i_0}$):

$$\left.\frac{\partial \mathcal{L}(\mathbf{P})}{\partial s_{i_0^l}}\right|_{\mathbf{P}=\overline{\mathbf{P}}'} = \sum_{j \in J} \overline{h}_{ji_0^l} \mathbf{d}_{ji_0^l}^{\mathbf{v}_{i_0^l}} \cdot \mathbf{v}_{i_0^l} = \gamma_l \sum_{j \in J} \overline{h}_{ji_0} \mathbf{d}_{ji_0}^{\mathbf{v}_{i_0}} \cdot \mathbf{v}_{i_0} = \gamma_l \left.\frac{\partial \mathcal{L}(\mathbf{P})}{\partial s_{i_0}}\right|_{\mathbf{P}=\overline{\mathbf{P}}} \neq 0, \tag{34}$$

if $\gamma_l > 0, \forall\, l \in L$.

For any direction $\mathbf{v}_{i_0}$ with $\left.\frac{\partial \mathcal{L}(\mathbf{P})}{\partial s_{i_0}}\right|_{\mathbf{P}=\overline{\mathbf{P}}} = \sum_{j \in J} \overline{h}_{ji_0} \mathbf{d}_{ji_0}^{\mathbf{v}_{i_0}} \cdot \mathbf{v}_{i_0} = 0$, we know that:

$$\mathbf{d}_{ji_0}^{\mathbf{v}_{i_0}} \cdot \mathbf{v}_{i_0} = 0, \ \forall\, j \in J, \tag{35}$$

since the parameter before unit replication is a type-1 local minimum. After unit replication, we have that, in the same direction, in the same direction ($\mathbf{v}_{i_0^l} = \mathbf{v}_{i_0}$):

$$\left.\frac{\partial \mathcal{L}(\mathbf{P})}{\partial s_{i_0^l}}\right|_{\mathbf{P}=\overline{\mathbf{P}}'} = \sum_{j \in J} \overline{h}_{ji_0^l} \mathbf{d}_{ji_0^l}^{\mathbf{v}_{i_0^l}} \cdot \mathbf{v}_{i_0^l} = \gamma_l \sum_{j \in J} \overline{h}_{ji_0} \mathbf{d}_{ji_0}^{\mathbf{v}_{i_0}} \cdot \mathbf{v}_{i_0} = \gamma_l \left.\frac{\partial \mathcal{L}(\mathbf{P})}{\partial s_{i_0}}\right|_{\mathbf{P}=\overline{\mathbf{P}}} = 0, \tag{36}$$

and we also have:

$$\mathbf{d}_{ji_0^l}^{\mathbf{v}_{i_0^l}} \cdot \mathbf{v}_{i_0^l} = \mathbf{d}_{ji_0}^{\mathbf{v}_{i_0}} \cdot \mathbf{v}_{i_0} = 0, \ \forall\, j \in J, \forall\, l \in L. \tag{37}$$

This shows that replicating with active propagation also avoids introducing escape neurons.

Thus, a unit replication strategy conforming to at least one of the two methods mentioned above is sufficient to avoid generating escape neurons.

■ **Necessary**:

We conduct this proof by contradiction. If we replicate a hidden neuron $i_0$ that is not tangentially flat (described by Equation (33)) with $\gamma_l = 0$ for some $l \in L$, we can prove that there must exist escape neurons in the resulting parameters. For convenience, let us specify that $\hat{l}$ has $\gamma_{\hat{l}} = 0$.

Since Equation (33) holds, we can find a direction, denoted by a unit vector $\hat{\mathbf{v}}_{i_0}$, satisfying $\hat{\mathbf{v}}_{i_0} \cdot \overline{\mathbf{w}}_{i_0} = 0$ and $\left.\frac{\partial \mathcal{L}(\mathbf{P})}{\partial \hat{s}_{i_0}}\right|_{\mathbf{P}=\overline{\mathbf{P}}} = \sum_{j \in J} \overline{h}_{ji_0} \mathbf{d}_{ji_0}^{\mathbf{v}_{i_0}} \cdot \hat{\mathbf{v}}_{i_0} > 0$, this must mean that there exists $\hat{j}$ with:

$$\mathbf{d}_{\hat{j}i_0}^{\mathbf{v}_{i_0}} \cdot \hat{\mathbf{v}}_{i_0} \neq 0. \tag{38}$$

After unit replication, we have that, for the hidden neuron $i_0^{\hat{l}}$, the loss function's derivative with respect to its input weight in the direction of $\hat{\mathbf{v}}_{i_0^{\hat{l}}} = \hat{\mathbf{v}}_{i_0}$ is:

$$\left.\frac{\partial \mathcal{L}(\mathbf{P})}{\partial s_{i_0^{\hat{l}}}}\right|_{\mathbf{P}=\overline{\mathbf{P}}} = \sum_{j \in J} \overline{h}_{ji_0^{\hat{l}}} \mathbf{d}_{ji_0^{\hat{l}}}^{\mathbf{v}} \cdot \hat{\mathbf{v}}_{i_0^{\hat{l}}} = \gamma_{\hat{l}} \sum_{j \in J} \overline{h}_{ji_0} \mathbf{d}_{ji_0}^{\mathbf{v}_{i_0}} \cdot \hat{\mathbf{v}}_{i_0} = 0. \tag{39}$$

Additionally, we also have:

$$\mathbf{d}_{\hat{j}i_0^{\hat{l}}}^{\mathbf{v}} \cdot \hat{\mathbf{v}}_{i_0^{\hat{l}}} = \mathbf{d}_{\hat{j}i_0}^{\mathbf{v}_{i_0}} \cdot \hat{\mathbf{v}}_{i_0} \neq 0. \tag{40}$$

Equation (39) and Equation (40) signifies the existence of one escape neuron $i_0^{\hat{l}}$ after unit replication.

$\square$

### M.2.2 Preserving Stationarity Conditions

The first half of Proposition 4.8 is the sufficient and necessary conditions for preserving conditions for stationarity during unit replication.

### M.3 Putting Things Together

We want to avoid generating escape neurons while preserving conditions for stationarity at the same time. The necessary and sufficient condition for the occurrence of these two events should be the intersection of the necessary and sufficient conditions in Appendix M.2.1 and Proposition 4.8, which is exactly the necessary and sufficient condition of Proposition 4.8.

## N    Results Regarding Inactive Units

There are 2 types of inactive units, depicted in Figure 18:

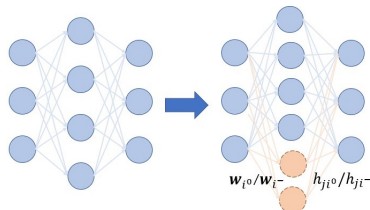

Figure 18: inactive units

*(2-1) Orthogonal units*[14]
Add more hidden neurons, which are indexed with $i^0 \in I^0$, where $h_{ji^0}$ is arbitrary, and $\mathbf{w}_{i^0} \cdot \mathbf{x}_k = 0$, for all $k \in K$.

*(2-2) Negative (positive) units.*
Suppose $\alpha^- = 0$. Add new hidden neurons $i^- \in I^-$, where $\mathbf{w}_{i^-} \cdot \mathbf{x}_k < 0$ for all $k \in K$, and $h_{ji^-}$ are arbitrary for all $j \in J$. This is what we call negative units. This embedding scheme can likewise be extrapolated for when $\alpha^+ = 0$, giving positive units.

### N.1    Results Regarding Orthogonal Units

This type of network embedding does not generally preserve conditions for stationarity or local minimality. Nonetheless, based on our definition of stationary points in Section 3.2 and our analysis of local minima in Section 4.1, one can carry out investigations into them on a case-by-case basis when analyzing a specific example.

Let us first discuss whether stationarity will be preserved under this network embedding scheme. Let us denote the parameters after adding orthogonal units by $\mathbf{P}'$. Since the newly added hidden neurons satisfy $\mathbf{w}_{i^0} \cdot \mathbf{x}_k = \mathbf{0}$ for all $k \in K$, we know that $\frac{\partial \mathcal{L}(\mathbf{P})}{\partial h_{ji^0}}\Big|_{\mathbf{P}=\mathbf{P}'} = 0$, and $\frac{\partial \mathcal{L}(\mathbf{P})}{\partial r_{i^0}}\Big|_{\mathbf{P}=\mathbf{P}'}$ equals zero. Thus, whether the tangential derivative preserves the condition in Definition 3.6 determines whether stationarity is preserved. For the tangential derivative, we have:

$$\frac{\partial \mathcal{L}(\mathbf{P})}{\partial s_{i^0}}\Big|_{\mathbf{P}=\mathbf{P}'} = \sum_{j \in J} h_{ji^0} \mathbf{d}_{ji^0}^{\mathbf{v}_{i^0}} \cdot \mathbf{v}_{i^0}. \tag{41}$$

Stationarity would require the above to be non-negative for all possible tangential direction $\mathbf{v}_{i^0}$, which cannot be guaranteed for general cases. Certain trivial cases where definite conclusions can be established are when the network has reached zero loss ($\mathbf{d}_{i^0}^{\mathbf{v}_{i^0}} = 0$ for all possible $\mathbf{v}_{i^0}$) or $h_{ji^0} = 0$ for all $j \in J$. In those cases, stationarity will be preserved.

We are not able to have general conclusions regarding whether the insertion of orthogonal units preserves type-1 local minimality as well since orthogonal units do not specify anything regarding whether they are escape neurons in general.

---

[14]Orthogonal units are not discussed in Fukumizu et al. (2019).

### N.2 RESULTS REGARDING NEGATIVE UNITS

**Proposition N.1.** *Suppose that $\alpha^- = 0$. Adding **negative units** preserves the stationarity of stationary points.*

*Proof.* Let us denote the parameters after the insertion of negative units by $\overline{\mathbf{P}}'$. It is easy to check that the parameters associated with the originally existing hidden neurons $i \in I$ still satisfy the conditions for stationarity after inserting the negative units, since the negative units do not change the network output $\mathbf{y}_k$ for all inputs $\mathbf{x}_k$. For the negative units $i^- \in I^-$, we observe the following.

$$\mathbf{d}_{ji^-} = \sum_{\substack{k: \\ \mathbf{w}_i^- \cdot \mathbf{x}_k > 0}} \alpha^+ e_{kj} \mathbf{x}_k = 0, \tag{42}$$

since this summation will be over an empty set of $k$. This leads to $\left.\frac{\partial \mathcal{L}(\mathbf{P})}{\partial h_{ji^-}}\right|_{\mathbf{P}=\overline{\mathbf{P}}'} = 0$ and $\left.\frac{\partial \mathcal{L}(\mathbf{P})}{\partial r_{i^-}}\right|_{\mathbf{P}=\overline{\mathbf{P}}'} = 0$ according to their formulae in Equations (4) and (5).

Moreover, we have $\mathbf{d}_{ji^-}^{\mathbf{v}_{i^-}} = \mathbf{d}_{ji^-}$ for tangential direction $\mathbf{v}_{i^-}$ of $\mathbf{w}_{i^-}$, since $\mathbf{w}_{i^-}$ is orthogonal to no $\mathbf{x}_k$'s. This shows that the tangential derivative of the loss function with respect to $\mathbf{w}_{i^-}$ is also zero, according to Equation (6).

$\square$

*Remark* N.2. We remind the readers that Fukumizu et al. (2019) has already proved that adding negative units will preserve the local minimality of the network.

## O RESULTS REGARDING INACTIVE PROPAGATION

Inactive propagation can be carried out as shown in Figure 19: add hidden neurons $i^\times \in I^\times$, where $\mathbf{w}_{i^\times}$ is taken arbitrarily and $h_{ji^\times} = 0, \forall j \in J$, as shown in Figure 19.

This type of network embedding also does not preserve stationarity or type-1 local minimality generally.

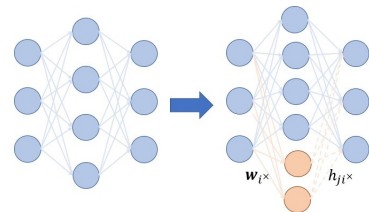

Figure 19: inactive propagation

We start by discussing the preservation of stationarity. Let us denote the parameters after adding units with inactive propagation by $\mathbf{P}'$. In this case, the added hidden neurons have $h_{ji^\times} = 0$ for all $j \in J$. As a result, according to Equation (5) and Equation (6), the radial derivative (which exists only when $\mathbf{w}_{i^\times} \neq \mathbf{0}$) and tangential derivatives (towards all tangential directions) must all be zero. However, the derivative with respect to output weight might not be zero:

$$\left.\frac{\partial \mathcal{L}(\mathbf{P})}{\partial h_{ji^\times}}\right|_{\mathbf{P}=\mathbf{P}'} = \mathbf{w}_{i^\times} \cdot \mathbf{d}_{ji^\times}. \tag{43}$$

Thus, the stationarity condition is not guaranteed to hold in general after adding inactive propagation units.

However, one may also notice that there are certain ways of enforcing stationarity: If we choose $\mathbf{w}_{i^\times} = \mathbf{0}$ or $\mathbf{w}_{i^\times} = \beta\mathbf{w}_i$ with $i \in I$, $\beta > 0$, then we equate Equation (43) to zero and preserves stationarity. The former also belongs to the case of orthogonal units, and the latter also belongs to the case of unit replication.

Then, we discuss whether type-1 local minimality is preserved. Preserving type-1 local minimality entails preserving stationarity and avoiding escape neurons, according to Theorem 4.2. We have discussed in the above that stationarity is not necessarily preserved in general. Moreover, regarding escape neurons, one can find that since an inactive propagation unit must have their tangential derivative being zero (since $h_{ji^\times} = 0$), without further strong restriction on the $\mathbf{d}_{ji^\times}$'s, the inactive propagation unit is highly likely an escape neuron.

## P   COMPARISON OF THEOREM 10 OF FUKUMIZU ET AL. (2019) AND PROPOSITION 4.8

Fukumizu et al. (2019) attempted to address the same question of whether local minimality is preserved by network embedding for our setup (which is also non-smooth) with its Theorem 10. It managed to find a specific scheme of unit replication that turns local minima into saddles, potentially contradicting Proposition 4.8. However, one can verify that the assumption in their theorem does not conform to our setting, which we explicate below.

Theorem 10 of Fukumizu et al. (2019) requires a constructed $F$ matrix not to be a zero matrix to construct positive and negative eigenvalues for the Hessian matrix at the resulting parameters after unit replication. In this way, it is shown that the parameters after unit replication constitute a strict saddle. Please refer to Fukumizu et al. (2019) for more detail.

However, it turns out that, for the empirical squared loss, which is a widely used loss, the F matrix is zero at a type-1 local minimum, which we prove in the rest of this section. Hence Theorem 10 of Fukumizu et al. (2019) cannot say anything about such a situation.

**Lemma P.1.** *Suppose a set of parameters $\overline{\mathbf{P}}$ is a stationary point in our setting. If an input $\overline{\mathbf{w}}_i$ is such that $\mathbf{x}_k \cdot \overline{\mathbf{w}}_i \neq 0$, for all $k \in K$ (an assumption also taken by Theorem 10 of Fukumizu et al. (2019)), we must have that $\mathbf{d}_{ji} = \mathbf{0}$ for all $j \in J$.*

*Proof.* If $\mathbf{w}_i \cdot \mathbf{x}_k \neq 0$ for all $k \in K$, then the network is continuously differentiable. Thus we must have $\frac{\partial \mathcal{L}(\overline{\mathbf{P}})}{\partial s_i} = 0$ for all $\mathbf{v}_i$'s. Otherwise, it will not be a stationary point.

Moreover, the fact that the stationary point we are investigating is a type-1 local minimum gives:

$$\mathbf{d}_{ji} \cdot \mathbf{v}_i = \mathbf{d}_{ji}^{\mathbf{v}_i} \cdot \mathbf{v}_i = 0, \text{ for any tangential direction } \mathbf{v}_i \text{ of } \overline{\mathbf{w}}_i. \tag{44}$$

Moreover, since the parameter before unit replication $\overline{\mathbf{P}}$ is a stationary point, we must have:

$$\left. \frac{\partial \mathcal{L}(\mathbf{P})}{\partial h_{ji}} \right|_{\mathbf{P}=\overline{\mathbf{P}}} = \overline{\mathbf{w}}_i \cdot \mathbf{d}_{ji} = 0. \tag{45}$$

Thus, $\mathbf{d}_{ji_0}$ must lies in the tangential space of $\overline{\mathbf{w}}_i$. If $\mathbf{d}_{ji_0} \neq \mathbf{0}$, then it must be parallel to some unit vector $\mathbf{v}_i$ satisfying $\mathbf{v}_i \cdot \overline{\mathbf{w}}_i = 0$, which contradicts Equation (44). $\qquad\square$

Then we investigate the $F$ matrix. It is helpful to recap the setting of Theorem 10 by Fukumizu et al. (2019). First, they studied the ReLU activation function, meaning $\alpha^+ = 1, \alpha^- = 0$. Moreover, they only discussed replicating a hidden neuron $i_0$ with

$$\overline{\mathbf{w}}_{i_0} \cdot \mathbf{x}_k \neq 0, \forall\, k \in K, \tag{46}$$

which we account for in the above helper lemma. From Fukumizu et al. (2019), we know that $F \in \mathbb{R}^{d \times |J|}$ has each of its column being:

$$F_{:,j} = \sum_{k \in K} e_{kj} \frac{\partial \rho(\mathbf{x}_k \cdot \mathbf{w}_{i_0})}{\partial \mathbf{w}_{i_0}} = \sum_{\substack{k: \\ \mathbf{x}_k \cdot \mathbf{w}_{i_0} > 0}} e_{kj} \mathbf{x}_k = \mathbf{d}_{ji_0}. \tag{47}$$

Notice that the above derivative is not hindered by the non-differentiability in the activation function thanks to Equation (46).

Remember that we have proved $\mathbf{d}_{ji_0} = \mathbf{0}$ for all $j \in J$, rendering F a zero matrix.

## Q   THE APPLICATIONS OF THE INSIGHT FOR NETWORK EMBEDDING

Our discussion on network embedding in Section 4.3 extends multiple previous results that did not consider non-differentiable cases.

Wu et al. (2019); Wang et al. (2024) proposed a training scheme where one neuron is split at an underfitting local minimum to create an escapable saddle point. Such a training scheme can lower

the cost of training since the training can start from a smaller (pretrained) network. Proposition 4.8 effectively shows how to implement such a scheme for scalar-output networks, where all the local minima are all of type-1: simply replicate a neuron whose input weight has non-zero tangential derivative and choose $\gamma_l \leq 0$. Such an insight might also be instrumental for multidimensional-output networks since type-2 local minima should be rare, as we discussed in Appendix F.2.1.

Our results regarding stationarity preservation also serve as constructive proof that one can always embed a stationary point of a narrower network into a wider network. In other words, the stationary points of wider networks "contain" those of the narrower networks. This is termed the *embedding principle*, discussed in Zhang et al. (2021) for smooth networks.

Section 4.3 also provides a perspective of understanding the merit of overparameterization for optimization. Stationary points and spurious local minima might worsen the performance of GD-based optimization. However, to our rescue, network embedding can cause some stationary points (local minima, resp.) to lose stationarity (local minimality, resp.). This might explain why wider networks tend to reach better training loss. It is also easy to check that embedding parameters that are not stationary points (local minima) will not result in stationary points (local minima). Similar phenomena are noted in Şimşek et al. (2021); Fukumizu et al. (2019); Zhang et al. (2021) for smooth networks. A meaningful next step is to quantify how manifolds of saddle points and local minima scale with network width (Şimşek et al., 2021) for ReLU-like networks.

