# OpenReview forum: "Loss Landscape of Shallow ReLU-like Neural Networks: Stationary Points, Saddle Escape, and Network Embedding"
_ICLR.cc/2025/Conference — ICLR 2025 Poster_

### Official Review · Reviewer_BZd8 · 2024-10-28

**Soundness:** 3
**Presentation:** 3
**Contribution:** 3
**Rating:** 8
**Confidence:** 4

**Summary:**

This paper characterizes the (possibly non-differentiable) stationary points of a one-hidden layer ReLU-like neural networks. By ReLU-like, I mean a piecewise linear activation with a single kink at x = 0. The authors specifically define the notion of a stationary point in their problem of interest, using the notion of directional derivatives. With the definition they are able to show that if a stationary point does not contain "escape neurons" - which is a specific type of neuron, the stationary point is a local minimum. This insight enables the authors to understand saddle-to-saddle dynamics, and network embedding principles with their own perspective, and extend existing results to non-differentiable settings.

**Strengths:**

The idea of considering a different coordinate system to exploit the fact that the nonlinearity is only non-differentiable at x = 0 is interesting.

**Weaknesses:**

1. I am not convinced on the definition of stationary points for this paper (Definition 3.3). Here, the authors define stationary points as points that have zero derivative with respect to second-layer weights and radial direction and nonnegative directional derivative with respect to tangent directions. The point I am not convinced is "why are the radial terms and tangential terms considered differently?" - is it because the radial term is differentiable but the tangential term is non-differentiable?

The definition of stationary points also felt more like a "local minimum" to me (this could possibly be wrong - but my understanding is that the defined saddle point has zero differential at differentiable parameters and local minimum for non-differentiable parameters).

Overall, the definition of stationary points in Definition 3.3 seems to be quite restricted.

2. Also, it would be good to explicitly mention that the notion of stationary points this paper discusses may be different from the notion from other papers in the literature, at the introduction and in the abstract. For example, there exists papers that completely characterize Clarke stationary points of a one-hidden layer ReLU network [1], so to make the contributions more concise, describing the notion of stationarity could help.

3. It is unclear to me how their approach is different from using one-sided directional derivatives (e.g. [2]). I think this paper also uses the notion of one-sided differential to mitigate non-differentiability (from the definition of radial and tangential derivative in pg 4)? Probably another major difference is radial/tangential derivatives - but I don't see the necessity of such concepts except when the authors define stationarity.

Overall, it is rather unclear how non-differentiability mitigates an analysis of saddle points, and how the authors actually overcame the difficulty with their idea.

4. Most part of main results are discussions rather than concise theorems or statements. Furthermore, while mentioning saddle points, it is confusing whether the authors are referring to "their notion of saddle points" or the notion of saddle points in the literature.

[1] Wang, Yifei, Jonathan Lacotte, and Mert Pilanci. "The hidden convex optimization landscape of two-layer relu neural networks: an exact characterization of the optimal solutions." arXiv preprint arXiv:2006.05900 (2020).

[2] Cheridito, Patrick, Arnulf Jentzen, and Florian Rossmannek. "Landscape analysis for shallow neural networks: complete classification of critical points for affine target functions." Journal of Nonlinear Science 32.5 (2022): 64.

**Questions:**

1. Addressing the weaknesses would be good. If I can see that the author's definition of saddle points makes sense, and how they are related to saddle points in other papers, I would raise my score.

2. Some clearer wording is needed: in Lemma 3.2, we have \delta(s_i) > 0 sufficiently small. Is there a clearer way to state this? In pg. 8, could you be more specific what larger initialization scale 8.75 x 10^-4 means? In pg.20, I am not sure if we can justify the existence of higher order directional derivatives just with continuity, because in general, continuity does not imply the existence of directional derivatives.

3. I got curious about the link between existing work and this paper.
 - It is proven that in [3], first-order methods avoids strict saddle points. The intuition in this paper is that strict saddle points have a descent direction where gradient descent falls down. I think this notion is similar to the notion of escape neuron, where we have a certain direction that the loss decreases. Do the two concepts have something in common, or are they intrinsically different?
 - In pg 6, you mention the notion of dead neurons. [2] also uses the term dead neurons - are they related?

[3] Lee, Jason D., et al. "First-order methods almost always avoid strict saddle points." Mathematical programming 176 (2019): 311-337.

---

> ### Author Response · Authors · 2024-11-23
>
> Thank you very much for your review! We appreciate that you consider our approach **interesting**.
>
> We made modifications to our script, highlighted in red, and marked with the orange-colored text 'BZd8' (your reviewer ID) to indicate changes based on your suggestions.
>
> You mentioned a few weaknesses, which we address below.
>
> ---
>
> >"1. I am not convinced on the definition of stationary points for this paper (Definition 3.3)... The point I am not convinced is 'why are the radial terms and tangential terms considered differently?' - is it because the radial term is differentiable but the tangential term is non-differentiable?"
>
> Yes, your understanding is correct. To facilitate the understanding of the difference, we also produce new plots in Appendix G. There, one can observe that the loss is always differentiable in the radial direction, which is not the case for the tangential direction (see the paragraph from line 1481).
>
> At a high level, Relu-like networks can be understood as several patches of linear networks pieced together. Changing the norm of the input weights (moving along the radial direction) of a hidden neuron will not change the activation pattern of that hidden neuron. Namely, the training inputs that activate the hidden neuron through the positive/negative leg of the activation function will not switch to another leg. So, moving along the radial direction will not cross the non-differentiable boundary between the pieces of linear networks. However, moving along one tangential direction and the opposite might result in different activation patterns of a hidden neuron and thus enter a different piece of linear network. Therefore, there is a fundamental difference between these two types of directions.
>
> >"The definition of stationary points also felt more like a "local minimum" to me (this could possibly be wrong - but...)."
>
> *We point out that not all stationary points by our definition are local minima*. Our definition of stationary points specifies that all the first-order directional derivatives should be non-negative, but there could be loss-decreasing direction resulting from higher-order directional derivatives. A good example of this is the origin of function $f(x) =x^3$, which is "flat" in terms of first-order directional derivatives (so it is a stationary point by our definition), but has decreasing directions given by third-order terms in Taylor expansion. A less toy, non-differentiable example is provided in Appendix C2.
>
> >"...my understanding is that the defined saddle point has zero differential at differentiable parameters and local minimum for non-differentiable parameters"
>
> Your understanding of saddle points is correct. Non-differentiable directions must bend upward at saddle points. Otherwise, GD will slide away without significantly slowing down, disqualifying the point as a stationary point.
>
> >"Overall, the definition of stationary points in Definition 3.3 seems to be quite restricted."
>
> We believe our definition of stationary points is not restricted because *all the conventional stationary points of smooth networks are still stationary points by our definition. In other words, our Definition 3.3 extends the conventional notion of stationarity.* The conventional stationary points should have zero directional derivatives toward all directions, which qualifies them as stationary points by our definition (which states that stationary points should have non-negative directional derivatives toward all directions).
>
> ---
> > "2. Also, it would be good to explicitly mention that the notion of stationary points this paper discusses may be different from the notion from other papers in the literature, at the introduction and in the abstract. For example, there exist papers that completely characterize Clarke stationary points of a one-hidden layer ReLU network [1], so to make the contributions more concise, describing the notion of stationarity could help."
>
> Thank you very much for this suggestion. We have added one sentence in the abstract and one paragraph in the introduction to clarify the difference between our notion of stationarity and other previous notions. We have also included the reference you suggested. Please see line 18 and line 73 for more details.

---

> > ### Author Response · Authors · 2024-11-23
> >
> > >"3. It is unclear to me how their approach is different from using one-sided directional derivatives (e.g. [2]). I think this paper also uses the notion of one-sided differential to mitigate non-differentiability (from the definition of radial and tangential derivative in pg 4)?"
> >
> > We are using one-sided directional derivatives in a specially designed coordinate system so that we can account for non-smoothness without compromising rigorousness.
> >
> > In below, we show how our method handles stationarity better than the method in [2]. In our answer to the next weakness, we will demonstrate why our method is necessary for Taylor expansion.
> >
> > According to Definition 2.1 and Equation 2.3 of [2], their definition of stationarity is the following: The right-hand derivative on any canonical coordinate axis (whose direction corresponds to one-hot unit vector, $\mathbf e_k$ in [2]) is zero. Such a definition might fail due to two problems (which our definition overcomes).
> >
> > First, some points that should be considered as stationary points have non-zero directional derivatives along all canonical coordinate axes. One example is the origin of $f(x) =\vert x \vert$. We solve this problem by requiring the directional derivatives to be non-negative rather than zero.
> >
> > Second, using the directional derivative along the canonical coordinate axes might fail to capture negative first-order directional derivatives within the orthants (the area between the axes) due to the nonsmoothness within the orthants. One example is the origin of $f(x,y) =\vert x-y \vert - \vert x+y \vert$, which has zero left and right hand directional derivatives along both axes of $x$ and $y$ (so it is a stationary point by the definition from [2]). But it has negative first-order directional derivatives into the first quadrant, and thus should not qualify as a stationary point. To solve this problem, *we need to align our axes with the non-differentiable edges on the function surface so that the function surface over each orthant is always differentiable.* As a result, if the derivatives on the aligned axes are zero (non-negative), then the directional derivatives into the orthant between the axes must also be zero (non-negative), which is guaranteed by the definition of directional derivatives on smooth functions. In this way, stationarity is rigorously captured.
> >
> > We make a new visualization in Appendix G to clarify how we align our coordinate axes with the non-differentiable part of the loss landscape by using the radial and tangential directions, and why it is necessary.
> >
> >
> > >"Probably another major difference is radial/tangential derivatives - but I don't see the necessity of such concepts except when the authors define stationarity. Overall, it is rather unclear how non-differentiability mitigates an analysis of saddle points, and how the authors actually overcame the difficulty with their idea."
> >
> > To analyze saddle points, we need to perform Taylor expansion. Taylor expansion requires differentiability. As a result, the non-differentiability of ReLU-like networks prohibits us from directly applying Taylor expansion.
> >
> > To overcome this, we introduce the coordinate system containing axes corresponding to the output weights and the radial/tangential directions of the input weights. The axes are aligned with the non-differentiable hyperplanes in the parameter space so that the loss within every orthant (the area between coordinate axes) is always differentiable. (Please refer to the newly added Appendix G for a clearer visualization.) In this way, we can use Taylor expansion based on the derivatives (of any orders) along the aligned coordinate axes to accurately characterize the function within the orthants, as it would be as if we are studying a differentiable function, but limiting the scope to an orthant. Note that this is not possible if we directly use the canonical coordinate axes, since the loss in the orthants there is normally non-differentiable, which is visualized in the newly added Figure 9(a).
> >
> > We believe this discussion clarifies our presentation significantly! Thank you very much for this question!

---

> ### Author Response · Authors · 2024-11-23
>
> >"4. Most part of main results are discussions rather than concise theorems or statements."
>
> We believe you are referring to the parts where we introduce the numerical experiment (Section 4.1.1) and contextualize Figure 4 (Section 4.2). These parts are relevant for presenting the implications of our main results, thus we kept them in the main text and actually deferred several theorems/lemmas/propositions regarding the rarity of local maxima and network embedding entirely to the appendix (Appendix D,M,O). Nevertheless, we agree that such an arrangement made our presentation style seem too heuristic. Thus, we make the following modifications to the script.
>
> * To disambiguate our result that precludes other saddle types encountered during training, we compose Corollary 4.7 (see line 383) which replaces the originally loose discussions.
>
> * As it caused confusion for other reviewers, we specify that Fact 4.6 and Fact 4.8 are rigorously proved results from a previous paper and our paper, respectively, rather than empirical observations (see footnote 4 and line 403).
>
> * We defer the discussion regarding training dynamics from small but non-vanishing (previously called "finitely small") initialization to Appendix J. It meaningfully extends our discussion but could seem digressive.
>
> >"Furthermore, while mentioning saddle points, it is confusing whether the authors are referring to 'their notion of saddle points' or the notion of saddle points in the literature."
>
> We believe such confusion is due to our discussion of training dynamics (Section 4.2), where we mentioned "saddles" in the "saddle-to-saddle dynamics" studied by [3-5] several times. Their notion of saddle points coincides with ours entirely but they failed to formalize such a notion. Defining such a notion is one of our contributions, which we explicate below.
>
>
> [3-5] investigated the training dynamics of ReLU networks and found that there could exist points around which GD lingers extremely long, they loosely refer to these points as saddle points without rigorously defining the potentially non-differentiable saddle points. (Please refer to Assumption 3 of [5] and its discussion, which summarizes the saddle points studied by [3,4], for further details.) As a side note, they referred to such GD-slowing points as saddle points since the slowdown effect of such points on GD is indicative of a small "effective" gradient in the vicinity, reminiscent of the morphology of the conventional smooth saddle points.
>
> We formally identify such GD-slowing points: First, GD being trapped near a point for a long time is equivalent to saying that there does not exist any directions with negative first-order directional derivative around that point, which is our definition of a stationary point. Second, if such a stationary point is not a local minimum or maximum, we call it a saddle point. We demonstrate the effectiveness of our definition in Section 3.2 and Appendix C. We also add a paragraph to clarify the connection between our notion of stationary point and the one in previous works [3-5] in our introduction (See line 73).
>
> It is also noteworthy that the term "saddle-to-saddle" that [3-5] invoked (without a formal definition of saddle point) was first coined for deep linear networks [6], where the loss is smooth and the saddles are well-defined. [3-5] slightly abused the term for the nonsmooth case based on phenomenological similarity.
>
> ---
> You also asked the following questions.
> >"1. Addressing the weaknesses would be good. If I can see that the author's definition of saddle points makes sense, and how they are related to saddle points in other papers, I would raise my score."
>
> We believe our previous answers have addressed this question.
>
> ---
> >"2. Some clearer wording is needed: in Lemma 3.2, we have \delta(s_i) > 0 sufficiently small. Is there a clearer way to state this?"
>
> We have changed the following
>
>
> >>$\displaystyle\\mathbf d_{ji}^{\\mathbf v_i}=\\sum_{\\substack{k:\\\\ (\\mathbf w_i+\\Delta s_i\\mathbf v_i)\\cdot\\mathbf x_k>0}}
> \\alpha^+e_{kj}\\mathbf x_k + \\sum_{\\substack{k:\\\\ (\\mathbf w_i+\\Delta s_i\\mathbf v_i)\\cdot\\mathbf x_k<0}}
> \\alpha^-e_{kj}\\mathbf x_k$, where $\\Delta s_i > 0$ is sufficiently small.
>
> to
>
> >>$\displaystyle\\mathbf d_{ji}^{\\mathbf v_i}=\\lim_{\\Delta s_i \\searrow0^+}\\left(\\sum_{\\substack{k:\\\\ (\\mathbf w_i+\\Delta s_i\\mathbf v_i)\\cdot\\mathbf x_k>0}}
> \\alpha^+e_{kj}\\mathbf x_k + \\sum_{\\substack{k:\\\\ (\\mathbf w_i+\\Delta s_i\\mathbf v_i)\\cdot\\mathbf x_k<0}}
> \\alpha^-e_{kj}\\mathbf x_k\\right)$
>
> >" In pg. 8, could you be more specific what larger initialization scale 8.75 x 10^-4 means?"
>
> It means that we initialize the parameter with $\mathcal{N}(0,(8.75 \times 10^{-4})^2)$. We have changed our script accordingly (see line 1696). This part of discussion is now deferred to Appendix J since it slightly deviates from our main theory.

---

> > ### Author Response · Authors · 2024-11-23
> >
> > >"In pg.20, I am not sure if we can justify the existence of higher order directional derivatives just with continuity, because in general, continuity does not imply the existence of directional derivatives."
> >
> > We have replaced that sentence with the following:
> >
> > >>The loss of ReLU-like networks can be understood as numerous patches of linear network loss pieced together. Thus, moving along a fixed direction $\Delta$ locally exploits the loss of a linear network, which is a polynomial with respect to all the parameters. As a result, directional derivatives of any order along $\Delta$ are always definable.
> >
> > ---
> > >3."It is proven that in [3], first-order methods avoid strict saddle points. The intuition in this paper is that strict saddle points have a descent direction where gradient descent falls down. I think this notion is similar to the notion of escape neuron, where we have a certain direction that the loss decreases. Do the two concepts have something in common, or are they intrinsically different?"
> >
> >
> > Yes, they are closely connected. In the proof of Corollary 4.4, we show that moving the parameters of an escape neuron in a certain way reveals a second-order loss-decreasing direction. Thus, the potentially non-differentiable saddle points in our problem are analogous to the smooth strict saddles. (Because moving along the eigendirection of a smooth strict saddle with negative eigenvalues reveals a second-order decreasing direction.) Hence, first-order methods seem to always escape from the saddles in our case.
> >
> > Just in case, by second-order loss-decreasing direction, I mean a direction along which the first-order directional derivative is zero, but the second-order directional derivative is negative.
> >
> >
> > >"In pg 6, you mention the notion of dead neurons. [2] also uses the term dead neurons - are they related?"
> >
> >
> > Yes, they are the same concept.
> >
> > Thank you very much for your in-depth and comprehensive review! Your valuable opinions have significantly improved the quality of our script. We believe we have addressed all your concerns and hope you consider increasing your score or sub-ratings.
> >
> >
> >
> >
> >
> >
> > **References**
> >
> >
> > [1] Wang, Y., Lacotte, J., and Pilanci, M. The hidden convex optimization landscape of regularized two-layer ReLU networks: An exact characterization of optimal solutions, ICLR '22.
> >
> >
> > [2] Cheridito, P., Jentzen, A., and Rossmannek, F. Landscape analysis for shallow neural networks: Complete classification of critical points for affine target functions, Journal of Nonlinear Science, 2022.
> >
> >
> > [3] Boursier, E., Pillaud-Vivien, L., and Flammarion, N. Gradient flow dynamics of shallow ReLU networks for square loss and orthogonal inputs, NeurIPS'22.
> >
> >
> > [4] Chistikov, D., Englert, M., and Lazic, R. Learning a neuron by a shallow ReLU network: Dynamics and implicit bias for correlated inputs, NeurIPS '23.
> >
> >
> > [5] Kumar, A., and Haupt, J. Directional convergence near small initializations and saddles in two-homogeneous neural networks, Transactions on Machine Learning Research, 2024.
> >
> >
> > [6] Jacot, A., Ged, F., Şimşek, B., Hongler, C., and Gabriel, F. Saddle-to-saddle dynamics in deep linear networks: Small initialization training, symmetry, and sparsity. 2022.

---

> ### Comment · Reviewer_BZd8 · 2024-11-24
>
> I'd like to thank the authors very much for their comprehensive response, as well as accommodating most of my feedback. I wanted to ask you a few final remarks before making my final decision:
>
> 1. I now think I understand the intuition behind considering the differentiable and non-differentiable parts differently: in "saddle-to- saddle" dynamics when the loss function stagnates, the non-differentiable part of the loss function should bend upwards, as if not, gradient descent will escape the point "even though the subdifferential contains a 0". I now understand why Clarke subdifferentials cannot fully grasp the phenomenon. I think this is a very good intuition to understand saddle-to-saddle dynamics, and in fact shows that saddle-to-saddle dynamics is a misnomer - because we are not moving from a saddle point to another, rather, we are moving from a "specific point in the loss landscape where the loss function stagnates (which is not a saddle point in classical notion)" to another.
>
> One thing I would like to know (which I believe makes the paper much stronger) is rigorously checking if this is always the case (i.e. for non-differentiable cases, if the loss is not bent upwards, gradient descent escapes too easily) - at least in the simplest case, i.e. for scalar functions. However, without such rigorous analysis, I am now quite convinced that the notion of the author's stationary point is an important object in understanding the saddle-to-saddle dynamics.
>
> 2. Some minor confusions were:
>
> 1) In Line 73-74, the sentence "where first-order directional derivatives toward all directions are nonnegative" could be made clearer if we write "first-order one-sided directional derivatives".
>  2) In Line 1098 - 1101, I think the sentences are vague. Also, I am curious if the authors don't have to mention anything about coordinate alignment, which (if I understood correctly) is important in guaranteeing the existence of derivatives.
>  3) I didn't understand why "all higher order derivatives are null" in Line 1105-1106.
>  4) I see that you changed the definition of $d_{ji}^{v_i}$ in equation (3). However, I am confused: is this same as just saying
> $$
> \sum_{k: w_i \cdot x_k \geq 0} \alpha^{+}e_{kj}x_k + \sum_{k: w_i \cdot x_k \leq 0} \alpha^{-}e_{kj}x_k,
> $$
> sending $\Delta s_i \rightarrow 0$. Or something similar?
>
> I would like to thank the authors again for their sincere response. It helped me a lot in understanding the value of the work.

---

> > ### Comment · Reviewer_BZd8 · 2024-11-25
> >
> > 3. As far as I know, stochastic gradient descent converges to Clarke stationary points with high probability [1]. Is interpreting your results as "refining the limit point of sgd or variants with Clarke stationary points with upper bending on nondifferentiable parts" the right way? If so, is there a justification about the fact?
> >
> > [1] Davis, Damek, et al. "Stochastic subgradient method converges on tame functions." Foundations of computational mathematics 20.1 (2020): 119-154.

---

> > > ### Author Response · Authors · 2024-11-26
> > > **Addressing Your Further Questions**
> > >
> > > Thank you very much for your prompt follow-up! We’re glad that our response helped clarify our contribution and provided a better understanding of the content and value of our paper. Below, we address your further questions.
> > >
> > >
> > > ---
> > > 1.
> > > > "One thing I would like to know (which I believe makes the paper much stronger) is rigorously checking if this is always the case (i.e. for non-differentiable cases, if the loss is not bent upwards, gradient descent escapes too easily) - at least in the simplest case, i.e. for scalar functions. However, without such rigorous analysis, I am now quite convinced that the notion of the author's stationary point is an important object in understanding the saddle-to-saddle dynamics."
> > >
> > >
> > > We prove the following Proposition (now in Appendix C.1) to state that the escape time for a kink that does not bend upwards on both sides is not like that for a saddle point, showing the "easiness" of escaping from such points. We base our statement on a random function to simulate that the loss landscape of neuron networks is usually also (a realization) of a random function, with randomness coming from the dataset. The function to the left and right of the kink is $C^\infty$, as is the case for the loss of ReLU-like network.
> > >
> > > >> Proposition C.1: Consider a random function $f(x) = \sum_{n = 1}^N\alpha_n x^n \mathbb{1}\_{\{x<0\}}+ \sum_{n = 1}^N\beta_n x^n \mathbb{1}\_{\{x\geq0\}}$, where $x\in \mathbb{R}$,  $N\in \mathbb{N}$, and the random coefficients $\{\boldsymbol\alpha,\boldsymbol\beta\}\triangleq\{\alpha\_1,\cdots \alpha\_N,\beta\_1,\cdots,\beta\_N\}$  are drawn independently from a distribution that is absolutely continuous with respect to Lebesgue measure.
> > > Denote left-hand and right-hand derivatives by $f\_-^{'}(\\cdot)$ and $f\_+^{'}(\\cdot)$.
> > > We study the GD process $x\_{t+1} = x\_{t} - \eta f\_+^{'}(x\_t)$, where $\eta > 0$ is the step size. Suppose that $f\_+^{'}(0) < 0$ or $f\_-^{'}(0)<0$.
> > > Then, with probability $1$, the following holds.
> > > There exists an interval containing the origin $\chi(\boldsymbol \alpha, \boldsymbol \beta )\triangleq \left[ a(\boldsymbol \alpha, \boldsymbol\beta), b(\boldsymbol \alpha, \boldsymbol\beta)\right]$, where $a(\boldsymbol \alpha, \boldsymbol\beta)<0$ and $b(\boldsymbol \alpha, \boldsymbol\beta)>0$, and the time for escaping from this interval can be upper bounded by $\infty>\tilde t (\eta,\boldsymbol\alpha, \boldsymbol\beta) > 0$, which means if $x\_t \in \chi$, then there exist $0<t^\prime\leq \tilde{t}$ such that $x\_{t+t^\prime}\notin\mathbf{\chi}$.
> > >
> > >
> > > The above proposition shows that if the left-hand or right-hand derivative at the origin is negative, then the escape time from the origin $\tilde t$ only concerns the realization of the function (determined by $\boldsymbol \alpha, \boldsymbol\beta$) and the learning rate $\eta$, without involving how close GD gets to the origin. By contrast, saddle points, whether differentiable or not, trap GD or gradient flow for a longer time if the trajectory of GD or gradient flow reaches closer to them [1,2,3]. More concretely, if the origin $x = 0$ is a saddle point, then the upper bound for the escape time should be $\tilde t (\eta,\boldsymbol\alpha, \boldsymbol\beta,\hat d)$, where $\hat d\triangleq \inf_{t^\prime \geq 0}\vert x_{t+t^\prime}\vert$ is the shortest distance from the trajectory following $x_t$ to the origin $0$, and it must be part of the upper bound $\tilde t$. For example, suppose $x=0$ is a conventional smooth saddle point, then, if $\hat d \searrow 0$, we have $\tilde t\nearrow\infty$ (since gradient is smaller if you get closer to the saddle); and if $\hat d = 0$, we have $\tilde t=\infty$ (GD is permanently stuck in this case).
> > >
> > >
> > > For more details, please refer to the newly added Appendix C.1.

---

> > > > ### Author Response · Authors · 2024-11-26
> > > > **Addressing Your Further Questions**
> > > >
> > > > 2. You mentioned the following "minor confusions"
> > > >
> > > >
> > > > >"1. In Line 73-74, the sentence "where first-order directional derivatives toward all directions are nonnegative" could be made clearer if we write "first-order one-sided directional derivatives"."
> > > >
> > > >
> > > > Thank you for this comment! We have made this change to Lines 73-74 and also to other places to avoid confusion.
> > > >
> > > >
> > > > >"2. In Line 1098 - 1101, I think the sentences are vague."
> > > >
> > > >
> > > > We have changed that paragraph (now in Line 1165-1170) to the following.
> > > > >> As a side note, the loss of ReLU-like networks can be understood as numerous patches of linear network loss pieced together. Thus, moving along a fixed direction $\Delta$ **from a given point** locally exploits the loss of a linear network, which is a polynomial with respect to all the parameters. **In other words, the loss landscape is piecewise differentiable ($C^\infty$)**. As a result, one-sided directional derivatives of any orders along $\Delta$ **(whose computations are similar to Equation (8) and (9))** are always definable.
> > > >
> > > >
> > > > We mark the modifications in boldfaced letters above.
> > > >
> > > >
> > > > Equation (8) in our paper is:
> > > >
> > > >
> > > > $$\partial_\Delta\mathcal{L}(\mathbf{P})
> > > >     :=\lim_{\epsilon\to 0+}\frac{\mathcal{L}(\mathbf{P}+\epsilon\Delta)-\mathcal{L}(\mathbf{P})}{\epsilon},$$
> > > > and Equation (9) is:
> > > > $$
> > > > \partial_\Delta^2\mathcal{L}(\overline{\mathbf{P}})
> > > >     :=\lim_{\epsilon\to 0+}\frac{\partial_\Delta\mathcal{L}(\overline{\mathbf{P}}+\epsilon\Delta)-\partial_\Delta\mathcal{L}(\overline{\mathbf{P}})}{\epsilon}\cdots
> > > > $$
> > > >
> > > >
> > > > We hope to clarify our discussion by showing these two equations, which are exact computations for the first and second order one-sided directional derivatives.
> > > >
> > > >
> > > > >"Also, I am curious if the authors don't have to mention anything about coordinate alignment, which (if I understood correctly) is important in guaranteeing the existence of derivatives."
> > > >
> > > >
> > > > Notice that the one-sided directional derivatives can always be defined by performing computations like Equation (8) and (9), and do not require our aligned coordinate axes. The aligned coordinate axes are for guaranteeing that the function within each orthant is differentiable so that (1) stationarity can be identified with the first-order one-sided derivatives computed along the axes, (2) Taylor expansion can be performed within the orthant. Please refer to our answer to the third weakness that you raised before. We have a more detailed discussion there.
> > > >
> > > >
> > > > > "3. I didn't understand why "all higher order derivatives are null" in Line 1105-1106."
> > > >
> > > >
> > > > We have now added footnote 9 to explain this: All one-sided directional derivatives of the fourth order are constants, as shown in Appendix F.1.3.
> > > >
> > > >
> > > > At a high-level, roughly speaking, if we look into the composition of the loss function as introduced in Section 2, we see that the input weight $W$ multiplied with the output weight $H$ (with a positively homogeneous activation function $\rho(\cdot)$ between) gives rise to order $2$. Then, the empirical squared loss raises the order to $2^2 = 4$. Hence, the system has non-zero differentials of at most $4$ orders.
> > > >
> > > >
> > > > > "4. I see that you changed the definition of $\mathbf d_{ji}^{\mathbf v_i}$ in equation (3). However, I am confused: is this same as just saying:
> > > > $$\sum_{k: w_i: x_x \geq 0} \alpha^{+} e_{k j} x_k+\sum_{k: w_i: x_k \leq 0} \alpha^{-} e_{k j} x_k$$
> > > > sending $\Delta s_i\to 0$, or something similar?"
> > > >
> > > >
> > > > Yes. Your interpretation is mostly correct. Nonetheless, we stress that it is important that we take $\Delta s_i\searrow 0^+$ rather than just $\Delta s_i\to 0$, as $\mathbf d_{ji}^{\mathbf v_i}$ might be different if we take $\Delta s_i$ to zero from different sides. Taking $\Delta s_i\searrow 0^+$ is how we ensure we study the stationarity and the Taylor expansion within one single orthant, where the function is smooth, as discussed in lines 1500-1507 and visualized in Figure 9 (b).

---

> > > > > ### Author Response · Authors · 2024-11-26
> > > > > **Addressing Your Further Questions**
> > > > >
> > > > > 3.
> > > > > > "As far as I know, stochastic gradient descent converges to Clarke stationary points with high probability [1]. Is interpreting your results as ‘refining the limit point of sgd or variants with Clarke stationary points with upper bending on nondifferentiable parts’ the right way? If so, is there a justification about the fact?
> > > > > [1] Davis, Damek, et al. "Stochastic subgradient method converges on tame functions." Foundations of computational mathematics 20.1 (2020): 119-154."
> > > > >
> > > > >
> > > > > Yes, the interpretation is correct and there can be theoretical justifications. The refinement that we brought into the picture is two-fold:
> > > > >
> > > > >
> > > > > (1) Our notion of stationarity not only studied the limiting points of GD, but also the points where GD is slowed down but eventually escapes.
> > > > >
> > > > >
> > > > > (2) (Davis et al., 2020) proved that SGD converges to Clarke stationary points with high probability. But, we would not expect that SGD converges to Clarke stationary point like the origin of $-\vert x \vert$. Namely, there is a specific subset of Clarke stationary points that SGD converges to. Meanwhile, we can show that, if a point is a stationary point by our definition, then such a point is also a Clarke stationarity point (please see our newly added Appendix C.2 for a proof). Hence, our notion of stationarity is a refined subset of Clarke stationarity and can be more relevant for understanding the behavior of GD/SGD. Particularly, your interpretation that "our stationary points are Clarke stationary points with upward bending" is correct, as is stated and proved in our Fact C.4 in the newly added Appendix C.2. Thank you very much for this question which we believe enriches our discussion!
> > > > >
> > > > > ---
> > > > > Thank you again for your constructive comments! Should you have any further doubts or concerns, please feel free to reach out to us. We would be pleased to address them.
> > > > >
> > > > >
> > > > > **References**
> > > > >
> > > > >
> > > > > [1] Maennel, H., Bousquet, O., and Gelly, S. Gradient descent quantizes ReLU network features, 2018.
> > > > >
> > > > >
> > > > > [2] Boursier, E., Pillaud-Vivien, L., and Flammarion, N. Gradient flow dynamics of shallow ReLU networks for square loss and orthogonal inputs, NeurIPS'22.
> > > > >
> > > > >
> > > > > [3] Chistikov, D., Englert, M., and Lazic, R. Learning a neuron by a shallow ReLU network: Dynamics and implicit bias for correlated inputs, NeurIPS '23.

---

> > > > > > ### Comment · Reviewer_BZd8 · 2024-11-26
> > > > > >
> > > > > > Thank you for a clear deep response. I am now surely convinced that the authors made a good contribution in understanding the saddle-to-saddle dynamics, and suggest acceptance.

---

> > > > > > > ### Author Response · Authors · 2024-11-27
> > > > > > > **A Thank-you Note from the Authors**
> > > > > > >
> > > > > > > Thank you for your strong support! We sincerely appreciate the time and effort you invested in reviewing our paper, as well as the in-depth discussion, which has greatly helped us.

---

### Official Review · Reviewer_8foJ · 2024-10-31

**Soundness:** 3
**Presentation:** 3
**Contribution:** 3
**Rating:** 8
**Confidence:** 3

**Summary:**

In this paper, the authors study the dynamics of shallow ReLU-like neural networks. The analysis starts with the definition of stationary points, saddle points, and two types of local minima for ReLU-like networks, whose loss landscape is characterized by non-differentiability confined in hyper-planes. Then, the concept of an escape neuron is defined, whose existence indicates the possibility of escape. Numerical experiment with synthesized data and vanishing initialization motivates the theory that saddle escaping is driven by small living neurons gaining amplitude following alignment. Further experiments show the impact of small but finite initialization. Last but not least, the condition under which embedding preserves stationarity is given in the non-smooth setting.

**Strengths:**

The authors developed their own notation based on direction derivatives to tame the non-differentiability in ReLU-like networks, allowing them to study the saddle-to-saddle dynamics in the non-differentiable landscape. With theoretical argument and numerical experiment, the authors manage to explain the whole trajectory when training a shallow ReLU-like network with vanishing initialization. Although not every argument in the paper is original, I am happy to see a paper that collects a lot of existing results and presents them in an integrated manner. The authors put a lot of effort into making their result understandable.

**Weaknesses:**

There are constraints for the theoretical analysis, and the experiments are performed on synthesized data instead of real data. Also, I find the definition of stationary point a little counter-intuitive. I have the impression that the definition of stationary point in this work is not exactly the extension of stationary point of smooth function, which is where $f’(x)=0$ regardless of trapping GD or not. It appears to me that Clarke subdifferential the authors mentioned could be better in this regard, in the sense that the origin of both $|x|$ and $-|x|$ are considered to be a stationary point. With the authors’ definition of stationary point, I can imagine that local maxima are usually not stationary points.

**Questions:**

Although I appreciate the experiment in section 4.1.1 very much and agree that it supports the authors’ theory of saddle escaping, I don’t know if the evidence is strong enough to rule out all alternative theories (discovered or not).

Is there any specific reason for choosing $x_k$ and $y_k$ as given in the paper in section 4.1.1?

Line 453/454 should the “does get” be “doesn’t get”?

---

> ### Author Response · Authors · 2024-11-23
>
> Thank you very much for your review! We appreciate that you noted how our method **tame the non-differentiability in ReLU-like networks** and liked our effort to make our result **understandable**.
>
> We made modifications to our script, highlighted in red, and marked with the green-colored text '8foJ' (your reviewer ID) to indicate changes based on your suggestions.
>
> You mentioned a few weaknesses, which we address below:
>
> ---
> >"There are constraints for the theoretical analysis"
>
> We base our theories on minimal assumptions (line 131): (1) $\alpha^-\neq\alpha^+$, so that the activation function is non-differentiable, and (2) $d>1$ so that we can avoid lengthy discussions for degenerate cases. Our proof of the main result (Theorem 4.2) relies on a rigorous examination of Taylor expansion in all directions. In this sense, we believe our theoretical analysis is not subject to much constraint.
>
> Later in the weaknesses section, you also mentioned our characterization of stationarity is "counter-intuitive", and might not identify local maxima as stationary points. We figure this could be the "constraint" that you mention here. However, we believe our non-conventional definition is a strength and exactly suits our needs, which we will explain when answering to that weakness.
>
> ---
> > "the experiments are performed on synthesized data instead of real data."
>
> The experiment is an illustrative example of Theorem 4.2 and Fact 4.8 (originally Consequence 4.7). These theoretical results are justified by proofs rather than experiments. The simplicity of experiments is for better clarity. We need to track the angles and norms of all the weights during all saddle escaping. To clearly present this, we can only use a few training samples in low dimensions. In more complex setups, it will be hard to fully present the training dynamics since there will be much more neuron groups and escaped saddles. Moreover, in ReLU training dynamics papers [1-3], demonstrating theories by thoroughly probing toy examples is a common practice.
>
> To enhance the credibility of our empirical results, we added two more numerical experiments in Appendix I, investigating the training dynamics of networks with larger input/output dimensions.
>
> ---
> >"I have the impression that the definition of stationary point in this work is not exactly the extension of stationary point of smooth function, which is where f′(x)=0 regardless of trapping GD or not."
>
> First, we think that our definition of stationary point should be regarded as an extension of the stationary points of smooth functions since all the stationary points of smooth networks are still stationary points by our definition. The stationary points of smooth functions have zero directional derivatives towards all directions and thus qualify as stationary points by our definition (which states that stationary points should have *non-negative* directional derivatives toward all directions).
>
> Second, a definition for stationarity that can differentiate between whether gradient descent (GD) is trapped or not is more relevant for understanding the optimization process of neural networks. Other notions of stationarity that do not have implications for training dynamics might not be as useful in this context. Several previous works [1,2,4] investigated the training dynamics and noted the existence of GD-trapping points on the ReLU network landscape but were not able to systematically define and characterize such points. Our definition of stationarity fills in this gap in theory. We now also mention this to contextualize our contribution in line 73.
>
> > "It appears to me that Clarke subdifferential the authors mentioned could be better in this regard, in the sense that the origin of both |x| and −|x| are considered to be a stationary point."
>
> As mentioned in our previous answer, a reasonable and useful definition of stationarity for GD should capture the trapping or slowing down of GD in the vicinity of the points of interest. As we can see, when a GD reaches near the origin of $\vert x \vert$, it effectively halts, while the origin of $-\vert x\vert$ will not slow down GD at all. Thus, the former should count as a stationary point while the latter should not, which is why Clarke Subdifferential does not suit our needs.
>
>
> > "With the authors’ definition of a stationary point, I can imagine that local maxima are usually not stationary points."
>
> As mentioned in our previous answers, smooth local maxima are still characterized as stationary points by our definition. However, non-smooth local maxima, like the origin of $-\vert x \vert$, are not since they do not trap GD at all.

---

> > ### Author Response · Authors · 2024-11-23
> >
> > You also mentioned several questions, which we answer below:
> >
> >
> > > "Although I appreciate the experiment in section 4.1.1 very much and agree that it supports the authors’ theory of saddle escaping, I don’t know if the evidence is strong enough to rule out all alternative theories (discovered or not)."
> >
> >
> > Alternative conclusions (see the caption of Figure 4) are precluded via rigorous proof, and the experiments are illustrative examples. Nonetheless, to enhance the credibility of our empirical results, we conducted two more experiments in Appendix I to further visualize and clarify our theories. One studies a network with 3-dimensional input and scalar output. The other studies a network with 2-dimensional input and output. In all of the experiments, we observe that the decrease in loss is always triggered by the movement of escape neurons in the nearby saddle points, which is consistent with our theories.
> >
> > >" Is there any specific reason for choosing xk and yk as given in the paper in section 4.1.1?"
> >
> >
> > This set of $\mathbf x_k$ and  $y_k$’s are picked for illustration because all the stationary points encountered in this case are non-differentiable, which makes this case more interesting, as is noted in line 315. We did a lot more experiments, but their phenomena were similar and consistent with our theory.
> >
> > >"Line 453/454 should the 'does get' be 'doesn’t get'?"
> >
> >
> > Thank you for pointing out this typo. We have changed it. Please refer to line 1727. Since that subsection is slightly digressive, we have deferred it to the appendix.
> >
> >
> > Thank you again for your review! Your comprehensive comments have helped us improve our script. We believe our responses have addressed your concerns and hope you consider raising your score or confidence.
> >
> >
> >
> >
> >
> > **References**
> >
> >
> > [1] Kumar, A., and Haupt, J. Directional convergence near small initializations and saddles in two-homogeneous neural networks, Transactions on Machine Learning Research, 2024.
> >
> >
> > [2] Boursier, E., Pillaud-Vivien, L., and Flammarion, N. Gradient flow dynamics of shallow ReLU networks for square loss and orthogonal inputs, NeurIPS'22.
> >
> >
> > [3] Pesme, S., and Flammarion, N. Saddle-to-saddle dynamics in diagonal linear networks, NeurIPS'24.
> >
> >
> > [4] Chistikov, D., Englert, M., and Lazic, R. Learning a neuron by a shallow ReLU network: Dynamics and implicit bias for correlated inputs, NeurIPS '23.

---

> > > ### Comment · Reviewer_8foJ · 2024-11-25
> > >
> > > Thank you for the reply and revision. Most of my concerns are addressed.
> > >
> > > One comment regarding the stationary point. For smooth functions, a stationary point in calculus refers to a point where the gradient vanishes, independent of any dynamics. Similarly, the concept of a stationary point with the Clarke subdifferential shares this static interpretation. In contrast, I think the authors extend this notion by defining a stationary point in the context of gradient descent dynamics, and this separates their definition from existing works.
> > >
> > > I will keep my rating unchanged.

---

> > > > ### Author Response · Authors · 2024-11-26
> > > > **A Thank You Note From the Authors**
> > > >
> > > > Thank you very much for your kind comment highlighting our extension to previous works! We sincerely appreciate your rating of 8.

---

### Official Review · Reviewer_WEjG · 2024-11-03

**Soundness:** 3
**Presentation:** 3
**Contribution:** 2
**Rating:** 6
**Confidence:** 3

**Summary:**

This paper presents a new definition of stationary point for non-differentiable neural nets caused by ReLU-type activations. It characterizes the relationship between the local minimum and the "escape neuron", giving a sufficient and necessary condition for local minima under scalar output. Based on this notion, the authors provide a new perspective on saddle-to-saddle dynamics and show that escape from a saddle must perturb the escape neuron. Furthermore, the authors investigate the effect of unit replication on the stationary point.

**Strengths:**

The proposed definition is novel and interesting, and it provides a more complete characterization of saddle points in two-layer ReLU neural networks. This new definition can help to deepen our understanding of the loss landscape and training dynamics.

**Weaknesses:**

1. The current studies focus on the two-layer ReLU neural network and it is unclear how the result can be generalized to deep networks, especially since the characteristic of escape neurons can be complicated when the network is deep, it is unclear if we can draw meaningful conclusions there using the proposed notion.

2. The current study focuses on a non-differentiable activation function only, and the conclusion will be trivial if using a differentiable activation function. As more and more smooth activations are used such as GELU and ELU, the result will have less meaningful impact on practice.

3. Fact 4.6 is an important conclusion to discuss the dynamics, but it was provided as an empirical observation. As this paper mainly focuses on theoretical characterization, it would increase the rigor of the analysis by providing a theoretical justification.

**Questions:**

1. The escape neuron condition requires that the output weight of a specific neuron be zero. Does that imply the landscape of scalar output  2-layer neural network is rather trivial? It seems to be sufficient to just perturb the weight of this specific neuron to escape the saddle.

2. Although the authors argue that the sub-differentiable definition is unsuitable for a non-convex function, does it also fail specifically in two-layer neural networks? In particular, it would be helpful to provide an example in the ReLU network to show that the sub-gradient can not properly capture the saddle point. Since the proposed analysis is restricted to the two-layer neural nets, it would be helpful to understand the difference specifically in this setting.

---

> ### Author Response · Authors · 2024-11-23
>
> Thank you very much for your review! We appreciate that you consider our definition that handles stationarity **novel and interesting**.
>
> We also made modifications to our script, highlighted in red, and marked with the brown-colored text 'WEjG' (your reviewer ID) to indicate changes based on your suggestions.
>
> You mentioned a few weaknesses, which we address below.
>
> ---
>
> > "1. The current studies focus on the two-layer ReLU neural network and it is unclear how the result can be generalized to deep networks, especially since the characteristic of escape neurons can be complicated when the network is deep, it is unclear if we can draw meaningful conclusions there using the proposed notion."
>
>
> We have showcased how our method can be used for deeper networks in Appendix B, where we derived directional derivatives for arbitrarily deep networks. Given our method, it should be conceptually easy to perform Taylor expansion based on directional derivatives for deeper networks to characterize and classify the stationary points. But it will be technically involved since there will be terms of much higher orders to control. Nonetheless, if one conducts Taylor expansion using our method, the equivalent concept of "escape neuron" for deeper networks should naturally emerge.
>
> In this paper, we limit our main discussion to shallow networks since it already answers unresolved questions about non-smooth loss landscape [1-3], training dynamics [4-8], and nonsmooth network embedding [9]. These previous works all discussed shallow networks.
>
> ---
>
> > "2. The current study focuses on a non-differentiable activation function only, and the conclusion will be trivial if using a differentiable activation function. As more and more smooth activations are used such as GELU and ELU, the result will have less meaningful impact on practice."
>
> From a practical point of view, ReLU has the advantage of low memory and computation complexity [10]. Thus, it can be advantageous for resource-constrained scenarios. The community has been revisiting ReLU [10,11] recently for its potential for efficiency.
>
> From a theoretical point of view, in our paper, we developed tools for non-smooth, non-convex optimization landscape, which is generally hard and warrants close investigation.
>
> ---
>
> > "3. Fact 4.6 is an important conclusion to discuss the dynamics, but it was provided as an empirical observation. As this paper mainly focuses on theoretical characterization, it would increase the rigor of the analysis by providing a theoretical justification."
>
> Fact 4.6 was proved in Section 9.5 of [8]. For clarity, we now mention this in footnote 4 in P7. Thank you for mentioning this.
>
> ---
> You also raised the following questions, and we will answer them below.
>
>
> >"1. The escape neuron condition requires that the output weight of a specific neuron be zero. Does that imply the landscape of scalar output 2-layer neural network is rather trivial? It seems to be sufficient to just perturb the weight of this specific neuron to escape the saddle."
>
> You are correct about the structure and the descent direction near the saddle point of such networks. This landscape, being non-convex and non-smooth, is typically challenging. There is a long active line of research dedicated to understanding it [1-9], and it was not until our paper that the relative simpleness of its structure is fully revealed. Nevertheless, we point out that the rigorous characterization of the training dynamics on such loss landscape in the general case is still an open question.
>
> As a side note, we provide two new numerical experiments in Appendix I, one of which is for multidimensional outputs and discovers escape neurons whose output weights are not zero.

---

> ### Author Response · Authors · 2024-11-23
>
> >"2. Although the authors argue that the sub-differentiable definition is unsuitable for a non-convex function, does it also fail specifically in two-layer neural networks? In particular, it would be helpful to provide an example in the ReLU network to show that the sub-gradient can not properly capture the saddle point."
>
> The definition of subgradient is the following:
>
>
> >> Let $f: \mathbb{R}^n \to \mathbb{R}$ be a convex function. The subgradient set of $f$ at $\mathbf x_0$ is defined as:
> $$
> \\partial f(\\mathbf x_0) = \\left\\{ \\mathbf g \\in \\mathbb{R}^n : f(\\mathbf x) \\geq f(\\mathbf x_0) + \\mathbf g^T (\\mathbf x - \\mathbf x_0), \\, \\forall \\mathbf x \\in \\mathbb{R}^n \\right\\}.
> $$
>
>
> One can verify that the subgradient cannot be defined for the origin of the loss landscape of our setup in general. Since the directional derivatives toward all the directions are zero at the origin (by Lemma 3.2), if the subgradient $\{\mathbf g\}$ can be defined, it must mean that $\mathbf g = \mathbf 0$. Taking it to the definition of the subgradient, we find that, if the subgradient at the origin is $\{\mathbf g = \mathbf 0\}$, the origin must be a local minimum, which is generally not the case (see all of our numerical examples in Section 4.1.1 and Appendix I).
>
> The origin is a stationary point (since the directional derivatives toward all directions are zero), but not a local minimum or a local maximum (Theorem D.1) in general. Thus, it is a saddle point where subgradients cannot be defined.
>
> We have also added this example to line 1082.
>
> ---
>
> Thank you again for your review! Your valuable suggestions have helped us improve our script. We believe our responses above address your questions, and we hope you consider raising your rating.
>
>
> **References**
>
> [1] Sahs, J., Pyle, R., Damaraju, A., Caro, J. O., Tavaslioglu, O., Lu, A., Anselmi, F., and Patel, A. B. Shallow univariate ReLU networks as splines: Initialization, loss surface, Hessian, and gradient flow dynamics, Frontiers in Artificial Intelligence, 2022.
>
>
> [2] Cheridito, P., Jentzen, A., and Rossmannek, F. Landscape analysis for shallow neural networks: Complete classification of critical points for affine target functions, Journal of Nonlinear Science, 2022.
>
>
> [3] Liu, B., Liu, Z., Zhang, T., and Yuan, T. Non-differentiable saddle points and sub-optimal local minima exist for deep ReLU networks, Neural Networks, 144, 2021.
>
>
> [4] Chistikov, D., Englert, M., and Lazic, R. Learning a neuron by a shallow ReLU network: Dynamics and implicit bias for correlated inputs, NeurIPS '23.
>
>
> [5] Kumar, A., and Haupt, J. Directional convergence near small initializations and saddles in two-homogeneous neural networks, Transactions on Machine Learning Research, 2024.
>
>
> [6] Boursier, E., Pillaud-Vivien, L., and Flammarion, N. Gradient flow dynamics of shallow ReLU networks for square loss and orthogonal inputs, NeurIPS'22.
>
>
> [7] Boursier, E., and Flammarion, N. Early alignment in two-layer networks training is a two-edged sword, 2024.
>
>
> [8] Maennel, H., Bousquet, O., and Gelly, S. Gradient descent quantizes ReLU network features, 2018.
>
>
> [9] Fukumizu, K., Yamaguchi, S., Mototake, Y., and Tanaka, M. Semi-flat minima and saddle points by embedding neural networks to overparameterization, NeurIPS'19.
>
>
> [10] Tayaranian, M., Mozafari, S. H., Clark, J. J., Meyer, B., and Gross, W. Faster inference of integer SWIN transformer by removing the GELU activation, 2024.
>
>
> [11] Wortsman, M., Lee, J., Gilmer, J., and Kornblith, S. Replacing softmax with ReLU in vision transformers, 2023.

---

> ### Author Response · Authors · 2024-11-26
> **Follow-Up on Revision Suggestions**
>
> Thank you again for your constructive comments! As the deadline for revising scripts draws near, we would like to know if you have further suggestions or concerns. We would be pleased to address them.
>
> Best,
>
> Authors

---

> > ### Comment · Reviewer_WEjG · 2024-11-28
> >
> > Thank you for the detailed response. My concern has been properly addressed, and I raised my score to 6.

---

> > > ### Author Response · Authors · 2024-11-28
> > > **A Thank-you Note From the Authors**
> > >
> > > Thank you once again for taking the time and effort to review our paper! We are delighted that our response addressed your doubts and concerns. We sincerely appreciate your support!
> > >
> > > Best regards,
> > >
> > > Authors

---

### Official Review · Reviewer_k7xc · 2024-11-03

**Soundness:** 3
**Presentation:** 3
**Contribution:** 3
**Rating:** 6
**Confidence:** 3

**Summary:**

The paper studies the loss landscape of one hidden layer with ReLU-like activations trained on squared loss. It provides conditions for stationary points that work for both differentiable and non-differentiable parts of the loss landscape. The authors introduce "escape neurons" as the indicator of whether stationary points are local minimums. This conclusion is necessary and sufficient for scalar-output nets. With this characterization, they take a step further to understand the saddle-to-saddle dynamics of two-layer networks. They also discuss how network embedding helps reshape the stationary points.

**Strengths:**

- The ReLU networks' stationary points analysis is novel and interesting. The authors are able to characterize the "escape neurons" as the indicator of two-layer ReLU networks defined with first-order conditions, which can help to characterize whether a stationary point is a strict saddle to escape. This rigorous analysis can help us further understand the saddle-to-saddle dynamics from a landscape perspective.

- The toy experiments verify the theoretical results above with vanishing initialization, both displaying examples for stationary points without escape direction and saddle-escaping phenomena when "escape neurons" exist. The explanation of Figure 4 is intuitive and insightful.

- The escape neuron analysis helps to understand how the interesting "network embedding" technique reshapes the landscape.

**Weaknesses:**

- The analysis for the landscape seems to be a bit limited to the local stationary point analysis as the supplement of [Kumar and Haupt, 2024], precluding other types of saddles. To understand the saddle escaping or saddle-to-saddle dynamics, it would be better to more analyses of gradient dynamics in the neighborhood of saddle points. For example, can we have the escape direction in Corollary 4.4 related to the gradient, implying that GF/GD can help escape the saddle points?

- The discussion of the saddle-to-saddle dynamics with vanishing/finitely small initialization (sec 4.2) looks too heuristic to me. It is better to present those implications or corollaries of the stationary point characterization more formally instead of descriptions like Consequence 4.7.

- As a minor weakness, the toy examples only have one-dimensional inputs (with bias), which is not comprehensive. More empirical results need to be shown to corroborate the theoretical results' implications.

**Questions:**

Is it possible to come up with a more rigorous example (with gradient dynamics analysis) for Figure 4 to understand the saddle-to-saddle dynamics?

---

> ### Author Response · Authors · 2024-11-23
>
> Thank you very much for your review! We appreciate that you consider our analysis to be **novel and interesting** and our presentation to be **intuitive and insightful**.
>
> We made modifications to our script, highlighted in red, and marked with the blue-colored text 'k7xc' (your reviewer ID) to indicate changes based on your suggestions.
>
> You also raised a few concerns, which we address below.
>
> ---
> (1) You mentioned the following as one of the weaknesses:
> > "To understand the saddle escaping or saddle-to-saddle dynamics, it would be better to more analyses of gradient dynamics in the neighborhood of saddle points. For example, can we have the escape direction in Corollary 4.4 related to the gradient, implying that GF/GD can help escape the saddle points?"
>
> You also asked the following question:
> > "Is it possible to come up with a more rigorous example (with gradient dynamics analysis) for Figure 4 to understand the saddle-to-saddle dynamics?"
>
> Dynamics analysis around saddle points is a contribution of previous works [1-3], which our work extends.
>
> Previous works, such as [1,2], gave rigorous examples with gradient dynamics analysis. Concretely, [1] studied a case where the training inputs are all orthogonal, and [2] studied a case where there is only one effective neuron learned. But as [3] pointed out, these previous works could not preclude other types of saddle points, as the techniques they developed for the dynamics in the vicinity of certain saddle points cannot offer insight into the global structure of the loss landscape. We fill this gap in the theory by devising a method to systematically characterize and classify the stationary points on the entire loss landscape.
>
>
> To further contextualize our contribution, we add the following sentence to line 440:
> >>To describe such a process, [1,2] rigorously characterized the gradient flow for toy examples, revealing trajectories consistent with Figure 4.
>
>
> ---
>
> (2) You suggested:
> >"It is better to present those implications or corollaries of the stationary point characterization more formally instead of descriptions like Consequence 4.7."
>
>
> Thank you for this suggestion. We will use more formal explanations.
>
> * Consequence 4.7 (now Fact 4.8) is, in fact, rigorously proved with Taylor expansion, as is stated in Remark F.1. Nevertheless, to clarify the rigorousness of this result, we will mention in line 403 that this consequence is actually proved. We chose not to call it a corollary as it is a partial restatement of Theorem 4.2. But, to avoid confusion, we will now call it a "Fact" instead of a "Consequence" and stress that it is a partial restatement of Theorem 4.2.
>
>
> * Moreover, to avoid ambiguity, we compile our results that preclude other types of saddle points into the following corollary.
> >>Corollary 4.7: Following the setup and notation introduced in Section 2, let $\\mathbf P(t) = (\\mathbf w\_i(t), h\_{j\_0 i}(t))\_{i\\in I}\\in \\mathbb{R}^D$ be the parameter trajectory of a one-hidden-layer scalar-output ReLU network trained with the empirical squared loss $\\mathcal{L}$, where $t \\geq 0$ denotes time. Suppose the trajectory satisfies $\\frac{\\mathrm d\\mathbf P}{\\mathrm d t} = -\\frac{\\partial \\mathcal{L}}{\\partial \\mathbf P}$, in which we specify $\\frac{\\partial \\mathrm{ReLU}(x)}{\\partial x} = \\mathbb{1}\_{\\{x> 0\\}}$ in the chain rule. Let $\mathbf A\in \mathbb{R}^D$ be an arbitrary vector. We initialize the network with $\mathbf{P}(0)=\sigma \mathbf A$. Under these conditions, if $\lim\_{\sigma \searrow 0}\left(\inf\_t\Vert\mathbf P(t) - \overline{\mathbf P}  \Vert\right) = 0$, where $\overline{\mathbf P}= (\overline{\mathbf w}\_i, \overline{ h}\_{j\_0 i})\_{i\in I}$ is a non-minimum stationary point, we must have $\overline{\mathbf w}\_i = \mathbf{0}$ and $\overline{ h}\_{j\_0 i} = 0$ for some $i\in I$.
>
> Please refer to line 383 for more details.
>
> ---
> (3)
> >"the toy examples only have one-dimensional inputs (with bias)... More empirical results need to be shown to corroborate the theoretical results' implications."
>
>
> We supplement our discussion with two new numerical experiments in Appendix I, one with 3-dimensional input and scalar output, the other with 2-dimensional input and output. Both are consistent with our theories.
>
> ---
> Thank you again for your review! Your valuable suggestions have helped us improve our script. We believe our responses above address your questions, and we hope you consider raising your rating.
>
> **References**
>
> [1] Boursier, E., Pillaud-Vivien, L., and Flammarion, N. Gradient flow dynamics of shallow ReLU networks for square loss and orthogonal inputs, NeurIPS'22.
>
>
> [2] Chistikov, D., Englert, M., and Lazic, R. Learning a neuron by a shallow ReLU network: Dynamics and implicit bias for correlated inputs, NeurIPS '23.
>
>
> [3] Kumar, A., and Haupt, J. Directional convergence near small initializations and saddles in two-homogeneous neural networks, Transactions on Machine Learning Research, 2024.

---

> > ### Comment · Reviewer_k7xc · 2024-11-24
> >
> > Thank you for the detailed responses and additional experiments. The additional information addressed my concerns and misunderstandings. I decide to adjust my rating and lean toward acceptance.

---

> > > ### Author Response · Authors · 2024-11-26
> > > **A Thank You Note From The Authors**
> > >
> > > We are glad that you are satisfied with our response and appreciate your support. Thank you for your valuable suggestions and comments!

---

### Author Response · Authors · 2024-12-04
**Summary of Author Feedbacks**

We thank the reviewers again for their time and effort in reviewing our paper. We sincerely appreciate their insightful comments and suggestions, which helped us improve our paper.

During the rebuttal, there are three recurring questions. Here, we reiterate them along with our responses.

**Soundness of the notion of stationary points and the techniques to characterize them (asked by 8foJ,BZd8):** The stationarity notion studied in this paper (which is that stationary points are where there do not exist negative first-order one-sided directional derivatives around) is an extension of the traditional notion of stationarity for smooth functions and a refinement of Clarke stationarity, a standard tool for non-smooth, non-convex analysis [1,2]. We have established with our reviewers that the unconventional stationarity studied in this paper is particularly relevant for understanding the training dynamics, as it excludes points that do not significantly slow down gradient descent. More importantly, to rigorously analyze the non-smooth non-convex loss landscape, we construct a coordinate system with which the loss is always smooth within each orthant (the space between the coordinate axes), rather than following the common practice of using one-sided derivatives on the original axes directly [3]. Such a technique allows for Taylor expansion based on the derivatives along the axes (Appendix G), as it will be like studying a smooth function but limiting the scope to respective orthants. Note that Taylor expansion is otherwise prohibited due to non-smoothness. This framework can also be extrapolated to ReLU-like networks of arbitrary depth (Appendix B), pushing the boundaries of non-smooth non-convex optimization.


**Additional empirical results (asked by k7xc,8foJ):** To enhance the credibility of our empirical results, we have presented two more experiments in Appendix I. These experiments have more input/output dimensions than the one originally presented in the paper and align well with the prediction of our theory.

**Presentation style (asked by k7xc,WEjG, BZd8):** The original submission contained extensive discussions in the main text, which may have blurred the distinction between rigorous and empirical results. To sharpen our presentation, we have moved some discussion to the appendix (Appendix J), explicitly clarified the rigor of some results (footnote 4 and Fact 4.8), and condensed a discussion-based result into a more precise corollary (Corollary 4.7).

Overall, according to the reviewers’ responses, we believe we have adequately addressed their concerns. Our scores were increased from (8,5,5,5) to **(8,8,6,6)**.

### References


[1] Wang, Y., Lacotte, J., and Pilanci, M. The hidden convex optimization landscape of regularized two-layer ReLU networks: an exact characterization of optimal solutions, ICLR'22.

[2] Davis, D., Drusvyatskiy, D., Kakade, S. M., and Lee, J. D. Stochastic subgradient method converges on tame functions, Foundations of Computational Mathematics, 2022.

[3] Cheridito, P., Jentzen, A., & Rossmannek, F. Landscape analysis for shallow neural networks: Complete classification of critical points for affine target functions. Journal of Nonlinear Science, 2022.

---

### Meta-Review · Area_Chair_15Tc · 2024-12-10

**Metareview:**

The paper provides a novel analysis of stationary points in one-hidden-layer ReLU-like neural networks, introducing the concept of escape neurons and characterizing saddle-to-saddle dynamics and the impact of network embeddings. By leveraging directional derivatives and introducing a new coordinate system to handle non-differentiability, the authors offer insights into the training dynamics.

### **Strengths**
- Extends prior work on saddle-to-saddle dynamics, offering rigorous analysis of non-differentiable stationary points using a unique notion of directional derivatives.
- Rigorous proofs support key claims (e.g., characterization of stationary points, saddle dynamics).
- Numerical experiments validate theoretical predictions, exploring cases with multidimensional inputs/outputs.

### **Weaknesses**
- The analysis is restricted to shallow ReLU networks, leaving generalization to deeper or differentiable networks (e.g., with GELU, ELU) as future work.
- Practical experiments use toy datasets and synthesized data, which may limit real-world applicability.
- Some reviewers initially found the main definition counter-intuitive, as it diverges from classical notions like Clarke stationary points. However, the authors clarified its relevance to GD/SGD dynamics.

Overall, the paper provides a rigorous analysis that extends prior work. I believe the paper is of interest and recommend acceptance. I do encourage the authors to continue improving clarity, particularly for broader audiences.

**Additional Comments On Reviewer Discussion:**

Some reviewers initially found the main definition counter-intuitive, as it diverges from classical notions like Clarke stationary points. However, the authors clarified its relevance to GD/SGD dynamics.

---

### Decision · Program_Chairs · 2025-01-22

Accept (Poster)